# H2BK120ub and its reader RNF169 sequentially regulate replication fork remodeling and stability

Filip D Duzanic[1,5], Vaishnavi Mohana-Natarajan[1,2,5], Samuele Fisicaro [1], Collin Bakker [3,4], Moses Aouami[1], Gabriel Amaral [1], Massimo Lopes [1], Nitika Taneja [3,4] & Lorenza Penengo [1✉]

## Abstract

Ubiquitination of the C-terminus of histone H2B (H2BK120ub) is a key histone modification with functions in a wide array of DNA-related processes, best characterized in gene transcription and repair. A role for H2B ubiquitination in DNA replication has been postulated and investigated in yeast but is still elusive in human cells. Here, we uncovered a critical function of H2BK120ub in replication fork dynamics. H2BK120ub is present at replication forks and accumulates upon replication stress in a manner dependent on ATR and RAD51. Loss of RNF20, the main ubiquitin ligase promoting H2BK120ub, leads to RECQ1-mediated unrestrained replication fork progression and defective fork reversal upon mild replication stress, restoring fork stability in BRCA2-deficient cells. Furthermore, we identified RNF169, a factor involved in the DNA damage response and repair, as a reader of the H2BK120ub mark at stalled replication forks, where it is required to protect the nascent DNA from excessive nucleolytic degradation. Hence, RNF20, H2BK120ub and RNF169 are key novel players orchestrating replication stress response and fork plasticity in human cells.

**Keywords** Histone H2B Ubiquitination; DNA Replication Stress Response; Fork Plasticity and Restart; RNF169; RNF20/RNF40
**Subject Categories** Chromatin, Transcription & Genomics; DNA Replication, Recombination & Repair; Post-translational Modifications & Proteolysis

## Introduction

Genomic DNA is neatly packed within the limited nuclear space into chromatin. As basic structural and functional units of chromatin, nucleosomes are composed of a histone core—an H3-H4 tetramer flanked by two H2A-H2B dimers— around which a 147 bp long stretch of DNA is tightly wrapped (Luger et al, 2012; Yadav et al, 2018). Nucleosome structure, which is defined by the histone variants present in its core and their post-translational modifications (histone marks), largely determines the chromatin landscape. All DNA-templated processes occur in chromatin context, which in turn governs the accessibility of DNA to factors involved in these processes (Luger et al, 2012; Yadav et al, 2018).

During DNA replication, nucleosomes ahead of the replication forks are disassembled from parental DNA and swiftly reassembled—with addition of newly synthesized histones—onto the daughter chromatids in the wake of the fork. This process is referred to as chromatin replication and is mediated by the histone chaperone machinery, which is an integrative part of the replisome (Alabert and Groth, 2012; Stewart-Morgan et al, 2020; Alabert et al, 2017). Hence, chromatin replication is tightly coupled and inherently coordinated with DNA replication itself, which is essential for the preservation of chromatin landscape (i.e., epigenome) through mitotic cell division (Alabert and Groth, 2012; Stewart-Morgan et al, 2020; Alabert et al, 2017). The functional interplay between DNA and chromatin replication extends beyond epigenetic mechanisms, as shown by the evidence that the newly synthesized histone supply and the histone chaperone machinery at sites of DNA synthesis dictate replication fork speed, S phase duration and cell cycle progression (Mejlvang et al, 2014; Günesdogan et al, 2014; Dreyer et al, 2024).

To ensure accurate and complete genome duplication, DNA replication forks exhibit remarkable plasticity upon encountering various obstacles to their progression, leading to a condition known as replication stress (Zeman and Cimprich, 2014; Berti et al, 2020). When stressed, replication forks transition from canonical DNA synthesis to alternative modes via three main mechanisms, collectively known as DNA damage tolerance (DDT), namely (i) fork reversal, (ii) PrimPol-mediated repriming, and (iii) translesion DNA synthesis (Berti et al, 2020; Quinet et al, 2021). RAD51 recombinase and DNA translocases from the SNF2 family mediate fork reversal, during which the canonical three-way junction at the fork is remodeled into a four-way junction, by transient reannealing of parental strands and hybridization of nascent DNA strands, giving rise to the regressed arm of reversed forks (Berti et al, 2020; Adolph and Cortez, 2024; Berti and Vindigni, 2016). Regressed arms resemble one-ended DNA double-strand breaks and are tightly protected from excessive resection by several fork-protecting factors, including BRCA1/2 tumor suppressors (Schlacher et al, 2011; Mijic et al, 2017; Schlacher et al, 2012;

[1]University of Zurich, Institute of Molecular Cancer Research, 8057 Zurich, Switzerland. [2]Medical School OWL, Anatomy and Cell Biology, Bielefeld University, 33501 Bielefeld, Germany. [3]Department of Molecular Genetics, Erasmus University Medical Center, Erasmus MC Cancer Institute, Rotterdam, The Netherlands. [4]Oncode Institute, Erasmus University Medical Center, Erasmus MC Cancer Institute, Rotterdam, The Netherlands. [5]These authors contributed equally: Filip D Duzanic, Vaishnavi Mohana-Natarajan.
✉E-mail: penengo@imcr.uzh.ch

Taglialatela et al, 2017; Moro et al, 2023). Fork reversal provides time for the DNA repair machinery to remove the DNA lesion that initially caused fork stalling. This is required for the later resumption of normal DNA synthesis, which is coordinated with stress/damage removal and typically mediated by the RECQ1 helicase (Berti et al, 2013; Zellweger et al, 2015). Overall, fork reversal maintains high-fidelity rates of DNA synthesis in conditions of replication stress at the cost of decreased rates of replication fork progression (Berti et al, 2020; Adolph and Cortez, 2024; Zellweger et al, 2015; Vujanovic et al, 2017; Schmid et al, 2018; Bai et al, 2020).

The roles and dynamics of chromatin assembly at the replication forks under replication stress are largely elusive. Recently, it has been reported that replication stress induces replication heterochromatinization of nascent DNA, as evidenced by increased nucleosome occupancy and accumulation of repressive histone marks (i.e., H3K9me1/2/3), which in turn stabilizes forks and enables efficient recovery (Gaggioli et al, 2023). Importantly, stress-induced accumulation of H3K9 methylation—traditionally considered as an epigenetic mark of heterochromatin—is transient, as these modifications are rapidly erased after release from the stress. Moreover, H3 methylation at K4 and K27 was suggested to promote pathological nascent DNA resection in BRCA1/2-defective cells (Chaudhuri et al, 2016; Rondinelli et al, 2017).

Monoubiquitination of H2B at K120 (H2BK120ub) deposited by the concerted action of RAD6 as an E2 ubiquitin conjugating enzyme and the RNF20/RNF40 E3 ubiquitin ligase complex is a critical histone mark with important roles in gene transcription and DNA repair (Fetian et al, 2024; Marsh et al, 2020). The roles of this histone mark in DNA replication have been studied in yeast, where it was reported to regulate the intra S phase checkpoint and DNA damage tolerance, contributing to replication fork stability upon replication stress (Lin et al, 2014; Hung et al, 2017; Trujillo and Osley, 2012; Northam and Trujillo, 2016). However, although other histone modifications have recently emerged as key modulators of replication fork remodeling and protection in human cells (Lin et al, 2014; Hung et al, 2017; Trujillo and Osley, 2012; Northam and Trujillo, 2016; Bhattacharya et al, 2025), the roles of RNF20/RNF40 and H2BK120ub in the DNA replication stress response and fork dynamics in human cells have not been thoroughly investigated.

Here, we report that H2BK120ub is present at replication forks in human cells where, in an ATR- and RAD51-dependent manner, it transiently accumulates in response to replication stress, presumably via de novo deposition by RNF20/40. Loss of H2BK120ub at the sites of DNA synthesis leads to RECQ1-dependent unrestrained replication fork progression and defective fork reversal under mild replication stress, and it restores nascent DNA stability at stalled forks in BRCA2-deficient cells, suggesting a key role for H2BK120ub in replication fork remodeling and restart. Moreover, we found that H2BK120ub at stalled forks contributes to the recruitment of RNF169, a DNA damage response (DDR) factor, which in turn protects newly synthesized DNA from unscheduled nucleolytic processing of stalled forks.

# Results

## H2BK120ub accumulates at replication forks under replication stress

Histone H2B ubiquitination is known to play key roles in the regulation of DNA metabolism, including DNA replication, transcription and repair. However, most of the previous studies were performed in yeast, representing a simplified system to achieve H2B depletion and reconstitution with mutated forms (Lin et al, 2014; Hung et al, 2017; Trujillo and Osley, 2012; Northam and Trujillo, 2016). In this study, we interrogated the role of histone H2B ubiquitination (H2BK120ub) in replication fork dynamics in human cells. First, we assessed the specificity of the H2BK120ub antibody we used throughout the study by immunoblot and confirmed that it recognizes a band at the size corresponding to ubiquitinated histone H2B while not cross-reacting with the unmodified H2B (molecular weights around 23 and 15 kDa, respectively; Fig. EV1A). Importantly, depletion of the E3 ubiquitin ligase RNF20 (siRNF20), which is the main writer of this histone mark (Kim et al, 2005, 2009), leads to a strong reduction in the H2BK120ub signal (Fig. EV1A), while the total levels of ubiquitination are not affected (Fig. EV1B). We obtained similar results by immunofluorescence (IF) staining, showing that the pan-nuclear staining of H2BK120ub is readily reduced upon siRNF20 (Fig. EV1C,D).

We then sought to monitor the presence of H2BK120ub at the replication forks in unperturbed cells and in conditions of replication stress. For this purpose, we treated cells with either mild dose of the topoisomerase-1 inhibitor camptothecin (CPT; 100 nM, 1 h) to perturb replication fork progression inducing only marginal fork breaking (Zellweger et al, 2015; Vujanovic et al, 2017) or high dose of hydroxyurea, (HU; 4 mM, 2 h), which depletes the cellular pool of dNTPs by inhibiting the ribonucleotide reductase, leading to fork stalling (Petermann et al, 2010). Although these treatments likely cause minimal levels of DSBs, they have been widely used to study replication fork remodeling and stability and are known to induce widespread fork slowing (CPT) and stalling (HU). To follow the H2BK120ub mark throughout the cell cycle, we performed in vivo labeling of cells with the nucleotide analog 5-ethynyl-2-deoxyuridine (EdU), which is incorporated during DNA synthesis. Quantitative IF analysis revealed that the H2BK120ub signal intensity increases in EdU-positive cells upon HU-mediated fork stalling and is drastically reduced in cells depleted of RNF20 (Fig. 1A,B). Interestingly, H2BK120ub is present in different phases of the cell cycle and shows an increase in S phase (i.e., in EdU-positive cells; Fig. EV1E).

To investigate whether H2BK120ub associates with DNA replication forks, we took advantage of the "in situ protein interaction with nascent DNA replication forks" (SIRF), an IF-based assay that combines proximity ligation assay (PLA) with EdU-coupled click chemistry to allow monitoring of proteins in close proximity to newly synthesized DNA (Roy et al, 2018). SIRF analysis showed that H2BK120ub is present at sites of DNA synthesis, increases in case of acute fork stalling (HU), albeit not significantly upon mild replicative stress induced by CPT, and is strongly reduced by RNF20 depletion (Figs. 1C,D and EV1F). Similar results were obtained by using a different siRNA targeting RNF20 or upon depletion of the RNF20 partner RNF40 (Fig. EV1G,H). A similar experimental design and SIRF analysis was adopted to monitor RNF20 localization, showing that the main E3 ligase of H2B is also present at replication forks during S phase (Fig. EV1I,J).

We further corroborated these results via iPOND, a biochemical-based technique that allows the isolation of proteins associated, directly or indirectly, with nascent DNA. It combines

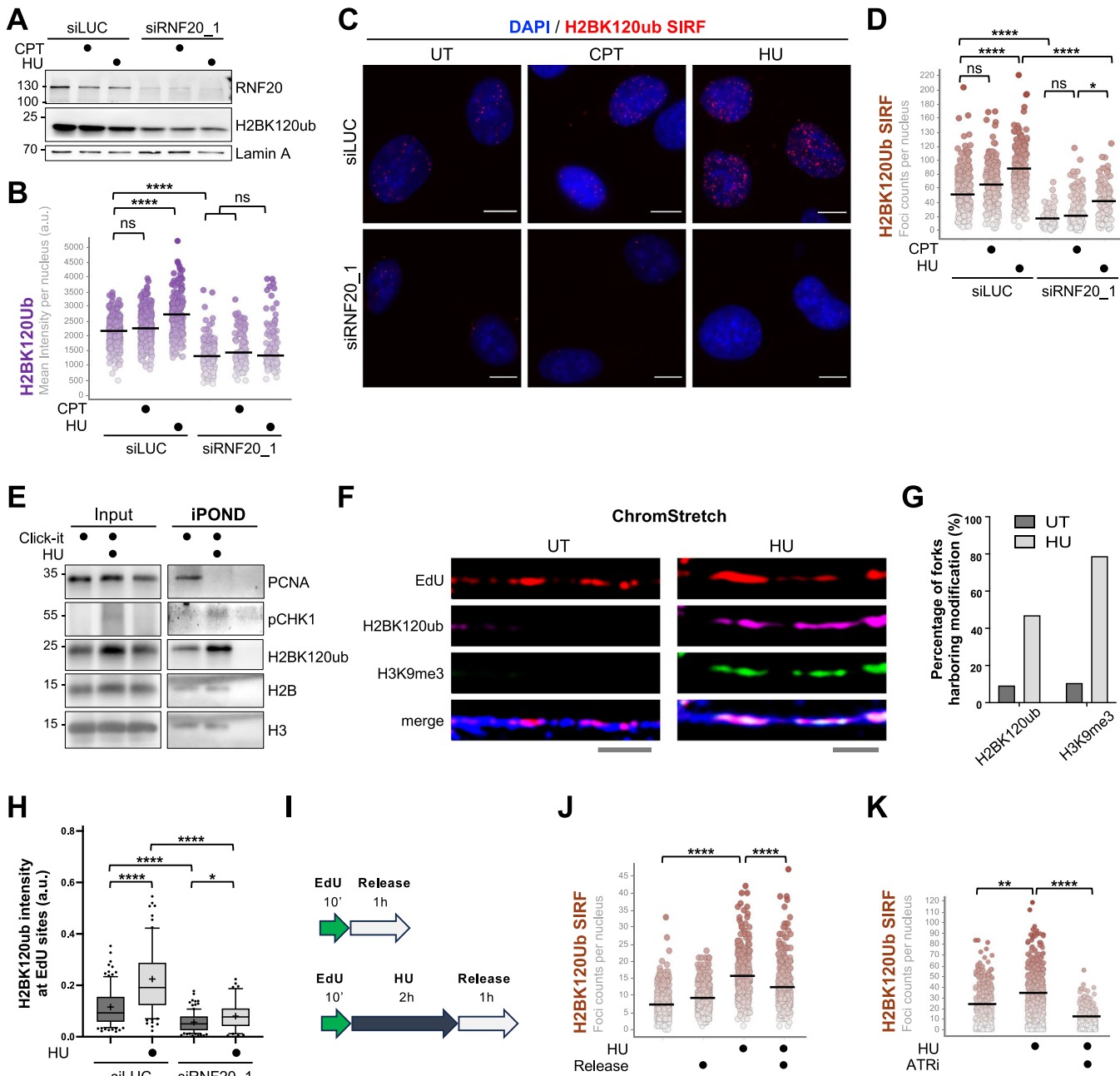

the EdU labeling of nascent DNA with biotin conjugation using click chemistry, which allows streptavidin purification of proteins bound to EdU-labeled DNA (Sirbu et al, 2013). Using iPOND, we could confirm the presence of H2BK120ub at replication forks and its increase in response to replication stress (Fig. 1E). Lastly, we employed ChromStretch, a single-molecule approach recently optimized to isolate and stretch individual single-molecule chromatin fibers and monitor chromatin components on replicated tracks (Gaggioli et al, 2023). We could confirm previous observations, showing the accumulation of the H3K9me3 histone mark at nascent DNA upon replication stress induced by HU treatment (Gaggioli et al, 2023). Remarkably, we found that H2BK120ub also localizes at EdU-positive tracks and accumulates

upon HU-induced fork stalling (Figs. 1F,G and EV1K). RNF20 depletion strongly decreases the presence of H2BK120ub on chromatin but does not affect the H3K9me3 mark (Figs. 1H and EV1L,M). We noticed that the percentage of forks with increased H2BK120ub reached 40–50% in HU across three independent biological replicates, compared to 80–86% for H3K9me3 after excluding constitutive heterochromatin regions of the genome (Fig. 1G). These findings suggest that H2BK120ub marks only a subset of stressed forks, whereas H3K9me3 is present at nearly all stalled replication forks, as previously reported (Gaggioli et al, 2023). This data also indicates that replication stress-induced transient post-translational modifications on nucleosomes can coexist on the same replicated tracks. Similarly

**Figure 1. H2BK120ub transiently accumulates at the stressed DNA replication forks in an ATR-dependent manner.**

(A) Immunoblot analysis of U2OS cell extracts upon siLUC and siRNF20, optionally treated with CPT (100 nM, 1 h) and HU (4 mM, 2 h), using the indicated antibodies. (B) Anti-H2BK120ub immunofluorescence analysis of EdU-labeled U2OS cells treated as in A. Quantification of the average nuclear H2BK120ub signal intensity per EdU-positive cells (at least 120 EdU-positive cells were analyzed per condition; ****$P < 0.0001$, ns nonsignificant, $P$ (siLUC UT vs. siLUC CPT) > 0.9999, $P$(siRNF20_1 UT vs. siRNF20_1 CPT) = 0.8996, $P$ (siRNF20_1 UT vs. siRNF20_1 HU) = 0.1532, $P$ (siRNF20_1 CPT vs. siRNF20_1 HU) > 0.9999. $P$ values were calculated by one-way ANOVA, followed by the Kruskal–Wallis test. Bars represent median values. $n = 3$ experiments. (C, D) Representative images of the H2BK120ub SIRF (in red; scale bars: 10 µm) in EdU-labeled U2OS cells treated as in (A), and quantification of nuclear SIRF foci per EdU-positive cell (more than 120 EdU-positive cells were analyzed per condition; ****$P < 0.0001$, *$P < 0.05$, ns nonsignificant, $P$ (siLUC UT vs. siLUC CPT) > 0.9999, $P$ (siRNF20_1 UT vs. siRNF20_1 CPT) = 0.1059, $P$ (siRNF20_1 CPT vs. siRNF20_1 HU) = 0.0240; $P$ values were determined using one-way ANOVA, followed by Kruskal–Wallis test). Bars indicate median values. $n = 3$ experiments. (E) Analysis of proteins associated with nascent DNA by the iPOND technique. EdU-pulse labeled (10 µM, 10 min), untreated or HU-treated (4 mM, 2 h) HEK293T cells were processed as described in Materials and Methods and analyzed by immunoblot using the indicated antibodies. No-click sample was used as a negative control. H2B and H3 were used as the loading controls. $n = 3$ experiments. (F) Representative images of chromatin fibers acquired by ChromStretch and stained for EdU (red), H2BK120ub (magenta), H3K9me3 (green) and H3 (blue). Scale bars: 5 µM. (G) Bar chart indicating percentage of replication forks (EdU-positive DNA fibers) harboring H2BK120ub and H3K9me3 histone marks in unperturbed and HU-treated (4 mM, 1 h) U2OS cells. $n = >70$ fibers examined per condition for each of three independent experiments with similar results. (H) Whiskers plot indicating H2BK120ub signal intensity at EdU-labeled chromatin fibers upon siLUC or siRNF20, in untreated and HU-treated (4 mM, 1 h) U2OS cells. ****$P < 0.0001$, *$P < 0.05$, $P$ (siRNF20_1 UT vs siRNF20_1 HU) = 0.0308; $P$ values were determined using the Kruskal–Wallis test, followed by Dunn's multiple comparisons test. Box plots show the 25th, 50th (median), and 75th percentiles; whiskers indicate the 10th and 90th percentiles; "+" represents the mean. $n = 3$ experiments. (I, J) Graphical illustration of the experimental design and H2BK120ub SIRF analysis in EdU-labeled U2OS cells upon optional HU treatment (4 mM, 2 h) and 1 h release. At least 150 cells were analyzed per condition. ****$P < 0.0001$, $P$ values determined using one-way ANOVA coupled with the Kruskal–Wallis test. $n = 3$ experiments. (K) H2BK120ub SIRF analysis in HU-treated (4 mM, 2 h) U2OS cells upon optional ATR inhibition (ATRi Ceralasertib, 1 µM 1 h pretreatment + 1 µM during the treatment). Quantification of H2BK120ub SIRF foci per EdU-positive cells (at least 120 EdU-positive cells were analyzed per condition; **$P < 0.01$, ****$P < 0.0001$, $P$ (UT vs. HU) = 0.0034; $P$ values were calculated using one-way ANOVA coupled with Kruskal–Wallis test). Bars indicate median values. $n = 2$ experiments. Source data are available online for this figure.

to H3K9me3, the replication stress-induced accumulation of H2BK120ub at replication forks (marked by EdU incorporation) is transient and decreases 1 h after HU release (Figs. 1I,J and EV1N). Finally, we tested whether the HU-dependent accumulation of H2BK120ub at the replication forks depends on ATR activation and found that treating cells with ATR inhibitor (ATRi) Ceralasertib (1 µM) strongly reduced the accumulation of H2BK120ub on nascent DNA (Figs. 1K and EV1O).

Taken together, these data indicate potential roles for RNF20/H2BK120ub in DNA replication and replication stress response in human cells, extending previous observations in yeast (Northam and Trujillo, 2016).

## Loss of RNF20/H2BK120ub leads to unrestrained replication fork progression under stress

Evidence in yeast also suggested a role for H2BK120Ub in limiting DNA synthesis in replication stress conditions (Lin et al, 2014). To ascertain a possible role of RNF20 in controlling replication fork progression in human cells, we first investigated whether RNF20 depletion undermines the integrity of replication forks, by examining PCNA levels at the newly synthesized DNA. SIRF analysis confirmed the previously reported decrease in PCNA levels at stalled forks in control cells, as evidenced by quantitative analysis of PLA signal intensity (Figs. 2A,B and EV2A). Notably, RNF20 depletion did not cause detectable PCNA dissociation from replication forks, thus ruling out potential major perturbations of replisome stability in RNF20-depleted cells. Moreover, to exclude significant alterations of cell cycle progression upon RNF20 depletion, we performed flow cytometry analysis of EdU-pulse-labeled cells. The results showed that RNF20 depletion did not cause a dramatic alteration of the cell cycle progression, but only a limited delay of cell entry into the S phase and consequently a mild accumulation in G1 phase (Fig. EV2B).

We then used the single-molecule DNA fiber spreading assay (Nieminuszczy et al, 2016) to investigate the effect of RNF20 depletion on different aspects of DNA replication fork progression

and dynamics. This method relies on sequential labeling of newly replicated DNA by providing cells with halogenated nucleotides—5-iodo-2-deoxyuridine (IdU) and 5-chloro-2'-deoxyuridine (CldU)—detectable by specific antibodies. First, we addressed the role of H2BK120ub in normal DNA replication by measuring tract length on DNA fibers, and found that it did not differ significantly between control (siLUC) and RNF20-depleted (siRNF20) cells (Figs. 2C and EV2C), indicating that reduced levels of H2BK120ub do not affect DNA replication under unperturbed conditions. Next, we assessed the possible involvement of H2BK120ub in the stability of newly synthesized DNA by performing a modified version of the DNA fiber spreading to measure replication fork degradation upon fork stalling. Cells sequentially labeled with CldU and IdU were treated with a high dose of HU (4 mM, 4 h) to pause DNA replication (see scheme in Fig. 2D). In control cells, the nascent DNA is protected from degradation by several factors (Schlacher et al, 2011; Mijic et al, 2017; Schlacher et al, 2012; Taglialatela et al, 2017; Moro et al, 2023), and hence no extensive degradation of stalled forks is observed, resulting in the ratio between IdU (green) and CldU (red) of approximately 1 (Figs. 2D and EV2D). RNF20 depletion did not result in significant degradation of stalled forks, which, in contrast, is clearly detectable following BRCA2 loss, a condition characterized by highly unstable replication forks due to extensive nucleolytic degradation (Schlacher et al, 2011), leading to a shortening of IdU tracts and a reduced IdU/CldU ratio. Lastly, we sought to investigate the effect of RNF20 depletion on DNA replication upon mild CPT treatment that imposes hindrances to replication fork progression without detectable DNA damage (Zellweger et al, 2015). In this case, upon CldU labeling, cells were incubated with IdU either in the presence or absence of CPT (100 nM), and the ratio between IdU and CldU was measured to assess the replication slowdown induced by CPT treatment (see scheme in Fig. 2E). As expected, replication fork speed was significantly reduced in control cells treated with CPT. Conversely, RNF20- or RNF40-depleted cells exhibited unrestrained fork progression, maintaining an IdU/CldU ratio around 1 (Figs. 2E,F and EV2E).

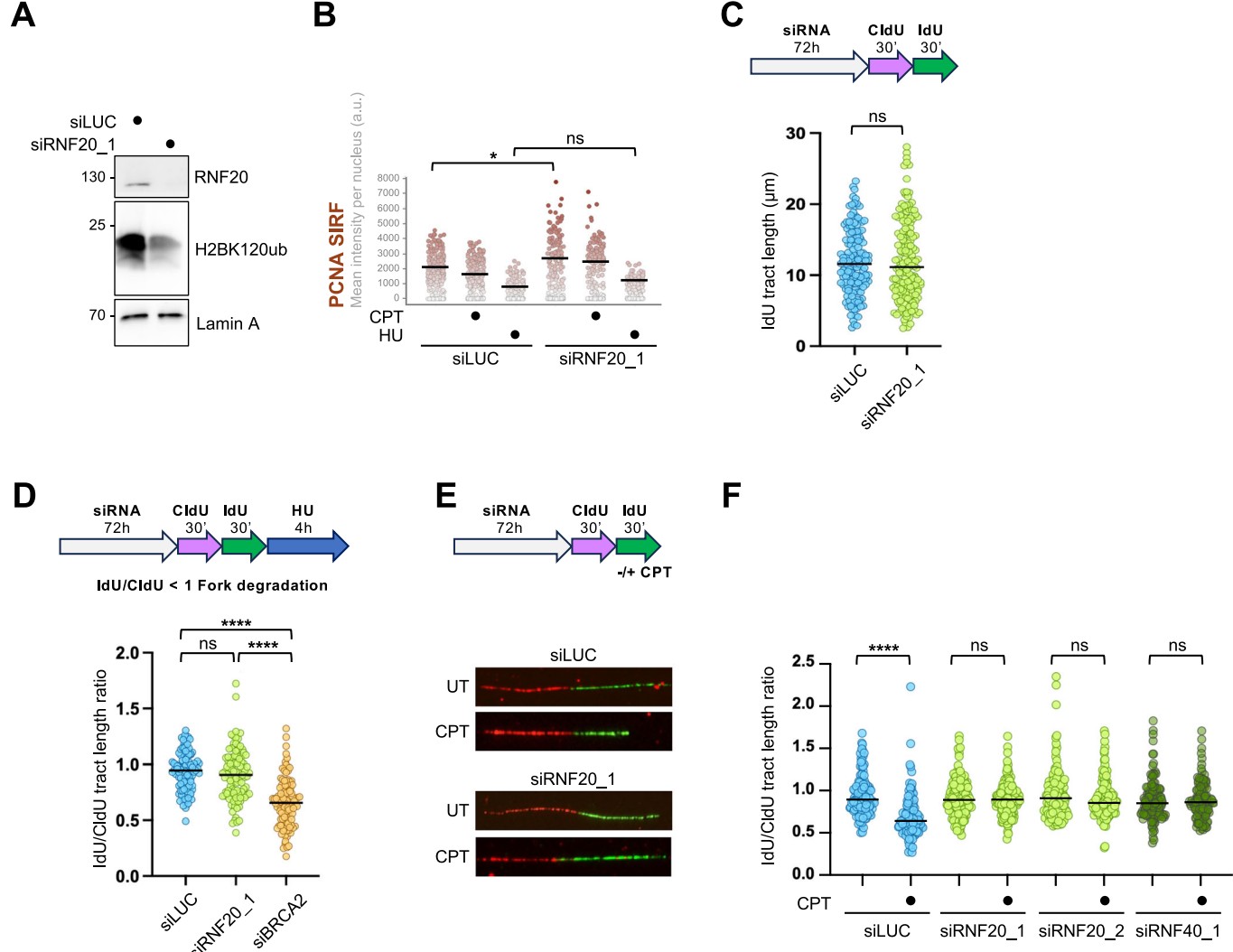

**Figure 2. Loss of RNF20/H2BK120ub leads to unrestrained fork progression.**

(A) Immunoblot analysis of RNF20 and H2BK120ub levels in siLUC- and siRNF20-transfected U2OS cells. Lamin A is used as a loading control. (B) SIRF detection of PCNA at DNA replication forks in unperturbed, CPT-treated (100 nM, 1 h) and HU-treated (4 mM, 2 h) control (siLUC) and RNF20-depleted cells. Quantification of PCNA SIRF signal intensities per EdU-positive cells (more than 120 EdU-positive cells were analyzed per condition; *P < 0.05, ns nonsignificant. P (siLUC UT vs. siRNF20_1 UT) = 0.0387, P (siLUC HU vs. siRNF20_1 HU) = 0.6574; P values determined by one-way ANOVA and Kruskal–Wallis test. Bars represent median values. n = 2 experiments. (C) DNA fibers analysis of replication fork progression in siLUC and siRNF20-transfected U2OS cells. Upper panel: graphical illustration of experimental design. Lower panel: scatter plot representing individual IdU tract lengths (more than 120 fibers were analyzed per condition; ns nonsignificant, statistical analysis performed by unpaired t-test). Bars represent median values. n = 3 experiments. (D) DNA fibers analysis of replication fork protection in HU-treated (4 mM, 4 h) U2OS cells transfected with the indicated siRNAs. Upper panel: schematic of the fork degradation DNA fiber assay. Lower panel: statistical analysis of the IdU/CldU tract length ratios of individual DNA fibers (****P < 0.0001, ns nonsignificant, P (siLUC vs. siRNF20_1) = 0.6441; P values calculated by one-way ANOVA, followed by Kruskal–Wallis test). Bars indicate median values. BRCA2-depleted cells were used as a positive control for extensive fork degradation. n = 3 experiments. (E) DNA fibers analysis of replication fork progression in unperturbed conditions and following mild replicative stress (CPT 100 nM, 30 min). Upper panel: scheme depicting experimental design. Lower panel: representative DNA fibers. (F) Scatter plot representing IdU/CldU tract length ratios of individual DNA fibers in indicated conditions (more than 100 fibers analyzed per condition; ****P < 0.0001, P (siRNF20_1 UT vs. siRNF20_1 CPT) = 0.4416, P (siRNF20_2 UT vs. siRNF20_2 CPT) = 0.9875, P (siRNF40_1 UT vs. siRNF40_1 CPT) > 0.9999; P values calculated using one-way ANOVA, followed by Kruskal–Wallis test). Bars represent median values. n = 3 experiments. Source data are available online for this figure.

Overall, these results show that loss of RNF20/RNF40 and consequent lack of de novo deposition of H2BK120ub impairs the ability to actively pause replication fork progression in response to replicative stress, suggesting the involvement of the histone mark H2BK120ub in the regulation of DNA replication when ongoing forks are challenged by genotoxic stress.

## Loss of RNF20/H2BK120ub perturbs replication fork dynamics

The observed unrestrained fork progression in conditions of replicative stress has been repeatedly linked to defects in replication fork reversal ((Berti et al, 2020) and references therein). To test this

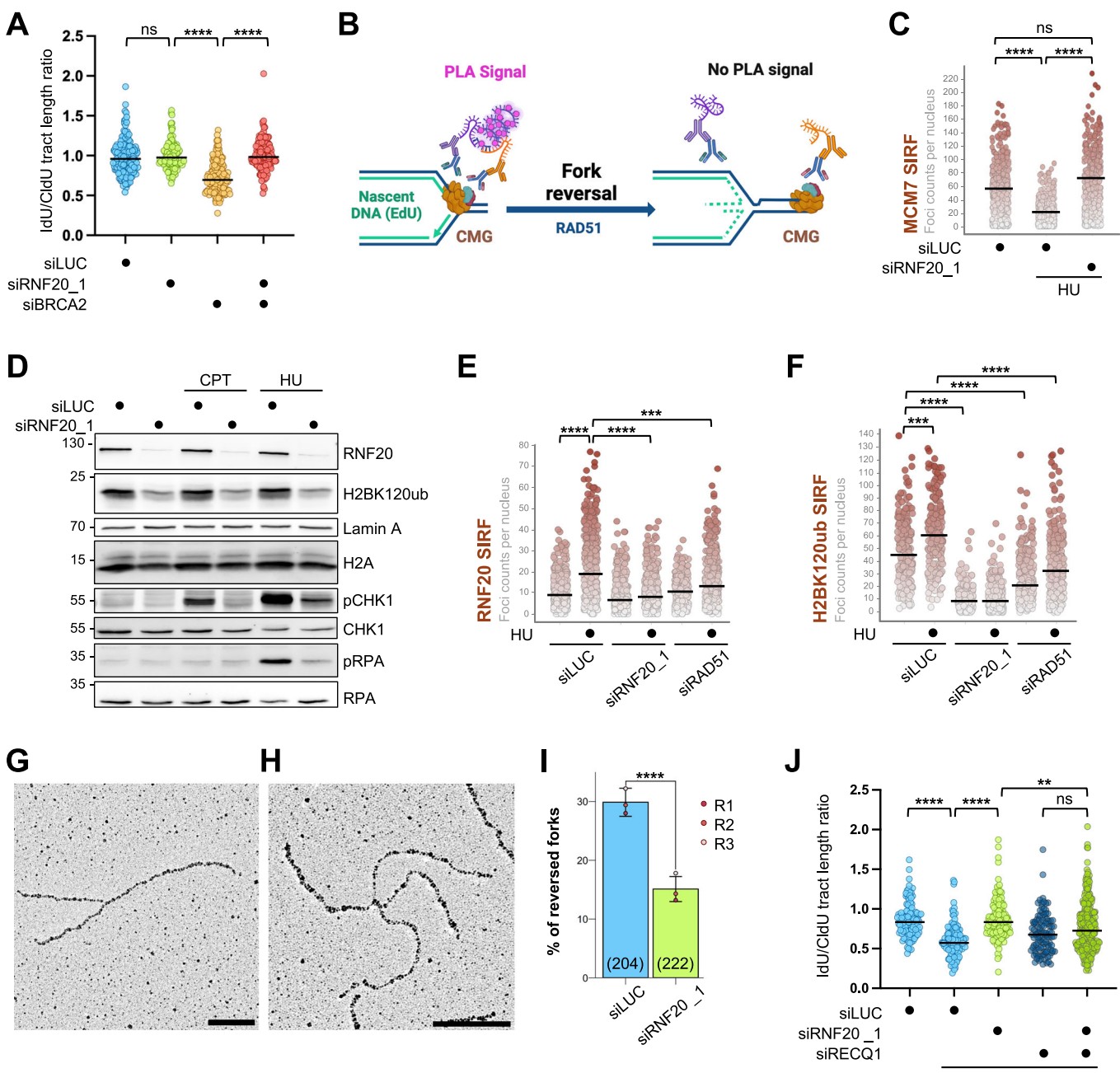

hypothesis, we employed a genetic approach to determine whether RNF20-depleted cells exhibited aberrant fork remodeling capabilities. It has been reported that the degradation of nascent DNA following BRCA2 inactivation occurs at the regressed arms that form upon reversal of HU-stalled forks. Indeed, blocking fork reversal—for instance by inactivating the fork remodeling factors SMARCAL1, ZRANB3 or RAD51 itself—results in the restoration of fork stability in BRCA2-deficient cells (Mijic et al, 2017; Taglialatela et al, 2017; Lemaçon et al, 2017; Kolinjivadi et al, 2017). Hence, to explore the implications of RNF20/H2BK120ub in fork reversal, we tested the effect of RNF20 loss on nascent strand stability in BRCA2-deficient cells. The results showed that depletion of RNF20 restores replication fork stability in BRCA2-

depleted cells, presumably by impairing H2BK120ub deposition and formation or accumulation of reversed forks, acting as an entry point for the consequent deregulated resection of newly synthesized DNA (Figs. 3A and EV3A).

The transaction occurring during fork reversal leads to an increased distance between the nascent DNA, formed by the two daughter strands that annealed and reversed, and the core replicative CMG (Cdc45, MCM2-7, and GINS) complex. This event can be monitored by SIRF analysis using components of the CMG complex, such as the MCM helicases (Fig. 3B; (Liu et al, 2023)). Hence, to further address if RNF20/H2BK120ub loss affects the formation of regressed arms, we performed the SIRF assay using an antibody specific for MCM7. Fork stalling induced by HU

◀ **Figure 3.  Loss of RNF20/H2BK120ub perturbs replication fork dynamics.**

(A) Fork degradation analysis of U2OS cells transfected with the indicated siRNAs. Quantification of IdU/CldU tract lengths ratios of individual DNA fibers (more than 120 fibers analyzed per sample; ****$P < 0.0001$, ns nonsignificant, $P$ (siLUC vs. siRNF20_1) > 0.9999; $P$ values determined using one-way ANOVA, followed by Kruskal–Wallis test). Bars represent median values. $n = 3$ experiments. (B) Graphical illustration of the CMG complex localization at DNA replication forks relative to the nascent DNA strands at progressing (left) and reversed (right) replication forks and the outcome of CMG-EdU PLA signal formation, as reported in (Liu et al, 2023). (C) MCM7 SIRF detection at DNA replication forks in HU-treated (4 mM, 2 h) U2OS cells transfected with the indicated siRNA molecules. Quantification of MCM7 SIRF foci numbers per EdU-positive cells (more than 120 EdU-positive cells were analyzed per condition; ****$P < 0.0001$, ns nonsignificant, $P$ (siLUC UT vs. siRNF20_1 HU) > 0.9999; $P$ values calculated by one-way ANOVA and Kruskal–Wallis test). Bars indicate median values. $n = 3$ experiments. (D) Immunoblot analysis using the indicated antibodies of extracts derived from control (siLUC) and siRNF20-transfected U2OS cells in unperturbed, CPT (100 nM, 1 h) and HU (4 mM, 2 h). Lamin A and histone H2A were used as loading controls. $n = 3$ experiments. (E, F) RNF20 and H2BK120ub detection at DNA replication forks by SIRF in untreated and HU-treated (4 mM, 2 h) U2OS cells transfected with siLUC, siRNF20 or siRAD51. Dot plot representing RNF20 (E) and H2BK120ub (F) SIRF foci numbers in EdU-positive cells (at least 120 EdU-positive cells were analyzed per condition; ***$P < 0.001$, ****$P < 0.0001$, $P$ (siLUC HU vs. siRAD51 HU) = 0.0004 for RNF20 SIRF, $P$ (siLUC UT vs. siLUC HU) = 0.0009 for H2Bub SIRF; $P$ values determined by one-way ANOVA coupled with Kruskal–Wallis test). Bars represent median values. $n = 3$ experiments. (G–I) Electron micrographs of a normal replicating fork (three-way junction; G) and a reversed fork (four-way junction; H), and frequency of reversed replication forks isolated from U2OS cells upon siLUC and siRNF20 transfection and treatment with HU (4 mM, 2 h). The number of total forks analyzed across three experiments, marked with different colors, is indicated in brackets. Error bars represent the standard deviation around the mean (siLUC: ±2.13%; siRNF20_1: ±2.39%). Scale bars = 100 nM. (J) DNA fibers analysis of replication fork progression following mild replicative stress (CPT 100 nM, 30 min). Scatter plot representing IdU/CldU tract length ratios of individual DNA fibers in indicated conditions (more than 100 fibers analyzed per condition; ****$P < 0.0001$, **$P < 0.01$, $P$ (siRNF20_1 CPT vs. siRNF20_1 + siRECQ1 CPT) = 0.0012, $P$ (siRECQ1 CPT vs. siRNF20_1 + siRECQ1 CPT) = 0.9545; $P$ values calculated using one-way ANOVA, followed by Kruskal–Wallis test). Bars represent median values. $n = 3$ experiments. Source data are available online for this figure.

treatment readily decreased the SIRF signal formation in control cells, indicative of the dissociation between MCM7 (i.e., the replisome) and EdU-labeled nascent DNA of regressed arms (Figs. 3C and EV3B,C). Surprisingly, MCM7-EdU SIRF signal in RNF20-depleted cells remained at a similar level as untreated control cells (siLUC). These results indicate that in the absence of RNF20 and H2BK120ub, the CMG complex remains in close association with the nascent DNA, further supporting the requirement for RNF20 and H2BK120ub for the formation and/or stabilization of reversed replication forks. To further elucidate the role of H2BK120ub in response to replication stress, we investigated the effect of RNF20 loss on checkpoint activation. Immunoblot analysis showed that levels of phosphorylated RPA (S4/S8) as well as phosphorylated CHK1 (S345)—i.e., the canonical markers of the replication stress response—are readily induced following CPT and HU treatments and significantly reduced in cells depleted of RNF20 compared to the control cells (Fig. 3D). Although this does not necessarily reflect defective fork reversal, similar checkpoint alterations were previously reported in HLTF-KO cells that show impaired fork reversal (Bai et al, 2020).

Next, to investigate whether RNF20 and H2BK120ub accumulation could be linked to the process of fork reversal, we performed SIRF assays to monitor their presence at HU-stalled replication forks in RAD51-depleted cells, which are reportedly defective in fork reversal (Zellweger et al, 2015). Remarkably, the association of both RNF20 and H2BK120ub with replication forks was largely reduced in RAD51-depleted cells, mimicking what can be observed in RNF20-depleted cells (Figs. 3E,F and EV3D-F), and suggesting that H2BK120ub is implicated in the formation or stabilization of reversed forks downstream of RAD51. Vice versa, depletion of RNF20 did not markedly affect the presence of RAD51 on nascent DNA (Fig. EV3G,H). To directly address the role of H2BK120ub in replication fork reversal and/or restart, we used an established method for direct visualization of replication intermediates by electron microscopy (Zellweger and Lopes, 2018), which allows distinguishing normal (Fig. 3G) and reversed (Fig. 3H) replication forks. In three independent experiments, we observed a consistent and significant reduction in the number of detectable reversed forks

in HU-treated RNF20-depleted cells (Figs. 3I and EV3I). Moreover, depletion of RECQ1— previously implicated in the restart of reversed forks —rescued the unrestrained fork progression associated with RNF20 depletion upon CPT treatment (Figs. 3J and EV3J). Collectively, these data strongly suggest that H2BK120ub deposition on nascent DNA stabilizes reversed replication forks, by preventing their untimely restart by RECQ1.

## H2BK120ub serves as a docking platform for RNF169

Building on these findings, we then proceeded to investigate which factors are recruited to the replication forks under stress conditions by interacting with H2BK120ub. Several factors have been reported to be specifically or preferentially recruited on chromatin carrying H2BK120ub (Shema-Yaacoby et al, 2013). Interestingly, among these factors is RNF169, a potential ubiquitin ligase involved in the DNA repair pathway choice (Wilson and Durocher, 2017; Poulsen et al, 2012; Hu et al, 2017). RNF169 was shown to be recruited to chromatin upon DSBs by direct interaction with the nucleosomes carrying the histone H2A ubiquitinated at lysine 13 and 15 (H2AK13ub and H2AK15ub), histone marks promoted by the key DDR ubiquitin ligase RNF168 (Gatti et al, 2012; Mattiroli et al, 2012). This interaction is mediated by a bipartite functional module in RNF169, composed by the ubiquitin binding domain (MIU2) and the juxtaposed LR motif or LRM (Fig. 4A), which interacts with the H2A-H2B acidic patch (Penengo et al, 2006; Poulsen et al, 2012; Panier et al, 2012; Hu et al, 2017), making the binding of RNF169 specific for ubiquitinated histones in the context of chromatin. The presence of the MIU2-LRM module renders RNF169 a promising candidate for binding to H2BK120ub in the context of replication stress.

We first aimed to validate this possible interaction by performing a pulldown assay using the C-terminal domain of RNF169, encompassing the MIU2-LRM module, fused to the maltose-binding protein (MBP) and the nucleosome core particles (NCPs) either unmodified or carrying the H2BK120ub. Western blot analysis of the pulldown fractions revealed that, while RNF169 cannot bind unmodified NCPs (Fig. EV4A), it readily binds NCPs-

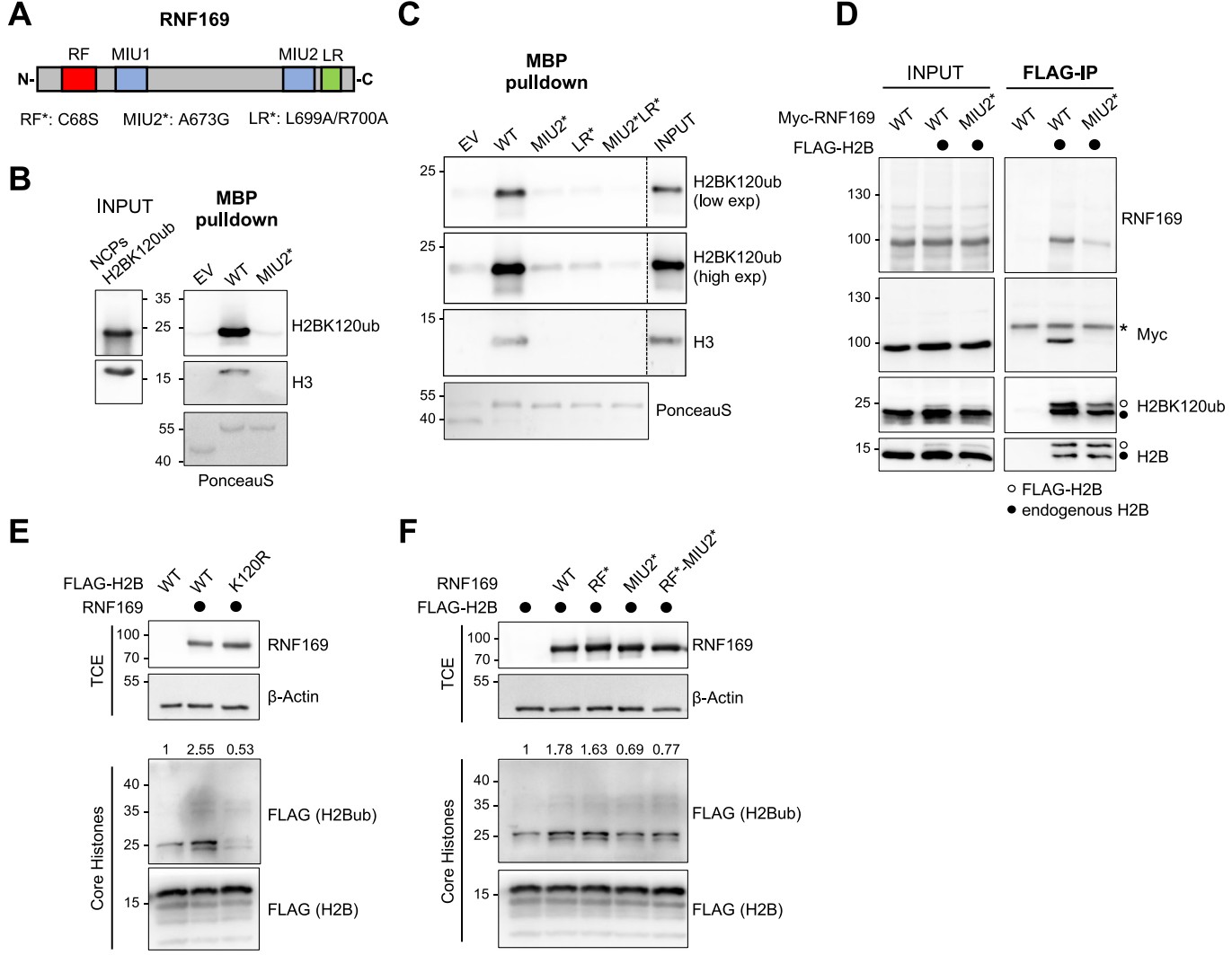

**Figure 4.  RNF169 is a reader of H2BK120ub.**

(A) Schematic illustration of RNF169 structure. RF: RING finger; MIU1/2: motif interacting with ubiquitin 1/2; LR: nucleosome-binding motif. (B) Pulldown assay using MBP constructs fused with C-terminal RNF169 (amino acid residues 662–708) in WT or MIU2* (A673G) mutated form and nucleosomes core particles (NCPs) encompassing H2BK120ub followed by immunoblot with the indicated antibodies. $n = 3$ experiments. (C) MBP pulldown assay as in (B) using WT, MIU2*, LR* (L699A/R700A), and the double MIU2*LR* mutants and nucleosomes core particles (NCPs) encompassing H2BK120ub followed by immunoblot with the indicated antibodies. $n = 3$ experiments. (D) HEK293T cells were transiently transfected with the indicated plasmids, and nuclear extracts were subjected to micrococcal nuclease (MNase) digestion to release nucleosomes and associated nucleosome-binding factors. Immunopurification was performed using anti-FLAG resin on the MNase-released fractions to isolate intact nucleosomes incorporating exogenous FLAG-H2B. The purified samples were then analyzed by immunoblot with the indicated antibodies. Asterisks indicate nonspecific signals. $n = 3$ experiments. (E) HEK293T transiently transfected with plasmids expressing FLAG-H2B WT or the ubiquitination-deficient mutant K120R, together with RNF169 or the empty vector, were subjected to acidic extraction to enrich for core histones. The ubiquitination status of exogenous H2B was monitored by anti-FLAG immunostaining. Total cell extracts (TCE) were analyzed by immunoblot to monitor RNF169 expression. β-Actin was used as a loading control. $n = 3$ experiments. (F) Similar to (E), core histones and TCEs of HEK293T transiently co-transfected with FLAG-H2B and the indicated RNF169 forms were analyzed by immunoblotting. RNF169 immunoblot revealed the expression levels of different variants. β-Actin was used as a loading control. $n = 3$ experiments. In (E, F), band intensities of FLAG (25 kDa), corresponding to monoubiquitinated H2B, were quantified using ImageJ and normalized to FLAG (15 kDa). Data were expressed as relative protein levels. Source data are available online for this figure.

H2BK120ub (Fig. 4B). Importantly, this interaction was abolished by introducing a single amino acid substitution (A673G) in the alpha helix of the MIU2 (referred as MIU2*), indicating that the ubiquitin binding ability of RNF169 is required to recognize nucleosomes containing H2BK120ub. To further assess the specificity of the interaction of RNF169 with ubiquitinated histone H2B, we tested the contribution of the LRM by introducing the

point mutation L699A/R700A. Notably, inactivation of the LRM highly impairs the binding to NCPs-H2BK120ub (Fig. 4C), strongly indicating that RNF169, via the MIU2-LRM module, specifically interacts with H2BK120ub in the chromatin context.

To further validate RNF169 binding to the NCPs-H2BK120ub in cells, we sought to isolate nucleosomes from HEK293T cells upon the ectopic expression of FLAG-tagged H2B and Myc-tagged

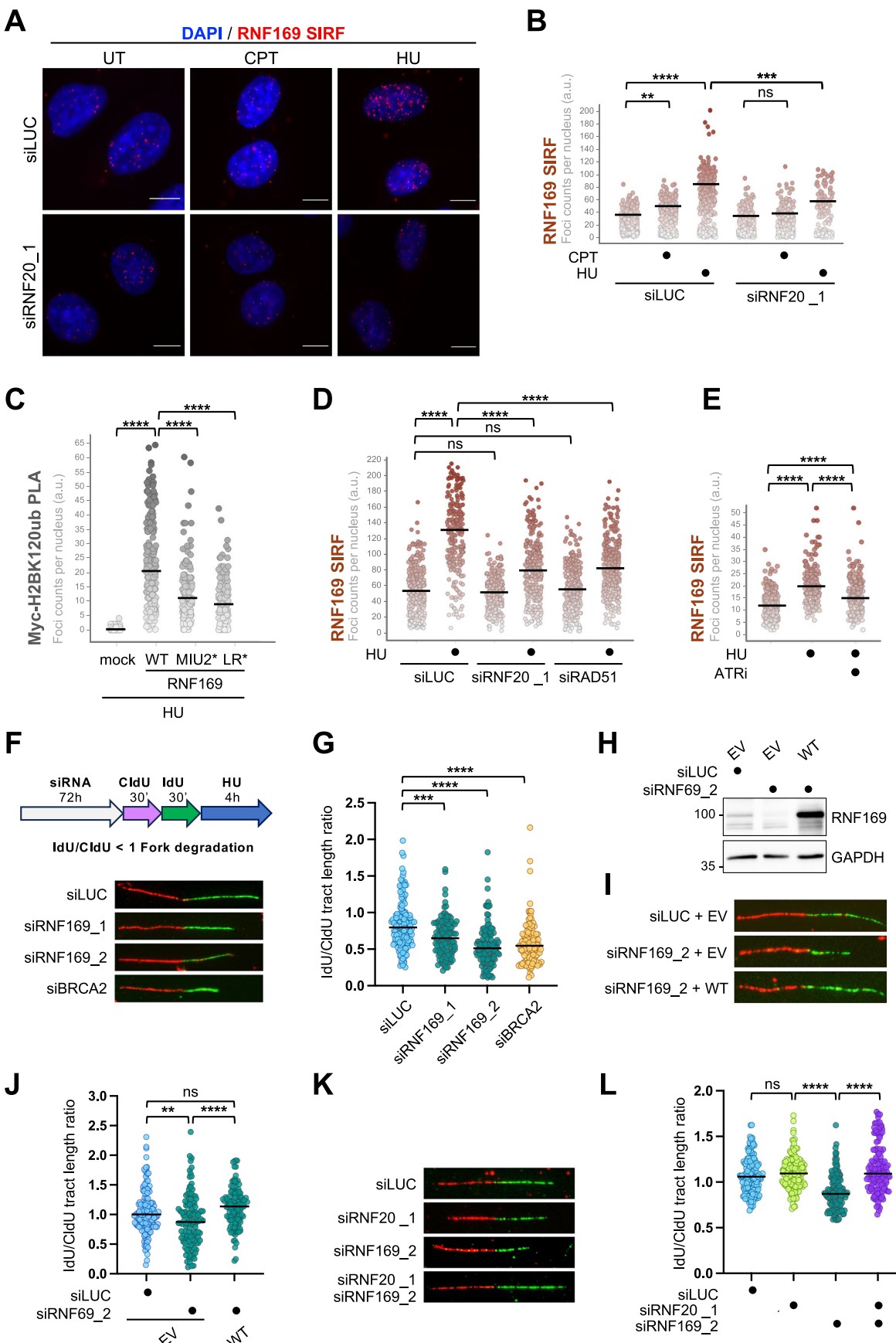

**Figure 5. RNF169 is recruited to the stressed DNA replication forks via H2BK120ub to limit fork resection.**

(A, B) Representative images and quantification of RNF169 SIRF foci (in red; scale bars: 10 μm) in EdU-labeled U2OS cells upon optional treatment with CPT (100 nM, 1 h) or HU (4 mM, 2 h), following siLUC or siRNF20_1 transfection. Dot plot represents RNF169 SIRF foci in EdU-positive cells across different conditions (more than 120 EdU-positive cells were analyzed per condition; ****$P < 0.0001$, **$P < 0.01$, ns nonsignificant, $P$ (siLUC UT vs. siLUC CPT) = 0.0038, $P$ (siRNF20_1 vs. siRNF20_1 CPT) = 0.0611, $P$ (siLUC HU vs. siRNF20_1 HU) = 0.0004; $P$ values calculated using one-way ANOVA, followed by Kruskal–Wallis test). Bars indicate median values. $n = 3$ experiments. (C) Quantification of the PLA foci indicating proximity between H2BK120ub and Myc-tagged RNF169 WT, the MIU2*, and the LR* mutant in cells treated with HU (4 mM, 2 h). At least 150 cells were analyzed per condition. ****$P < 0.0001$, $P$ values determined by one-way ANOVA and Kruskal–Wallis test. $n = 3$ experiments. (D) Quantification of RNF169 SIRF foci in EdU-positive U2OS cells upon depletion of RNF20 or RAD51 and optionally treated with HU (4 mM, 2 h). At least 120 cells were analyzed per condition; ****$P < 0.0001$, ns nonsignificant, $P$ (siLUC UT vs. siRNF20_1 UT) = 0.4015, $P$ (siLUC UT vs. siRAD51 UT) < 0.9999; $P$ values determined by one-way ANOVA coupled with Kruskal–Wallis test. Bars show median values. $n = 3$ experiments. (E) RNF169 SIRF analysis as in D, upon optional treatment with the ATR inhibitor Ceralasertib. Ceralasertib was added to the cells in a final concentration of 1 μM 1 h prior to EdU labeling and was present both during EdU and HU treatment. At least 150 cells were analyzed per condition. ****$P < 0.0001$, $P$ values determined using one-way ANOVA coupled with Kruskal–Wallis test. $n = 3$ experiments. (F, G) Replication fork degradation assay in U2OS depleted of RNF169 with two independent siRNAs. siBRCA2 is used as a control of fork degradation. Representative DNA fibers are shown. At least 150 fibers were analyzed per condition. ****$P < 0.0001$, ***$P < 0.001$, $P$ (siLUC vs. siRNF169_1) = 0.0004. $P$ values calculated by one-way ANOVA followed by Kruskal–Wallis test. (H–J) U2OS-FIT cells were transfected with siLUC or siRNF169_2 and treated with doxycycline to induce expression of EV and RNF169 WT. After 72 h, cells were subjected to replication fork degradation assay as in F. Scatter plot of IdU/CldU tract length ratios for individual DNA fibers across different conditions (more than 120 fibers were analyzed per condition; **$P < 0.0010$, ****$P < 0.0001$, ns nonsignificant, $P$ (siLUC EV vs. siRNF169_2 WT) = 0.0674. $P$ values determined using one-way ANOVA, followed by Tukey's multiple comparisons test. Bars indicate median values. $n = 3$ experiments. (K, L) Replication fork degradation assay in U2OS cells depleted of RNF169 and RNF20. Representative DNA fibers and scatter plot representing IdU/CldU tract length ratios of individual DNA fibers across indicated conditions (more than 120 fibers analyzed per condition; ****$P < 0.0001$, ns nonsignificant, $P$ (siLUC vs. siRNF20_1) > 0.9999; $P$ values calculated using one-way ANOVA, followed by Kruskal–Wallis test). Bars represent median values. Source data are available online for this figure.

RNF169 (WT and MIU2*), by combining the micrococcal nuclease (MNase) digestion of nuclear extracts and FLAG immunoprecipitation. MNase preferentially cleaves the linker DNA between the individual nucleosomes, hence releasing intact nucleosomes from chromatin into soluble fraction. Nucleosomes, along with their interacting partners, were then immunopurified from the MNase-released soluble fraction using FLAG beads. With this technique, we observed the interaction of nucleosomes carrying FLAG-H2B with Myc-RNF169 in cells. Notably, this interaction was impaired upon the expression of the RNF169-MIU2* mutant, further confirming that the nucleosome binding of RNF169 largely depends on the MIU2 motif (Figs. 4D and EV4B). Moreover, H2B ubiquitination levels were almost completely abolished with K120R substitution even when RNF169 was overexpressed, indicating that RNF169 specifically leads to increased levels of H2B ubiquitination at K120 (Fig. 4E). While performing these experiments, we observed an increase in H2B ubiquitination in cells overexpressing RNF169, suggesting a contribution of RNF169 in promoting or stabilizing this histone mark. To better elucidate this point, we tested the contribution of the different domains of RNF169 on H2B ubiquitination. Interestingly, we found that the RING finger domain—and hence the potential ubiquitin ligase activity—is dispensable for this function, since the expression of a mutant with a defective RF domain (C68S; RF*) still results in increased levels of H2B ubiquitination (Fig. 4F). Conversely, the expression of RNF169 variants carrying the MIU2 mutation alone or in combination with the RF* mutation (RF*-MIU2*) failed to exert any effect on H2B ubiquitination. Noteworthy, this effect is specific for histone H2B, since RNF169 overexpression does not increase the levels of H2A ubiquitination, which are instead promoted by RNF168 (Fig. EV4C; (Mattiroli et al, 2012; Gatti et al, 2012)).

Overall, these results indicate that RNF169 is able to bind NCPs carrying H2BK120ub, both in vitro and in vivo, and that the overexpression of RNF169 specifically leads to increased levels of H2B ubiquitination that are dependent on its ubiquitin binding capability.

## RNF169 is recruited to stressed replication forks via H2BK120ub to limit fork resection

RNF169 has been reported to associate with nascent DNA/chromatin in different mass spectrometry-based studies, but its role in DNA replication was not elucidated (Dungrawala et al, 2015; Nakamura et al, 2021). Hence, we first sought to assess whether RNF169 can be detected at replication forks. Using SIRF analysis, we were able to show that endogenous RNF169 is present at newly synthesized DNA in unperturbed conditions, as its depletion largely diminished the SIRF signals, corroborating the specificity of the assay (Fig. EV5A,B). Induction of replication stress leads to the accumulation of RNF169 at replication forks, which was significantly reduced following RNF20 depletion ($P = 0.0004$; Figs. 5A,B and EV5C,D), supporting the notion that H2BK120ub contributes to RNF169 recruitment on nascent DNA in conditions of replication stress. Similar to the dependency of the physical interaction of RNF169 with NCP-H2BK120ub in vitro, we found that both MIU2 and LR domains are required for their proximal location in cells, as measured by PLA between Myc-RNF169 and H2BK120ub (Figs. 5C and EV5E). Moreover, in line with RAD51- and ATR-dependent accumulation of H2BK120ub at the replication forks, depletion of RAD51 or treatment with ATRi impaired replication stress-induced RNF169 accumulation at the stressed replication forks (Figs. 5D,E and EV5F,G).

To better characterize the role of RNF169 in DNA replication and replication stress response, we performed DNA fibers experiments upon siRNA-mediated RNF169 downregulation. Differently from RNF20, depletion of RNF169 did not affect the speed of replication fork progression under conditions of mild replication stress induced by CPT treatment (Fig. EV5H,I), indicating that RNF169 is dispensable for the effective DNA replication and fork slowdown in response to stress. However, loss of RNF169 led to increased degradation of nascent DNA upon fork stalling, mirroring the effects of BRCA2 loss, albeit seemingly to a lesser extent (Figs. 5F,G and EV5J). To further strengthen the contribution of RNF169 to fork protection, we generated U2OS

Flp-In TREx (FIT) cell lines expressing either the empty vector (EV) or the siRNA-resistant version of RNF169 (WT) upon doxycycline exposure (Fig. 5H) to be used in complementation experiments. While the stability of stalled forks was compromised by RNF169 depletion, the complementation with siRNA-resistant RNF169 completely restored fork protection (Fig. 5I,J). Lastly, we observed that the concomitant depletion of RNF20—which affects the formation of regressed arms—restored the stability of nascent DNA in RNF169-depleted cells, further supporting the notion that RNF169 contributes to the stability of stalled forks (Figs. 5K,L and EV5K).

# Discussion

The role of histone modifications in the regulation of fork dynamics and stability is rapidly emerging. Several studies demonstrated that histone marks can have different functions in different biological processes, acting both as a heritable and persistent epigenetic modification and as an inducible and transient histone mark. H3K4 and H3K27 methylation (referred to as H3K4me and H3K27me) are well-known epigenetic marks regulating gene transcription but can trigger unscheduled degradation of newly synthesized DNA in cells carrying BRCA1/2-deficiencies by promoting excessive chromatin loading of MRE11 and MUS81 nucleases (Chaudhuri et al, 2016; Rondinelli et al, 2017). More recently, the histone marks H3K9me1/2/3, critical markers for transcriptional silencing and heterochromatin formation, have been shown to be transiently deposited at sites of replication forks in response to replication stress (Gaggioli et al, 2023). Accumulation of histone H3K9 methylation at stressed replication forks drives heterochromatin formation at these sites and is essential for stressed fork stabilization and recovery (Gaggioli et al, 2023). Importantly, these histone marks accumulate in a checkpoint-dependent manner and are swiftly erased upon stress removal, thereby preventing interference of the replication stress response with the re-establishment of the pre-existing chromatin landscape (Gaggioli et al, 2023).

Our study provides additional evidence that different histone modifications, induced upon replication stress, can coexist and play distinct roles. Histone H2B ubiquitination is characterized by its function in promoting gene transcription and chromatin remodeling, but it was also reported to participate in the DDR and DNA repair. More recently, emerging evidence in yeast suggests that this histone mark is also important to regulate DNA replication and the replication stress response (Moyal et al, 2011; Lin et al, 2014; Northam and Trujillo, 2016; Trujillo and Osley, 2012; Hung et al, 2017). In this study, we combined biochemical and imaging techniques to investigate the role of H2BK120ub in the regulation of DNA replication fork dynamics and stability in human cells. We found that H2BK120ub is present at replication forks in unperturbed cells, where it accumulates in response to replication stress in a manner dependent on its main ubiquitin ligase RNF20. Notably, accumulation of H2BK120ub at sites of stressed DNA synthesis depends on ATR activity and is transient, as evidenced by the gradual decrease in the H2BK120ub levels upon stress release, thereby avoiding prolonged changes in the chromatin landscape that may ultimately influence genome expression.

Interfering with the deposition of H2BK120ub at stalled forks leads to the inability of replication forks to actively slowdown in conditions of replicative stress, as revealed by monitoring DNA replication at the single-molecule level by DNA fiber spreading. Such a defect is suggestive and has consistently been linked to defective fork remodeling into reversed forks. Defective fork reversal upon impaired H2BK120ub was further supported by monitoring the distance between the replisome and the newly synthesized DNA in conditions of HU-induced fork stalling, mirroring the defective fork remodeling reported upon RAD51 loss (Liu et al, 2023). Direct visualization of replication intermediates by EM confirms a reduced number of reversed forks upon RNF20 depletion, and fiber assays suggest that these defects reflect deregulated restart of reversed forks by the RECQ1 helicase. Consistent with these defects, depletion of RNF20 restores fork protection in BRCA2-deficient cells, presumably by reducing the entry points (i.e., regressed arms) for excessive nuclease-mediated degradation. Albeit detrimental in BRCA2-defective cells, replication fork reversal represents a key protective mechanism that allows forks to limit DNA synthesis upon DNA lesions or dNTPs exhaustion, and to resume DNA synthesis, avoiding chromosomal breakage (Neelsen and Lopes, 2015). The implication of RNF20/H2BK120ub in the formation and/or stabilization of reversed replication forks might help explain why H2BK120ub levels in cancer are frequently downregulated, due to either loss of RNF20 or upregulation of the eraser USP22 (Marsh et al, 2020). Lack of H2BK120ub could boost unrestrained DNA replication, undermining the fidelity of DNA synthesis and further promoting genomic instability (Berti et al, 2020). These results extend to human cancer cells previous findings on the role of H2Bub (H2BK123ub) and Bre1 (RNF20 homolog) in yeast, where H2Bub was reported to facilitate replication fork stalling during replication stress (Hung et al, 2017; Lin et al, 2014; Northam and Trujillo, 2016). The evidence that the key fork reversal factor RAD51 is required for efficient RNF20/H2BK120ub accumulation at forks—while H2BK120ub itself is needed for active fork slowing and remodeling—suggests that this histone modification plays a key role promoting the accumulation of reversed forks downstream of RAD51, presumably by stabilizing reversed forks and preventing their untimely restart by RECQ1. However, the detailed molecular mechanisms underlying the crosstalk of RNF20/H2BK120ub with other reversal and/or restart factors deserve further investigation.

In line with our evidence on checkpoint activation upon fork stalling by HU, RNF20 and H2BK120ub were recently reported to potentiate ATM/ATR activation upon formaldehyde treatment (Mishra et al, 2024). Furthermore, loss of H2Bub in yeast was also reported to impair S phase checkpoint activation (Lin et al, 2014). Most recently, a screening conducted in yeast identified Lge1 as a key factor controlling replication initiation and fork stability, by promoting H2B ubiquitination by Bre1, the yeast homolog of RNF20 (van der Horst et al, 2025). Overall, this recent evidence in yeast and human cells suggests that the role of RNF20/H2BK120ub in sustaining fork remodeling/stability and checkpoint activation in S phase is relevant upon different conditions of replication stress and is evolutionary conserved.

While searching for readers of H2BK120ub in this context, we found that the DDR protein RNF169 physically interacts with this

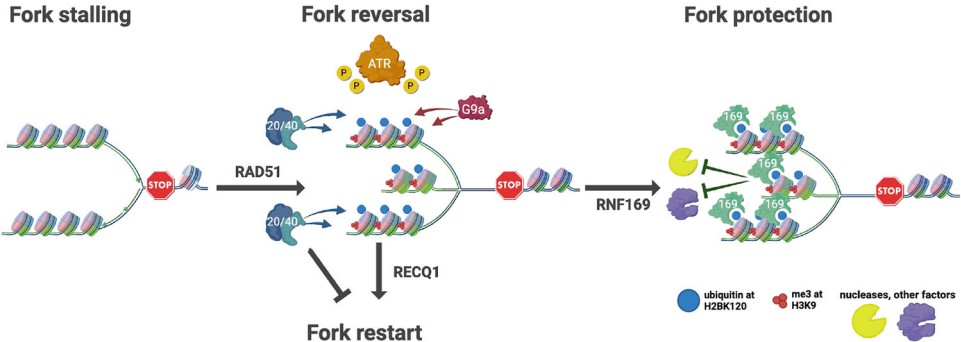

**Figure 6. Model.**

Graphical illustration of the proposed role of the RNF20/H2BK120ub/RNF169 axis in DNA replication fork dynamics. Replication fork stalling (indicated by the stop sign) induces the deposition of H2BK120ub by the ubiquitin ligases RNF20/40, which contributes to the formation/stabilization of regressed arms following the reversal of stalled forks. H2BK120ub coexists with the heterochromatin histone mark H3K9me3 (promoted by the histone methyltransferase G9a) at reversed forks and both are dependent on ATR kinase activity (indicated by the yellow circle with a P). RNF169 accumulates at stressed DNA replication forks by direct binding to H2BK120ub within the nucleosomes. H2BK120ub/RNF169 synergize to prevent the untimely restart by RECQ1 and the deregulated nucleolytic degradation of nascent DNA. Created in https://BioRender.com.

histone mark and localizes at replication forks in conditions of stress, in a RNF20- and ATR-dependent manner. This result complements previous studies reporting the recruitment of RNF169 into chromatin following the induction of DNA double-strand breaks, by binding the H2AK15ub histone mark—promoted by the ubiquitin ligase RNF168—through its ubiquitin binding domain MIU (Panier et al, 2012; Kitevski-Leblanc et al, 2017; Poulsen et al, 2012; Mattiroli et al, 2012; Gatti et al, 2012). Here, RNF169 competes with, and displaces, 53BP1 from damaged chromatin by exhibiting a higher binding affinity to H2AK15ub (Kitevski-Leblanc et al, 2017; Poulsen et al, 2012; Hu et al, 2017). On the other hand, in the replication context, RNF169 contributes to the stability of stalled forks, likely by binding H2BK120ub that accumulates at nascent DNA in replication stress conditions. Whether this RNF169 function is exerted by facilitating the binding of protective factors or rather competing with factors that promote fork degradation is still unknown. Intriguingly, these two histone modifications—i.e., H2AK15ub and H2BK120ub—are in close proximity in the nucleosome structure, suggesting that RNF169 might be positioned on ubiquitinated chromatin—as a consequence of DNA damage or replicative stress—in a similar mode. Moreover, since we have previously uncovered a role for RNF168, H2AK15ub and other canonical DDR factors at the sites of ongoing DNA synthesis to ensure effective DNA replication in unperturbed conditions (Schmid et al, 2018), it would be interesting to assess the contribution of H2AK15ub to the recruitment of RNF169 at stalled forks, and the possible crosstalk between these two histone modifications. Future studies are needed to better elucidate the role of these canonical DDR ubiquitin ligases in the replication stress response and dynamics.

Although the outcomes of RNF20 and RNF169 depletion on DNA replication is different, this does not exclude or contradict their involvement in distinct steps of the same replication stress response pathway. In fact, factors promoting fork reversal typically do not impact fork protection, while those safeguarding reversed

forks from degradation often do not affect fork slowing or remodeling (e.g., Liu et al, 2020; Mijic et al, 2017; Berti et al, 2020). A notable case is the differential effect of RAD51 and BRCA2 depletion: while BRCA2 loss leads to fork degradation, RAD51 depletion does not, and can even suppress the degradation seen in BRCA2-deficient cells. This reflects RAD51's BRCA2-independent role in reversed fork formation, which creates substrates for nucleolytic degradation in the absence of BRCA2-mediated protection. Thus, RAD51 and BRCA2 act sequentially in reversed fork formation and protection, yet exhibit distinct molecular phenotypes. We propose a similar model for RNF20 and RNF169. RNF20 and H2BK120ub promote/stabilize fork reversal, with their loss leading to unrestrained fork progression via RECQ1, rather than fork degradation. In contrast, RNF169 binds H2BK120ub and protects nascent DNA at reversed forks, analogous to BRCA2's protective role. Thus, RNF169 depletion impairs fork protection without affecting fork slowing.

Overall, our data support a model (Fig. 6) where de novo ubiquitination of histone H2B at replication forks upon stress, induced by RNF20/RNF40 and dependent on ATR and RAD51, is required for efficient fork reversal and to stabilize forks in their reversed state. While exerting this function, H2BK120ub coexists with the histone mark H3K9me3, promoted by the histone methyltransferase EHMT2/G9a and shown to induce heterochromatin assembly at stressed replication forks (Gaggioli et al, 2023). Here, the accumulation of H2BK120ub in turn serves as a docking platform for the recruitment of RNF169 to the reversed replication forks, which limits nucleolytic degradation of the nascent DNA. Hence, our evidence identifies a novel RNF20/H2BK120ub/RNF169 axis in DNA replication stress response, further highlighting the significance of histone modifications—and of its writers and readers – in replication fork dynamics and stability, and further supporting the promise of targeting histone modifiers to potentiate therapeutic options interfering with the replication process of cancer cells.

# Methods

### Reagents and tools table

| Reagent/resource | Reference or source | Identifier or catalog number |
|---|---|---|
| **Experimental models** | | |
| HEK293T cells (*H. sapiens*) | ATCC | CRL-3216 |
| U2OS cells (*H. sapiens*) | ATCC | HTB-96 |
| **Recombinant DNA** | | |
| pcDNA3.1 | Thermo Fisher Scientific | V79020 |
| pcDNA3.1 FLAG-H2B WT | This manuscript | N/A |
| pcDNA3.1 FLAG-H2B K120R | This manuscript | N/A |
| pcDNA3.1 FLAG-H2A | Pinato et al, 2009 | N/A |
| pcDNA3.1 MYC-RNF169 | This manuscript | N/A |
| pcDNA3.1 RNF168 | Gatti et al, 2012 | N/A |
| pOG44 | Thermo Fisher Scientific | V600520 |
| pcDNA5 FRT/TO | Thermo Fisher Scientific | V652020 |
| pcDNA5 FRT/TO RNF169 WT | This manuscript | N/A |
| pMAL6-c6T | New England Biolabs | N0378S |
| pMAL6 WT (RNF169 662–708) | Panier et al, 2012 | N/A |
| pMAL6 MIU* (RNF169 662–708; A673G) | This manuscript | N/A |
| pMAL6 LR* (RNF169 662–708; L699A; R700A) | This manuscript | N/A |
| **Antibodies** | | |
| Mouse anti-Actin | Sigma-Aldrich | RRID:AB_476744 |
| Mouse anti-Biotin | Jackson ImmunoResearch | RRID:AB_2339006 |
| Rabbit anti-Biotin | Cell Signaling | 39681 |
| Mouse anti-BRCA2 | Millipore | RRID:AB_2067762 |
| Rabbit anti-FLAG | Sigma-Aldrich | RRID:AB_439687 |
| Mouse anti-FLAG M2 | Sigma-Aldrich | RRID:AB_259529 |
| Mouse anti-c-Myc (9E10) | Santa Cruz Biotechnology | RRID:AB_627268 |
| Mouse anti-GAPDH | Merck | RRID:AB_2107445 |
| Rabbit anti-Histone H2A | Millipore | RRID:AB_11212920 |
| Rabbit anti-Histone H3 | Abcam | RRID:AB_302613 |
| Rabbit anti-Histone H2B | Abcam | RRID:AB_302612 |
| Rabbit anti-H2BK120ub | Cell Signaling | 5546S |
| Mouse anti-H3K9me3 | Abcam | ab317790 |
| Rabbit anti-Lamin A | Sigma-Aldrich | RRID:AB_532254 |
| Rabbit anti-MCM7 | Cell Signaling | RRID:AB_2142705 |
| Rabbit anti-pCHK1 (S345) | Cell Signaling | 2348S |
| Mouse anti-CHK1 | Santa Cruz Biotechnology | RRID:AB_627257 |
| Rabbit anti-pRPA32 (S4/S8) | Bethyl | RRID:AB_210547 |
| Mouse anti-RPA | Millipore | RRID:AB_11205561 |
| Mouse anti-TFIIH | Boster Biol. Technology | M03103 |
| Mouse anti-PCNA | Santa Cruz Biotechnology | RRID:AB_628110) |
| Rabbit anti-RNF20 | Abcam | RRID:AB_873630 |
| Rabbit anti-RNF169 | Abcam | ab188237 |
| Rabbit anti-RAD51 | Bio Academia | RRID:AB_2177110 |
| Rabbit Anti-RECQ1 | Bethyl | RRID:AB_420916 |
| Mouse anti-ubiquitin | Santa Cruz Biotechnology | RRID:AB_628423 |
| Anti-mouse HRP | Axon Lab | RRID:AB_3661795 |
| Anti-rabbit HRP | Axon Lab | RRID:AB_3661796 |
| Anti-rat HRP | Invitrogen | RRID:AB_10561556 |

| Reagent/resource | Reference or source | Identifier or catalog number |
|---|---|---|
| Mouse anti-IdU | BD Biosciences | RRID:AB_400326 |
| Rat anti-BrdU | Abcam | RRID:AB_305426 |
| Anti-mouse Alexa 488 | Thermo Fisher Scientific | RRID:AB_2633275 |
| Anti-rabbit Alexa 555 | Thermo Fisher Scientific | RRID:AB_2535849 |
| Anti-rat Cy3 | Jackson ImmunoResearch | RRID:AB_2340669 |
| **Oligonucleotides and other sequence-based reagents** | | |
| siLUC 5'-CUUACGCUGAGUACUUCGAdTdT-3' | Mycrosynth AG | N/A |
| siRNF20_1 5'-CAGGAAGCUGGAGACCACAAAGAAA-3' | Mycrosynth AG | N/A |
| siRNF20_2 5'-CCGTGCTGTAACATGCGTAAA-3' | Mycrosynth AG | N/A |
| siRNF40 5'-CAACGAGTCTCTGCAAGTGTT-3' | Mycrosynth AG | N/A |
| siRNF169-1 5'-GGUCCUCUCUGAGUAUACU-3' | Mycrosynth AG | N/A |
| siRNF169-2 5'-UAAUAAGGUCUGUAAAUGUGCUCUG-3' | Mycrosynth AG | N/A |
| siBRCA2 5'-UUGACUGAGGCUUGCUCAGUUdTdT-3' | Mycrosynth AG | N/A |
| siRECQ1 5'-UUACCAGUUACCAGCAUUAUU-3' | Mycrosynth AG | N/A |
| RNF169 RF* (C78S) Forward primer: 5'-GAGGAATCGGGCAGCGCCGGGTGCCTGGAG-3' Reverse primer 5'-CTCCAGGCACCCGGCGCTGCCCGATTCCTC-3 | Mycrosynth AG | N/A |
| RNF169 MIU* (A673G) Forward primer: 5'-GAAGACCGACAGTTGGGTCTGCAGTTGCAG-3' Reverse primer: 5'-CTGCAACTGCAGACCCAACTGTCGGTCTTC-3' | Mycrosynth AG | N/A |
| RNF169 LR* (L699A; R700A) Forward primer: 5'-GGATCAGTATCTCGCAGCGTCCAGCAACATGG-3' Reverse primer: 5'-CCATGTTGCTGGACGCTGCGAGATACTGATCC-3' | Mycrosynth AG | N/A |
| RNF20 RT-qPCR primers Forward primer: 5'-GAACAGCGACTCAACCGACA-3' Reverse primer: 5'-GGAATTCACCCGTTCTAGGACTT-3' | Integrated DNA Technologies | N/A |
| **Chemicals, Enzymes and other reagents** | | |
| Camptothecin (CPT) | Sigma-Aldrich | 7689-03-04 |
| Hydroxyurea (HU) | Sigma-Aldrich | H8627 |
| ATR inhibitor (Ceralasertib) | Selleckchem | S7693 |
| Anti-FLAG M2 magnetic beads | Sigma-Aldrich | M8823 |
| Amylose Resin | New England Biolabs | E8021 |
| Duolink In Situ PLA Probe anti-Rabbit MINUS | Sigma-Aldrich | DUO92005 |
| Duolink In Situ PLA Probe anti-Mouse PLUS | Sigma-Aldrich | DUO92001 |
| Duolink In Situ Detection Reagents FarRed | Sigma-Aldrich | DUO92013 |
| NCPs unmodified | https://doi.org/10.1016/j.molcel.2020.09.017 | N/A |
| NCPs-H2BK120ub | EpiCypher | 16-0396 |
| Oligofectamine transfection reagent | Thermo Fisher Scientific | L3000008 |
| PfuTurbo DNA polymerase | Agilent | 600250 |
| Micrococcal nuclease, MNase | Worthington | 27735 |
| CldU | Sigma-Aldrich | C6891 |
| IdU | Sigma-Aldrich | I7125 |
| WesternBright ECL HRP substrate | Advansta | K-12045-D50 |
| RNase A | Sigma-Aldrich | R5503 |
| FuGENE® 6 Transfection Reagent | Promega | Ref. No. E2691 |

| Reagent/resource | Reference or source | Identifier or catalog number |
|---|---|---|
| Flp-In T-REx Core Kit | Thermo Fisher Scientific | K650001 |
| **Software** | | |
| Olympus ScanR image analysis software (version 3.0.1) | Olympus Life Science | www.olympus-lifescience.com |
| Spotfire data visualization software (version 7.0.1) | TIBCO Software Inc. | www.spotfire.com |
| GraphPad Prism 10 for MAC OS | GraphPad Software | www.graphpad.com |
| FlowJo 10.6.1 | FlowJo Software | www.flowjo.com |

## Cell culture and treatments

All cell lines were maintained in standard DMEM (Dulbecco's Modified Eagle Medium) supplemented with 10% fetal bovine serum (FCS, tetracycline off) and penicillin (50 U/ml) and streptomycin (0.05 µg/ml) at 37 °C with 5% $CO_2$. Cells were subcultured every 2 to 3 days and maintained at below 80% confluency. Cells were treated with either 100 nM CPT for 1 h or 4 mM HU for 2 h. ATR inhibitor was given at a final concentration of 1 µM for 1 h prior, as well as during EdU labeling (10 min) and HU treatment (2 h).

For knockdown assays, siRNA molecules were transfected into low-confluent U2OS cells (around 30–40%), day after initial cell seeding in OptiMEM medium using Oligofectamine transfection reagent according to the manufacturer's instructions. Four hours post transfection cells were supplemented with 5% FCS. Twenty-four hours post transfection cells were supplemented with complete medium containing 10% FCS. siBRCA2 (40 nM, 48 h), siRAD51 (40 nM, 48 h), siRNF20 (25 nM, 72 h; double transfection—second round of transfection 24 h post the first round), siRNF169-1/2 (40 nM, 72 h; double transfection—second round of transfection 24 h post the first round).

For overexpression assays, indicated plasmids were transiently transfected into HEK293T cells using the $CaPO_4$ DNA precipitation transfection protocol. Cells were harvested and proteins extracted 48 h post transfection.

## Western blot

Unless otherwise stated, proteins were extracted in 1% SDS, 50 mM Tris, pH 8, in $ddH_2O$ and quantified using the BCA assay. Protein extracts were run on SDS-PAGE and transferred to a nitrocellulose membrane and blocked in 5% BSA in TBST. Incubation with primary antibodies was done overnight at 4 °C. Following dilutions were used: anti-H2BK120ub (1:1000), anti-RNF20 (1:1000), anti-RNF169 (1:2000), anti-H3 (1:1000), anti-H2B (1:2500), anti-H2A (1:5000), anti-Lamin A (1:1000), anti-RAD51 (1:1000), anti-Tubulin (1:8000), anti-BRCA2 (1:1000), anti-TFIIH (1:1000), anti-ubiquitin (1:1000), anti-pCHK1 (1:1000), anti-CHK1 (1:1000), anti-pRPA (1:1000), anti-RPA (1:1000), anti-Actin (1:2000), Rb anti-FLAG (1:1000), Mo anti-FLAG (1:1000), anti-MYC (1:1000). Following washing in TBST, blots were incubated with corresponding HRP-conjugated secondary antibodies in 1:1000 dilution and developed using ECL HRP substrate.

## Generation of U2OS-FIT cells expressing RNF169

U2OS Flp-In TREx (FIT) cells carrying inducible siRNA-resistant RNF169 WT were generated by Flp recombinase–mediated integration (Flp-In T-REx Core Kit). Briefly, cells were transfected with pcDNA5 empty vector (EV) or pcDNA5 RNF169 WT using FuGENE Transfection Reagent together with pOG44 vector (9:1 pOG44/pcDNA5) and 48 h after transfection selected with Hygromycin B 1(00 µg/ml) for 2 weeks. The pools of stable EV and RNF169 WT expressing FIT cells were expanded. RNF169 expression levels were tested by immunofluorescence and immunoblotting after 24, 48, and 72 h of 1 µg/ml doxycycline treatment.

## Site-directed PCR mutagenesis

RNF169 mutants were generated by site-directed PCR mutagenesis using high-fidelity PfU Turbo DNA polymerase (Agilent), according to the manufacturer's instructions, and primers described in the reagents and tools section.

## Histone acidic extraction

During the entire procedure, samples were kept at 4 °C and ice-cold buffers were used. HEK293T cells from 90% confluent 10 cm Petri dishes were harvested in PBS supplemented with protease inhibitor (PI) cocktail, 10 mM NEM, and 1 mM PMSF. Cells were washed twice in PBS. Non-histone proteins were extracted in three rounds of 10 min incubations in buffer A (5% PCA (perchloric acid), PI cocktail, 10 mM NEM, 1 mM PMSF in $ddH_2O$), followed by 10 min centrifugation at 4 °C at 13,000 RCF. Core histones were extracted in three rounds of incubations in Buffer B (0.4 N HCl, 10 mM NEM, PI cocktail in $ddH_2O$) followed by 10 min centrifugation at 4 °C at 13000 RCF. Buffer B from all three rounds of core histone extractions was collected, and histones were precipitated by adding TCA (trichloroacetic acid) to a final concentration of 25% in Buffer B, followed by on-ice incubation for 30 min and 20 min centrifugation at 4 °C at 13000 RCF. Histone pellets were washed in ice-cold acetone with 0.006% HCl and then in 100% acetone. After centrifugation (20 min, 13000 RCF, 4 °C), pellets were air-dried before being resuspended in 50 mM Tris pH 7.5 in $ddH_2O$. Protein quantification was performed using the Bradford assay.

## Nucleosome immunoprecipitation

Nucleosome immunoprecipitation was performed as previously described (Khan et al, 2020). In brief, HEK293T cells from 90% confluent 15 cm petri dishes were harvested and collected in 15 ml Falcon tubes and counted using an automated cell counter (Thermo Fisher Scientific). Cells were pelleted by centrifugation for 5 min at 4 °C at 500 RCF. Cells were washed twice in PBS in 1.5 ml tubes and centrifuged for 5 min at 4 °C at 500 RCF. Cytosolic fractions were extracted by 5 min incubation in buffer A (100 mM HEPES, pH 7.35, 10 mM KCl, 1.5 mM $MgCl_2$, 340 mM Sucrose, 10% glycerol, 1 mM DTT, 200 µM PMSF, 10 mM NEM, PI cocktail, 0.2% Triton X-100 in $ddH_2O$). Nuclei were pelleted by 5 min centrifugation at 4 °C at 1500 RCF. Nuclei were resuspended in cutting buffer (10 mM Tris, pH 7.5, 15 mM NaCl, 60 mM KCl, and 2 mM $CaCl_2$) and 1U of Micrococcal nuclease (Worthington) was added per 1 million cells. MNase digestion was performed for 30 min at 37 °C at 300 RPM and stopped by adding EGTA (ethylene glycol-bis(beta-aminoethyl-ether)-*N*,*N*,*N'*,*N'*-tetraacetic acid) to a final concentration of 20 mM. Nuclei were centrifuged 5 min at 4 °C at 3000 RCF, and supernatants containing MNase-released nucleosomes were

collected (S1 fraction). Nuclei were further washed in TE buffer (10 mM Tris pH8, 1 mM EDTA, 200 μM PMSF, 10 nM NEM, PI cocktail in ddH$_2$O) and centrifuged for 5 min at 4 °C at 3000 RCF and supernatants containing residual MNase-released nucleosome were collected (S2 fractions). Corresponding S1 and S2 fractions were combined and used for the later anti-FLAG immunoprecipitation. Samples were quantified by Bradford assay, and equal amounts of proteins were incubated with anti-FLAG resin for 2 h at 4 °C while rotating. Resin was washed five times with HNTG buffer (20 mM Hepes pH 7.5, 10% Glycerol, 150 mM NaCl, 0.1% Triton X-100 in ddH$_2$O). Samples were eluted by boiling at 95 °C for 10 min in 2X gel loading buffer (100 mM Tris, pH 6.8, 20% Glycerol, 3.2% SDS, 200 mM DTT, and 0.02% bromophenol blue in ddH$_2$O).

## Isolation of proteins on nascent DNA (iPOND)

iPOND protocol was performed as described in (Sirbu et al, 2013). Briefly, 90% confluent HEK293T cells in three 15 cm Petri dishes per condition were incubated in cell medium supplemented with 10 μM EdU for 10 min at 37 °C. For the HU-treated samples, after EdU labeling, cells were carefully washed with warm PBS and incubated in the presence of 4 mM HU for 2 h. Cells were fixed and proteins crosslinked with the DNA by adding FA (formaldehyde) to a final concentration of 1% in cell medium and incubated for 12 min at RT. Glycine was added to a final concentration of 0.1 M to stop the DNA-protein crosslinking and incubated for 5 min at RT. Cells were harvested by scraping and collected in 50 ml Falcon tubes. Cells were pelleted by centrifugation for 10 min at 4 °C at 2000 RCF. After three rounds of washing in PBS, cells were permeabilized in 0.25% Triton X-100 in PBS for 30 min at RT. Cells were washed in ice-cold 0.5% BSA in PBS, followed by a washing round in PBS. Cells were resuspended in click-it reaction buffer (10 mM sodium ascorbate, 2 mM CuSO$_4$ in the presence of 10 μM biotin azide in PBS). For the control samples (no click-it), biotin azide was omitted. Click-it reaction was performed for 2 h while rotating in the dark. Cells were washed twice with ice-cold PBS. Cells were resuspended in lysis buffer (50 mM Tris, pH 8, 1% SDS in ddH$_2$O), sonicated and centrifuged for 10 min at RT at 16,000 RCF. Supernatants were transferred to fresh tubes, diluted by adding an equal volume of PBS to dilute SDS concentration to 0.5% and incubated with streptavidin resin ON at 4 °C in the dark. Bead slurry was centrifuged for 2 min at 1600 RCF at RT and washed, once in 25 mM Tris, pH 8, 0.5% SDS in PBS, twice in 1 M NaCl and twice in PBS. Proteins were decrosslinked from the streptavidin beads by boiling for 30 min at 95 °C in 2x SDS-loading buffer.

## MBP constructs production and purification

pMAL6 bacterial shuttle vectors encoding MBP constructs only (EV) or fused with C-terminal RNF169 domain (residues 662–708) in WT, A673G (MIU2*), L699A R700A (LR*), or double mutant (MIU2*-LR*) variants were transformed in *BL21 E.coli* strain using the heat shock method. Transformed bacteria were grown in liquid LB medium at 37 °C at 180 RPM. When bacterial cultures reached an OD (optical density) of 0.3, IPTG was added to a final concentration of 0.5 mM to induce recombinant protein expression. Bacteria were grown in the presence of IPTG for 4 h under the same conditions. Bacteria were pelleted by centrifugation at RT.

Bacteria were resuspended in ice-cold lysis buffer (20 mM Tris, pH 7.4, 200 mM NaCl, 1 mM EDTA, 1 mM DTT, 200 μM PMSF, bacterial PI cocktail) and lysed by three rounds of sonication (5 min each round, high intensity, 15 s on/15 s off) on ice. Protein extracts were cleansed by centrifugation for 30 min at 4 °C at 20,000 RCF. Protein extracts were incubated with amylose resin for 4 h at 4 °C while rotating. Resin was washed in lysis buffer 5x followed by 3 min centrifugation at 4 °C at 1500 RCF. The efficacy of the protein purification and the protein quantification was performed by applying SDS-PAGE and subsequent Coomassie Blue protein staining.

## MBP pulldown assays

MBP constructs (1 μg) on amylose resin were incubated with 2.5 μg unmodified NCPs (nucleosome core particles) or NCPs encompassing H2BK120ub in 100 μl of pulldown buffer (20 mM Tris, pH 7.4, 200 mM NaCl, 1 mM EDTA, 1 mM DTT, 200 μM PMSF, bacterial PI cocktail) for 2 h at 4 °C at 500 RPM. Samples were centrifuged for 3 min at 4 °C at 1500 RCF and washed 5x in the pulldown buffer prior to 10 min boiling at 95 °C in 2x SDS-loading buffer and consequential Western blot analysis.

## Flow cytometry analysis of EdU incorporation

U2OS cells (80% confluent) in 6 cm petri dishes were incubated in the presence of EdU (10 μM) for 30 min at 37 °C. Cells were washed with PBS and collected in medium after 10 min of incubation in trypsin at 37 °C. Cells were pelleted by centrifugation at 500 RCF for 5 min at 4 °C, washed with PBS, and 300,000 cells were transferred to a fresh tube. Cells were fixed in 4% formaldehyde in PBS for 15 min at RT. Cells were washed three times with 1% BSA in PBS and permeabilized in saponin buffer (0.5x saponin in 1% BSA in PBS) for 5 min at RT. After pelleting, cells were incubated in Click-it reaction buffer (100 mM Tris, pH 8, 2 mg/ml sodium ascorbate, 4 mM CuSO$_4$, and 10 μM pycolyl azide) for 1 h at RT in the dark. Cells were afterwards washed twice with 1% BSA in PBS and incubated with RNAse (0.1 mg/ml) and DAPI (1 μg/ml) for 20 min at RT in the dark. Samples were analyzed using an Attune NxT Acoustic Focusing Cytometer (Invitrogen). Flow cytometry data were analyzed using FlowJo software (version 10.6.1). "n" indicates biological replicates.

## Proximity ligation assay (PLA) and SIRF

For PLA analysis, cells were transfected with Myc-RNF169 constructs using FuGENE transfection reagent in OptiMEM according to the manufacturer's instructions, 24 h post seeding on 13-mm glass coverslips. Four hours after transfection, cells were supplemented with 10% FBS and 16 h post transfection treated with 4 mM HU for 2 h, before fixation in ice-cold MetOH. Following two rounds of washing with PBS, permeabilization in 0.3% Triton X-100 and blocking in 3% BSA in PBS, cells were incubated ON at 4 °C with mouse anti-MYC and rabbit anti-H2Bub primary antibodies in 1:400 and 1:1600 dilutions, respectively. The rest of the PLA protocol is described below.

SIRF assay was performed as described in (Roy et al, 2018). In brief, 16 to 24 h prior to cell fixation, cells were seeded on coverslips. Cells were grown in the presence of EdU (25 μM) to

allow labeling of newly synthesized DNA. To obtain comparable levels of EdU incorporation across different conditions, the labeling time slightly varied following different treatments. Untreated cells were incubated in the presence of EdU for 10 min at 37 °C. For HU-treated samples, following 8 min EdU labeling, cells were incubated in the presence of 4 mM HU for 2 h at 37 °C. For CPT-treated samples, 100 nM CPT was added to the cell medium 47 min prior to 13 min EdU labeling, and CPT was present in the cell medium during EdU labeling. Cells were washed 3x with warm PBS and fixed in ice-cold methanol for 10 min at −20 °C. Cells were washed twice with PBS and permeabilized with 0.3% Triton X-100 in PBS for 5 min at RT. Cells were washed twice per 15 min with 3% BSA in PBS. Biotin was conjugated to the EdU by click-it reaction that was performed in 100 mM Tris pH 8, 2 µg/ml sodium ascorbate, 4 mM $CuSO_4$, and 10 µM Biotin azide for 2 h in a dark humidity chamber at 37 °C. After click-it reaction, cells were washed two times per 15 min in 3% BSA in PBS in dark while gently agitating, followed by 1 h of blocking under the same conditions. Cells were incubated with corresponding primary antibodies overnight at 4 °C in a dark, humid chamber. Following dilutions for primary antibodies were used: Rb anti-Biotin (1:500), Mo anti-Biotin (1:200), Rb anti-H2BK120ub (1:1600), Rb anti-RNF169 (1:2000), Rb anti-MCM7 (1:100), Mo anti-PCNA (1:500). Cells were washed with PBS prior to incubation with PLA probes (Sigma-Aldrich) according to the manufacturer's instruction for 1 h at 37 °C in a dark humidity chamber. Cells were washed twice for 5 min with Buffer A (0.01 M Tris, 0.15 M NaCl and 0.05% Tween-20, pH 7.4) at RT in the dark while gently agitating. Cells were then incubated in the PLA ligation buffer (Sigma-Aldrich) for 30 min at 37 °C in the dark humidity chamber, followed by two rounds of Buffer A washes and incubation in the PLA FarRed amplification buffer (Sigma-Aldrich) according to the manufacturer's instructions for 100 min at 37 °C in the dark humidity chamber. Cells were washed twice with Buffer B (0.2 M Tris and 0.1 M NaCl, pH 7.5), followed by two rounds of PBS washes and incubated with secondary antibodies (in 1:100 dilution) for 30 min at RT in the dark humidity chamber. DAPI staining was performed for 15 min at RT in the dark. Cells were washed twice with PBS, once with $ddH_2O$ and mounted on microscope plates using ProLong gold antifade mounting medium (Thermo Fisher Scientific). Cells were analyzed using a fluorescent wide-field Leica DM6 B microscope, a DMC2900 camera using a HCX PL APO 63x immersion oil objective. At least 120 EdU-positive cells were analyzed per condition under non-saturating conditions at a single z-position. All images were taken on the same day, and identical settings were applied to all samples within one experiment. "n" indicates biological replicates.

## Chromatin fiber analysis (ChromStretch)

Chromatin fiber analysis was performed using the ChromStretch technology as described previously (Gaggioli et al, 2023). In brief, cells were pulse-labeled with 10 µM EdU for 20 min, followed by 4 mM hydroxyurea treatment for 1 h. Nuclei were pre-extracted in buffer (10 mM HEPES, 10 mM KCl, 1.5 mM MgCl2, 0.34 M sucrose, 1 mM DTT, 10% glycerol, 0.1% Triton X-100, and protease inhibitor (cOmplete mini EDTA-free PIC; Roche)) for 5 min on ice and chromatin was isolated in hypotonic conditions (3 mM EDTA, 0.1 mM EGTA, 1 mM DTT, and protease inhibitor) before

spreading on a SuperFrost Plus glass microscope slide (Epredia). Slides were moved to a lysis chamber containing lysis buffer (12.5 mM MES hydrate, 12.5 mM MES sodium salt, 0.1 mM EDTA, 0.1 mM EGTA, 1 mM DTT, and 2% Triton X-100) for 20 min, after which chromatin fibers were stretched by a continuous laminar outflow of the lysis buffer in a device designed and built in the lab. The stretched chromatin fibers were fixed in 4% formaldehyde in PBS for 15 min. Replication tracks were visualized via click-reaction of Alexa Fluor 594-azide (Jena Bioscience) to EdU, after which slides were blocked in 5% BSA in PBS for 1 h. Slides were stained using the indicated antibodies, including mouse anti-H3K9me3 (1:500), rabbit anti-H2BK120ub (1:500) and counterstained with rabbit anti-H3 (1:1000). Slides were imaged using a Leica Stellaris 5 confocal microscope at 63x (HC PL APO CS2, NA 1.4). Quantification of H2BK120ub and H3K9me3 intensity at EdU sites was performed using ImageJ. "n" indicates biological replicates.

## DNA fibers assay and fork degradation assay

U2OS cells (80% confluent) were sequentially incubated in the presence of nucleotide analogs (33 µM CldU and 339 µM IdU) for exactly 30 min each at 37 °C. Between and after each labeling, cells were washed 3x with warm PBS. For the DNA replication analysis under conditions of mild replication stress, during the IdU labeling, 100 nM CPT was present in the medium. For the fork degradation assay, after the second (IdU) labeling, cells were incubated in the presence of 4 mM HU for 4 h at 37 °C. Cells were collected for the DNA fibers analysis by trypsinization. Cells were washed with PBS and counted. Cells were diluted in PBS to a concentration of 350,000 live cells per 1 ml. Five microliters of cell dilutions were added to the 7.5 µl lysis buffer (100 mM Tris, pH 7.5, 1% SDS, and 50 mM EDTA) drops on microscope slides, mixed and left to incubate at RT for 9 min. Afterwards, microscope slides were tilted to a 45° angle to let the DNA fibers stretch across the slide. Slides were air-dried and fixed in acetic acid:methanol (in 3:1 ratio) solution overnight at 4 °C. Slides were rehydrated by washing in PBS. DNA was denatured in 2.5 M HCl, rehydrated in PBS, and blocked for 60 min in 2% BSA, 0.1% Tween in PBS. Slides were incubated with 1:500 diluted anti-BrdU antibody recognizing CldU and 1:100 diluted anti-IdU antibody for 2 h at RT. Slides were washed and incubated with secondary antibodies, 1:150 diluted anti-mouse Alexa Fluor 488 and 1:300 diluted anti-rat Cy3 for 1 h in a dark, humidity chamber. Coverslips were mounted onto the microscope slides using ProLong gold antifade mounting medium (Thermo Fisher). DNA fibers images were acquired on a fluorescence wide-field Leica DM6 B microscope, a DMC2900 camera using a HCX PL APO 63x immersion oil objective. "n" indicates biological replicates.

## Electron microscopy analysis

Seventy-two hours after transfection with siLUC and siRNF20_1, U2OS cells were treated with 4 mM HU for 2 h. Following treatment, cells were harvested, resuspended in ice-cold phosphate-buffered saline (PBS), and crosslinked with 4,5′,8-trimethylpsoralen (final concentration: 10 µg/mL). Crosslinked samples were subsequently irradiated with 365 nm monochromatic ultraviolet light using a UV Stratalinker 1800 (Agilent Technologies). For the

isolation of genomic DNA, cells were lysed in buffer containing 1.28 M sucrose, 40 mM Tris-HCl (pH 7.5), 20 mM MgCl$_2$, and 4% Triton X-100, followed by protein digestion in buffer containing 800 mM guanidine-HCl, 30 mM Tris-HCl (pH 8.0), 30 mM EDTA (pH 8.0), 5% Tween-20, and 0.5% Triton X-100, supplemented with proteinase K (1 mg/mL), at 50 °C for 2 h. DNA was purified via chloroform/isoamyl alcohol extraction (24:1) and precipitated with an equal volume of isopropanol. The pellet was washed with 70% ethanol and resuspended in 200 μL of TE buffer. For restriction digestion, 6 μg of purified genomic DNA was incubated with 120 U of PvuII-HF (New England Biolabs) at 37 °C for 5 h, with RNase A (250 μg/mL; Sigma-Aldrich, R5503) added during the final 2 h of incubation. Digested DNA was purified using a Silica Bead Gel Extraction kit (Thermo Fisher Scientific) according to the manufacturer's instructions. DNA spreading was performed on carbon-coated 400-mesh nickel grids (G2400N, Plano GmbH) using the benzyl-dimethyl-alkyl-ammonium chloride (BAC) method, followed by platinum coating with a high vacuum evaporator (EM BAF060, Leica) as described by (Zellweger and Lopes, 2018).

EM imaging was conducted using a Talos 120 transmission electron microscope (FEI; LaB6 filament, accelerating voltage ≤120 kV) equipped with a bottom-mounted CMOS BM-Ceta camera (4096 × 4096 pixels). Automated imaging was performed at 28,000× magnification using MAPS 3 software (Thermo Fisher Scientific). Replication intermediates were annotated with the MAPS offline viewer (V3.28) and extracted for further analysis. Scoring of replication intermediates was conducted in Fiji under blinded conditions. For each experimental condition, a minimum of 65 replication fork structures were evaluated across three independent biological replicates. Statistical significance was assessed using Welch's $t$-test ($t = 7.98$, df = 4, $p = 0.00133$) and the graph was generated in R-Studio. "$n$" indicates biological replicates.

### Image and statistical analysis IF and SIRF

IF/SIRF images were converted into a file suitable for Olympus ScanR image analysis software (version 3.0.1). Using the DAPI channel, nuclei were set as the main analysis objects and using the spot detection module number of SIRF signal foci per nucleus were analyzed. Alternatively, using the mean intensity module, the mean signal intensity of the corresponding signal (i.e., H2Bub signal) was analyzed. Raw data from ScanR software were extracted for EdU-positive cells, and values were transferred to Prism 10 software for statistical analysis using the Kruskal–Wallis method. Graphical representation of each experiment was performed using Spotfire data visualization software (TIBCO, version 7.0.1). For each condition, at least 120 EdU-positive cells were analyzed.

### Image and statistical analysis of fibers

Images acquired using a fluorescent microscope were manually analyzed in Fiji software (NIH). The pixel values were converted to μm using the formula 1 pixel is equal to 0.146 μm. Depending on the experimental design, either IdU lengths or IdU/CldU tract length ratios were transferred to Prism software (version 10) for statistical analysis using the Kruskal–Wallis method and graphical illustration of the results. For each sample, at least 120 DNA fibers

were analyzed. A blinding approach was used to ensure unbiased analysis. Samples were assigned random codes, and evaluators had no access to information regarding their source or classification. This process remained in place until all data collection and analysis were finalized.

## Data availability

All data generated or analyzed for this study are available within the paper. This study includes no data deposited in external repositories.

The source data of this paper are collected in the following database record: biostudies:S-SCDT-10_1038-S44318-025-00602-1.

## Peer review information

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

## Acknowledgements

We thank Ramona N Moro, Julius Thomas, and the other members of the Penengo group and IMCR for technical support and helpful discussions. We are grateful to Daniel Durocher for sharing RNF169 plasmids. This work was supported by the Swiss National Science Foundation (SNSF; grants 310030_219393) to ML; by ERC award (ChOReS, grant no. 10107875), NWO-Vidi funding (project no. 114122) and Oncode Institute, which is partly financed by the Dutch Cancer Society (grant no. 115886) to NT; by the Swiss National Science Foundation (SNSF; grants 310030_220022/1 and 310030_184966/1), by the Swiss Cancer Research foundation (KFS-4577-08-2018) and by the Worldwide Cancer Research (22-0181) to LP.

## Author contributions

**Filip D Duzanic**: Data curation; Formal analysis; Methodology; Writing—original draft. **Vaishnavi Mohana-Natarajan**: Data curation. **Samuele Fisicaro**: Data curation; Formal analysis; Methodology. **Collin Bakker**: Data curation; Formal analysis. **Moses Aouami**: Data curation. **Gabriel Amaral**: Data curation. **Massimo Lopes**: Supervision. **Nitika Taneja**: Supervision. **Lorenza Penengo**: Conceptualization; Supervision; Funding acquisition; Writing—original draft; Writing—review and editing.

Source data underlying figure panels in this paper may have individual authorship assigned. Where available, figure panel/source data authorship is listed in the following database record: biostudies:S-SCDT-10_1038-S44318-025-00602-1.

## Disclosure and competing interests statement

NT holds an international patent for ChromStretch technology filed under PCT/NL2023/050120. No other authors have competing interests.

# Expanded View Figures

**Figure EV1. H2BK120ub transiently accumulates at the stressed DNA replication forks in an ATR-dependent manner (related to Fig. 1).**

(A, B) Immunoblot analysis of U2OS cell extracts upon siLUC and siRNF20 using the indicated antibodies. (C, D) Representative images (C; scale bars: 5 μm) and quantitative analysis of EdU-labeled U2OS cells upon siLUC and siRNF20, stained with H2BK120ub, DAPI and EdU antibodies. Quantification of the average nuclear H2BK120ub signal intensity per EdU-positive and EdU-negative cells (at least 120 cells were analyzed per condition; ****$P < 0.0001$, ***$P < 0.001$, ns nonsignificant, $P$(siRNF20 EdU pos. vs siRNF20 EdU neg) $= 0.0002$; $P$ values were calculated by one-way ANOVA, followed by Kruskal–Wallis test). Bars represent median values. $n = 3$ experiments. (E, F) Scatter plot representing the H2BK120ub signals (E) or the H2BK120ub SIRF (F) per nucleus throughout the cell cycle, based on the total DAPI intensity and EdU mean intensity, of siLUC or siRNF20-treated cells as in Fig. 1A. Each dot represents an individual nucleus. Dots are colored according to increasing average H2BK120ub or H2BK120ub SIRF signal intensities. (G, H) Immunoblot of U2OS cell extracts, using the indicated antibodies (G), and H2BK120ub SIRF analysis (H) upon transfection of different siRNAs, optionally treated with HU (4 mM, 2 h). At least 150 cells were analyzed per condition. ****$P < 0.0001$, $P$ values determined using one-way ANOVA coupled with Kruskal–Wallis test. $n = 3$ experiments. (I, J) Immunoblot of U2OS cell extracts, using the indicated antibodies, and RNF20 SIRF analysis upon siLUC and siRNF20, optionally treated with CPT and HU as in Fig. 1A. At least 120 EdU-positive cells were analyzed per condition. ****$P < 0.0001$, ns nonsignificant, $P$ (siLUC UT vs. siLUC CPT) $> 0.9999$, $P$ (siLUC UT vs. siLUC HU) $> 0.9999$, $P$ (siRNF20_1 UT vs. siRNF20_1 HU) $> 0.9999$; $P$ values were determined by one-way ANOVA followed by Kruskal–Wallis test. (K) Whiskers plot indicating H2BK120ub signal intensity per individual EdU-labeled chromatin fiber in untreated and HU-treated (4 mM, 1 h) U2OS cells. **$P = 0.0025$; $P$ values were determined using the Mann–Whitney $U$-test (nonparametric). Box plots show the 25th, 50th (median), and 75th percentiles; whiskers indicate the 10th and 90th percentiles; "+" represents the mean. $n = 3$ experiments. (L) Whiskers plot indicating H3K9me3 signal intensity per individual EdU-labeled chromatin fiber in untreated and HU-treated (4 mM, 1 h) U2OS cells, upon siLUC and siRNF20. ****$P < 0.0001$, ns nonsignificant, $P$ (siLUC UT vs siRNF20_1 UT) $= 0.2372$, $P$ (siLUC HU vs siRNF20_1 HU) $> 0.9999$; $P$ values were determined using Kruskal–Wallis test, followed by Dunn's multiple comparisons test. Box plots show the 25th, 50th (median), and 75th percentiles; whiskers indicate the 10th and 90th percentiles; "+" represents the mean. $n = 3$ experiments. (M) RNF20 mRNA levels were quantified by qPCR in U2OS cells transfected with siLUC or siRNF20. (N, O) Scatter plot representing the H2BK120ub SIRF per nucleus throughout the cell cycle, based on the total DAPI intensity and EdU mean intensity, of cells treated as in Fig. 1J, K, respectively.

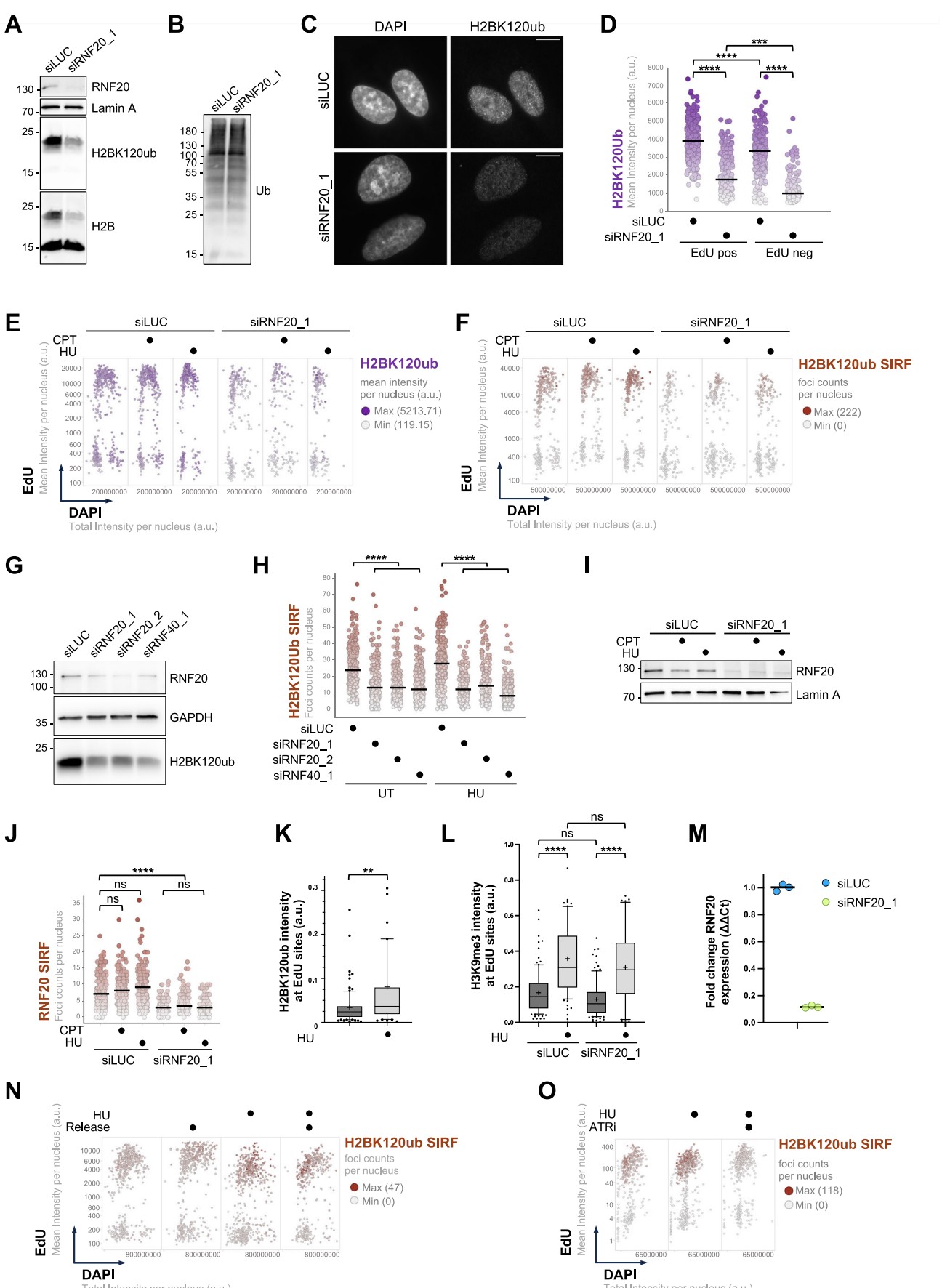

**A**

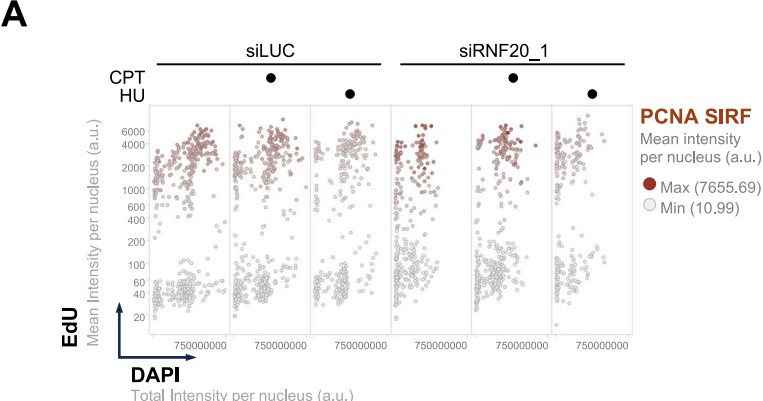

**B**

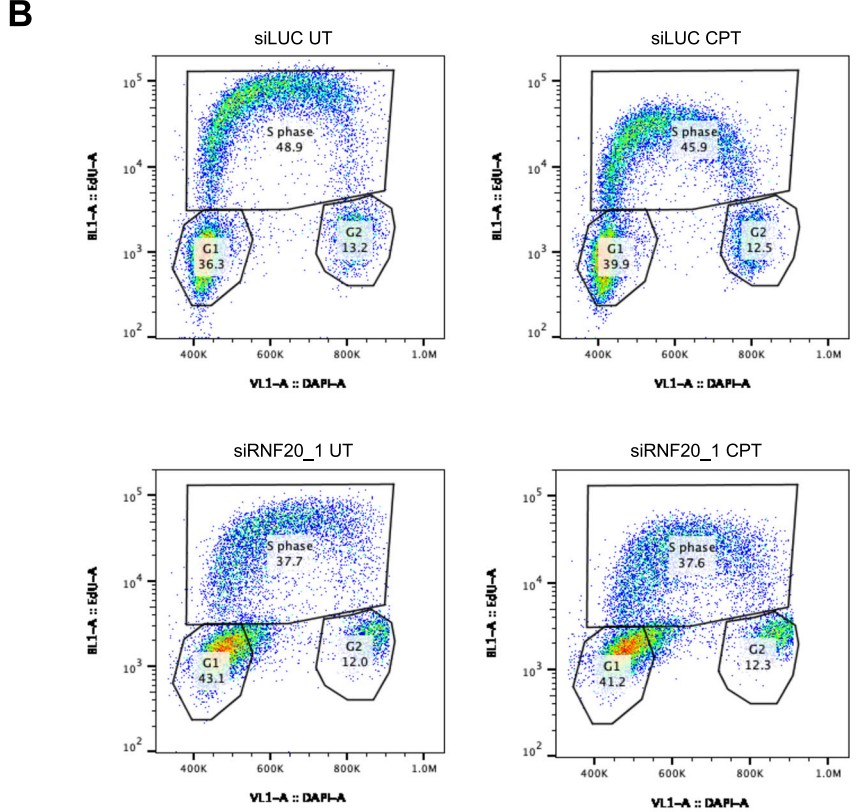

**C**       **D**       **E**

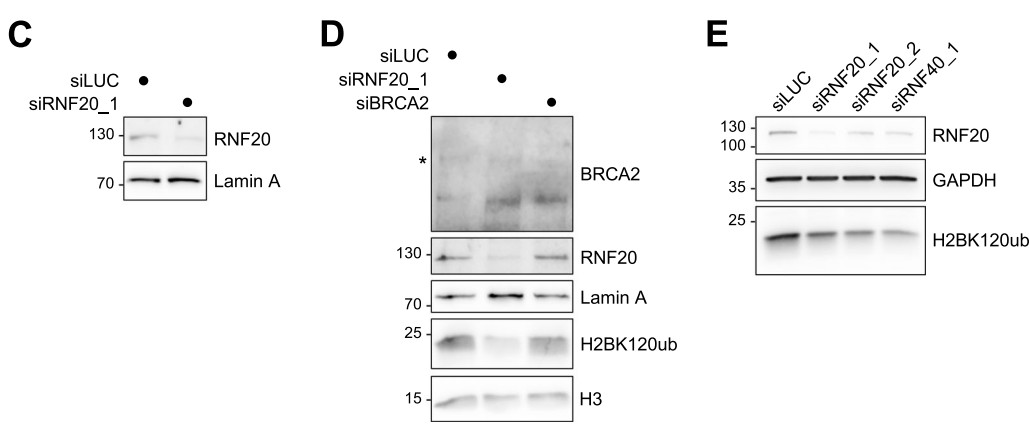

◀  **Figure EV2.  Loss of RNF20/H2BK120ub leads to unrestrained fork progression without major disruption of the cell cycle (related to Fig. 2).**

(A) Scatter plot representing the PCNA SIRF per nucleus throughout the cell cycle, based on the total DAPI intensity and EdU mean intensity, of cells treated as in Fig. 2B. Each dot represents an individual nucleus, with color coding reflecting the increasing intensity of PCNA SIRF intensity. (B) Flow cytometry analysis of the EdU incorporation (5 µM, 30 min) by control and U2OS cells depleted of RNF20 in unperturbed conditions or upon mild replicative stress (100 nM CPT 30 min pretreatment + 30 min during EdU labeling). $n = 3$ experiments. (C) Immunoblot analysis of RNF20 levels in siLUC- and siRNF20-transfected U2OS cells related to Fig. 2C. (D) Immunoblot analysis using the indicated antibodies of cell extracts upon siLUC, siRNF20 or siBRCA2 related to Fig. 2D. The asterisk indicates the size of BRCA2. (E) Immunoblot analysis using the indicated antibodies of extracts of U2OS cells upon siLUC, siRNF20_1, siRNF20_2, or siRNF40_1 related to Fig. 2F.

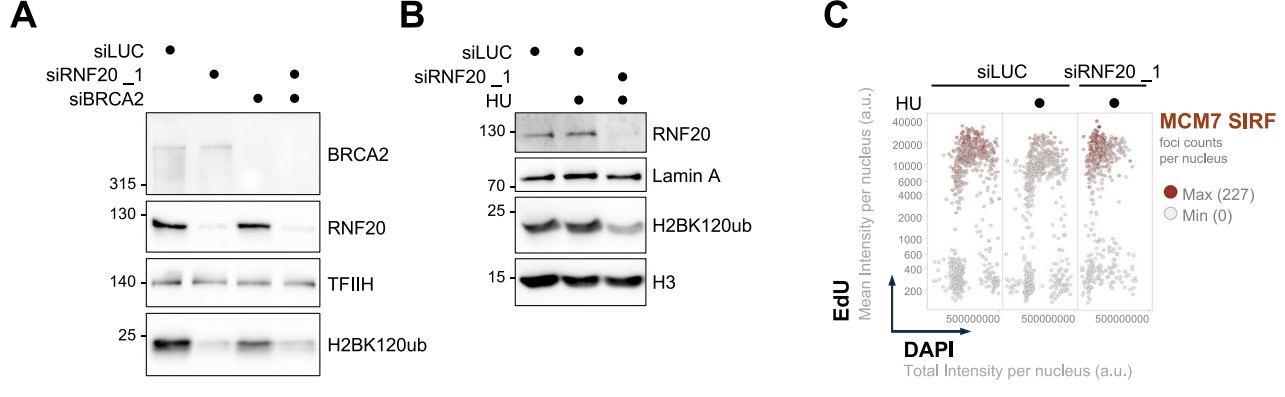

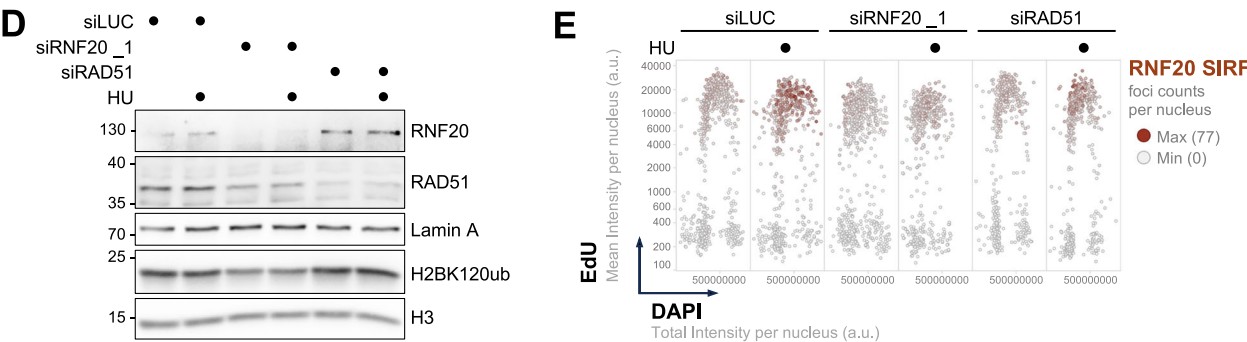

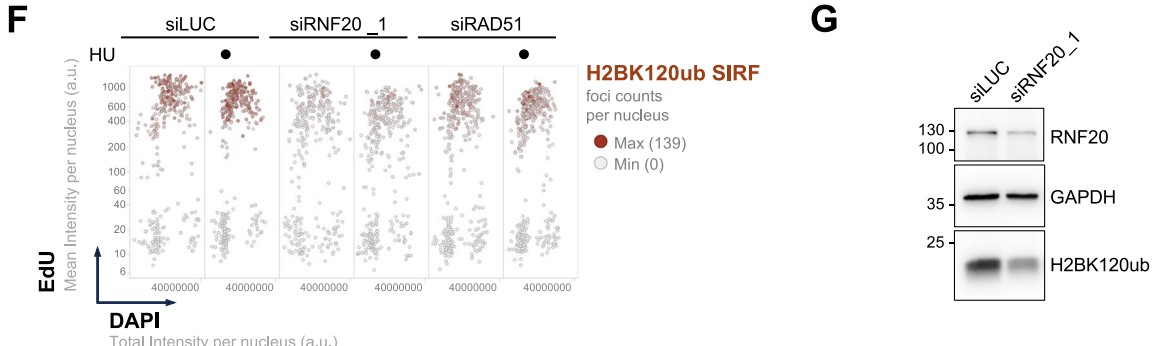

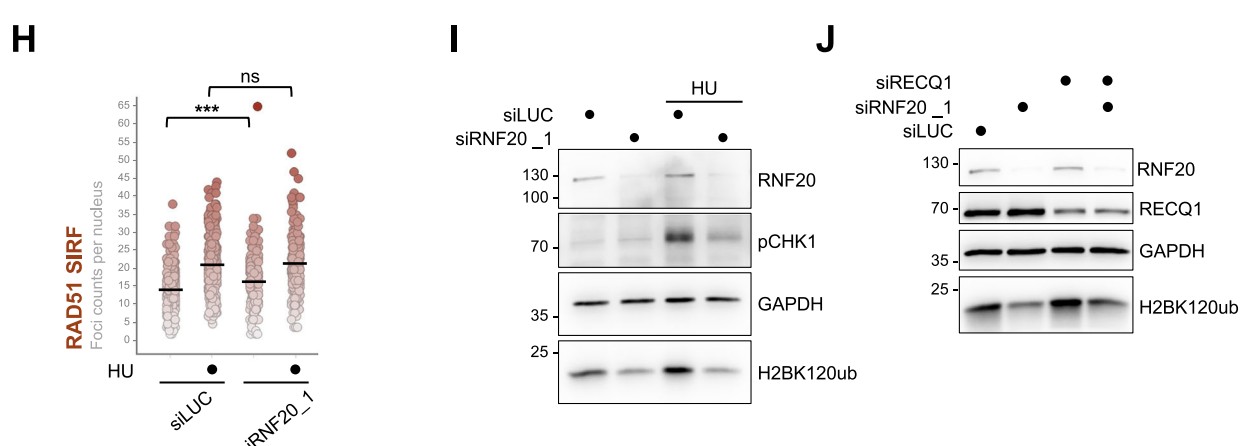

◄ **Figure EV3. Loss of RNF20/H2BK120ub perturbs replication fork dynamics (related to Fig. 3).**

(A) Immunoblot of U2OS cell extracts, using the indicated antibodies, upon transfection with siLUC, siRNF20_1 or siBRCA2. Related to Fig. 3A. (B) Immunoblot analysis using the indicated antibodies of extracts of U2OS cells transfected with siLUC and siRNF20_1, optionally treated with HU. Related to Fig. 3C. (C) Scatter plot representing the MCM7 SIRF per nucleus throughout the cell cycle, based on the total DAPI intensity and EdU mean intensity, of cells treated as in Fig. 3C. Each dot represents an individual nucleus, with color coding reflecting the increasing intensity of MCM7 SIRF intensity. (D) Immunoblot of U2OS cell extracts, using the indicated antibodies, upon transfection with siLUC, siRNF20_1 or siRAD51. Related to Fig. 3E,F. (E, F) Scatter plot representing the RNF20 SIRF and H2BK120ub per nucleus throughout the cell cycle, based on the total DAPI intensity and EdU mean intensity, of cells treated as in Fig. 3E,F, respectively. Each dot represents an individual nucleus, with color coding reflecting the increasing intensity of SIRF intensity. (G) Immunoblot of U2OS cell extracts, using the indicated antibodies, upon transfection with siLUC and siRNF20_1. (H) Scatter plot representing the RAD51 SIRF per nucleus throughout the cell cycle, based on the total DAPI intensity and EdU mean intensity, upon siLUC and siRNF20_1. Each dot represents an individual nucleus, with color coding reflecting the increasing intensity of RAD51 SIRF intensity. At least 150 EdU-positive cells were analyzed per condition. \*\*\*$P < 0.001$, ns nonsignificant, $P$ (siLUC HU vs. siRNF20_1 HU) > 0.9999, $P$ (siLUC UT vs. siRNF20_1 UT) = 0.0009; $P$ values were determined using one-way ANOVA followed by Kruskal–Wallis test. (I) Immunoblot analysis using the indicated antibodies of extracts of U2OS cells transfected with siLUC and siRNF20_1, optionally treated with HU. Related to Fig. 3I. (J) Immunoblot analysis using the indicated antibodies of extracts of U2OS cells transfected with siLUC, siRNF20_1 or siRECQ1. Related to Fig. 3G.

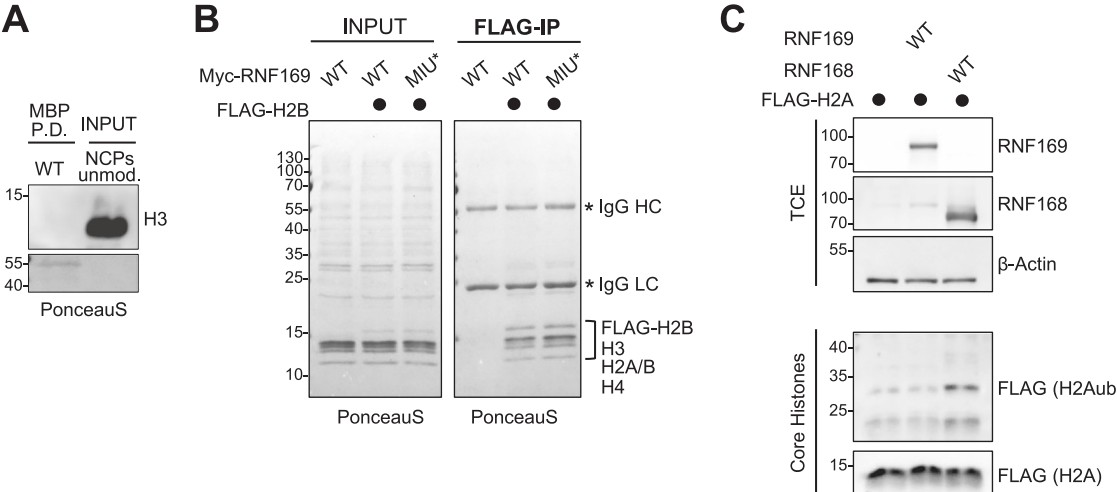

**Figure EV4. RNF169 is a reader of H2BK120ub (related to Fig. 4).**

(A) Pulldown assay using the C-terminal domain of RNF169 (amino acids 662–708) and unmodified nucleosome core particles (NCPs), followed by immunoblot. (B) Ponceau S staining of the membranes in Fig. 4D. (C) Immunoblot analysis on TCE and histone extracts of HEK293T cells transiently transfected with the indicated plasmids. $n = 3$ experiments.

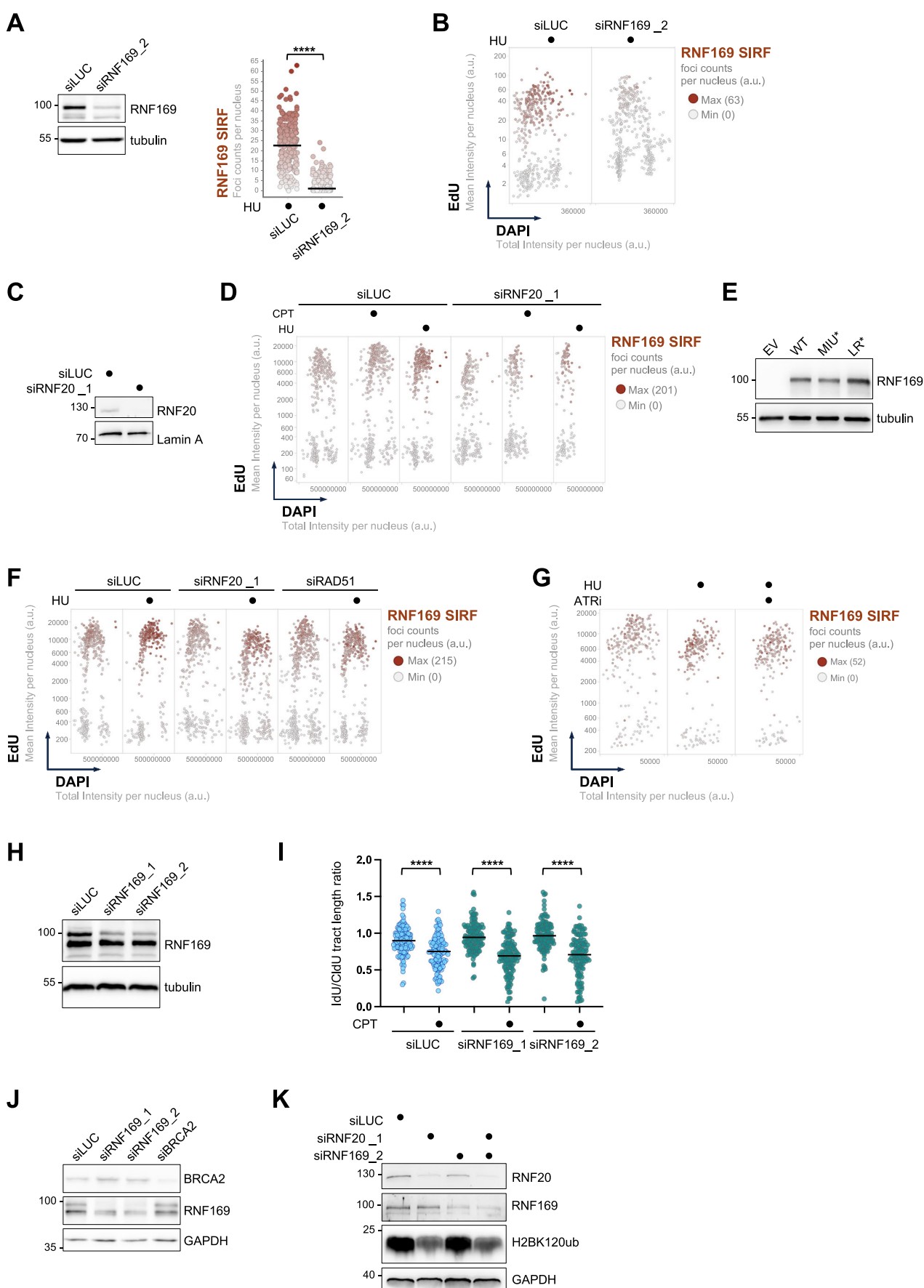

◀ **Figure EV5. RNF169 is recruited to the stressed DNA replication forks via H2BK120ub to limit fork resection (related to Fig. 5).**

(A, B) RNF169 immunoblot and RNF169 SIRF analysis upon optional treatment with HU (4 mM, 2 h) in U2OS upon siLUC and siRNF169 transfection. Quantification of the RNF169 SIRF foci in EdU-positive nuclei in indicated conditions (at least 120 EdU-positive cells analyzed per condition; ****$P < 0.0001$, $P$ value calculated using unpaired $t$-test). Bars indicate median values. $n = 3$ experiments. Color code in B indicates RNF169 SIRF foci numbers in individual nuclei across the cell cycle. (C, D) Immunoblot on U2OS cell extracts, using the indicated antibodies, upon transfection with siLUC, siRNF20_1 (C), and scatter plot representing the RNF169 SIRF per nucleus throughout the cell cycle, based on the total DAPI intensity and EdU mean intensity (D), of cells treated as in Fig. 5B. Each dot represents an individual nucleus, with color coding reflecting the increasing intensity of SIRF. (E) Immunoblot on U2OS cell extracts upon transient expression of the Myc-tagged forms of RNF169 (related to Fig. 5C). (F, G) Scatter plot representing the RNF169 SIRF per nucleus throughout the cell cycle, based on the total DAPI intensity and EdU mean intensity, of cells treated as in Fig. 5D,E, respectively. Each dot represents an individual nucleus, with color coding reflecting the increasing intensity of SIRF intensity. (H, I) U2OS cells were transfected with siLUC, siRNF169_1 and siRNF169_2 to assess replication fork progression in unperturbed conditions and following mild replicative stress (CPT 100 nM, 30 min; I) as in Fig. 2E,F (at least 115 fibers analyzed per condition; ****$P < 0.0001$; $P$ values were determined using Mann–Whitney $U$-test. Bars represent median values). Immunoblot on U2OS cell extracts reveals RNF169 protein levels (H). $n = 2$ experiments. (J, K) Immunoblot on U2OS cell extracts upon transfection with the indicated siRNAs, related to Fig. 5J–L.

