## [Peer Review File · The EMBO Journal]

H2BK120ub and its reader RNF169 sequentially regulate replication fork remodeling and stability

Filip Duzanic, Vaishnavi Mohana-Natarajan, Samuele Fiscaro, Collin Bakker, Moses Aouami, Gabriel Amaral, Massimo Lopes, Nitika Taneja, and Lorenza Penengo

Corresponding author(s): Lorenza Penengo (penengo@imcr.uzh.ch)

Review Timeline:

Submission Date:	7th Mar 25
Editorial Decision:	14th May 25
Revision Received:	12th Sep 25
Editorial Decision:	29th Sep 25
Revision Received:	2nd Oct 25
Accepted:	6th Oct 25

Editor: Hartmut Vodermaier

Transaction Report:

Dear Lorenza,

Thank you for your patience during the evaluation of your manuscript on RNF20 and RN169 regulation of replication fork dynamics. We have received the input of three expert referees, and have now also had the chance to discuss the below-copied reports in detail within our team. I am afraid that none of the referees is strongly in favor of publication of the study, at least not in the present form. The reviewers raise a number of important technical issues that would need to be addressed (such as insufficient controls for siRNA specificities, discrepancies between HU and CPT effects, antibody specificity in IF vs WB experiments...); but they also have more substantive conceptual reservations regarding the conclusiveness of two crucial aspects of the study: the proposed interplay/epistasis between RNF20 and RNF169, and the proposed involvement of fork reversal/remodeling. In my view, these both would constitute key advances to make this study a strong candidate for The EMBO Journal, beyond the general implication of RNF20 in replication dynamics. At the same time, it is uncertain whether they can be satisfactorily improved in a sufficiently straightforward manner, also given the still unpublished competing work - therefore, a more limited revision focussed predominantly on the specific technical concerns and rapid publication in our sister journal EMBO Reports might be a viable alternative here. In this situation, I would invite you to carefully consider the attached reports together with your coworkers, and send me a tentative point-by-point response to the referees' criticisms by early next week. I would then be happy to once more talk directly with you about the work and possible options later in the week, so that we could hopefully find a good way forward here.

Best regards, and looking forward to hearing from you,

Hartmut

Referee #1 (Report for Author)

This manuscript presents a study investigating the role of H2BK120 monoubiquitination

(H2BK120ub) and its E3 ligase RNF20 in replication fork remodeling. It further explores the function of RNF169 as a proposed reader of H2BK120ub at stalled forks, contributing to the protection of nascent DNA from degradation. These findings aim to expand our understanding of chromatin-associated regulation of replication stress responses. While the study uncovers a novel chromatin modification pathway influencing fork remodeling, several mechanistic gaps and interpretative overextensions limit the current impact of the work. In particular, the phenotypic divergence between RNF20 and RNF169 depletion is not reconciled, challenging the proposed sequential model. Therefore, publication in the current form is not recommended.

Major Concerns

1. The manuscript proposes that H2BK120ub (via RNF20) recruits RNF169 to stressed forks to stabilize reversed structures. However, the phenotypes of RNF20 and RNF169 depletion differ significantly:

RNF20 depletion prevents fork slowing and suppresses degradation in BRCA2-deficient cells, suggesting a role in promoting fork reversal.

In contrast, RNF169 depletion leads to fork degradation, indicating a role in protecting reversed forks.

These divergent outcomes imply that RNF20 and RNF169 act in functionally distinct, possibly independent branches of the fork remodeling process. The current data do not support a sequential epistasis model, as implied by the title.

2. The authors demonstrate H2BK120ub accumulation at stalled forks and show that RNF20 depletion impairs fork slowing. However, no mechanistic insight is provided into how H2BK120ub facilitates fork reversal, such as:

visualization of reversed forks (e.g., EM, S1-seq), interactions with RAD51 or fork remodeling enzymes, effects on chromatin accessibility or nucleosome positioning.

This is especially important in light of growing evidence that chromatin context dictates whether forks reverse or undergo repriming.

The authors are encouraged to focus on uncovering how RNF20-H2BK120ub contributes to fork reversal at the chromatin level, potentially through structural or genomic chromatin assays. Clarifying this would substantially elevate the manuscript's impact.

3. The manuscript refers to RNF169 as a "reader" of H2BK120ub at replication forks.

However:

The supporting assays (Fig. 4C-E) were conducted in cell lines under overexpression, without replication stress conditions.

The binding context is not demonstrated at endogenous expression levels or during active

fork remodeling.

While the in-cell interaction is suggestive, the term "reader" carries the implication of context-specific and functionally relevant recognition. Without supporting evidence under stress conditions, this claim remains premature.

The authors should either:

Provide additional evidence of replication stress-dependent recruitment or binding (e.g., ChIP-seq, CUT&RUN, or co-localization with fork-specific markers under HU/CPT treatment), or

Reframe RNF169 as a candidate effector that is functionally relevant at forks, without asserting definitive H2BK120ub reading.

This study provides an important starting point for understanding how H2BK120ub influences fork remodeling. However, key conclusions-particularly the proposed RNF20-H2BK120ub-RNF169 axis-are not fully supported by the data. A future revision that emphasizes mechanistic analysis of RNF20-H2BK120ub in fork reversal would be a valuable contribution to the field.

Additional Suggestions

Clarify whether H2BK120ub functions in parallel or coordination with other chromatin features, such as H3K9me3, in regulating PRIMPOL access and fork reversal.

Explore whether RNF20 influences fork remodeling enzymes or replisome components, providing mechanistic insight into how H2BK120ub alters fork dynamics.

Referee #2 (Report for Author)

In this study, Duzanic et al. report a function of histone H2B mono-ubiquitination at K120 (H2B-K120Ub), a key histone mark with important roles in promoting transcription and DNA repair, in replication stress responses and replication fork dynamics in human cells. They show that H2B-K120Ub catalyzed by the E3 ubiquitin ligase RNF20 accumulates at a subset of replication forks in response to fork stalling induced by hydroxyurea (HU). While RNF20 is dispensable for normal DNA replication and fork protection in response to HU treatment, its depletion impairs reduced replication fork progression after mild replication stress induced by treatment with camptothecin (CPT). Moreover, RNF20 knockdown suppresses checkpoint activation upon CPT or HU treatment and rescues the replication fork protection defect caused by BRCA2 depletion, involving a potential role of RNF20 in promoting replication fork reversal. The authors go on to show that H2B-K120Ub provides a docking platform at stalled forks for the E3 ubiquitin ligase RNF169, which has previously been implicated in DNA double-strand break repair. Finally, they show that RNF169

depletion has no impact on replication fork speed under normal and stressed conditions but impairs fork protection after HU treatment, which can be rescued by concomitant RNF20 knockdown. Based on these data, the authors propose a model wherein replication stress induces fork-associated RNF20-dependent H2B-K120Ub ubiquitination, driving fork reversal and RNF169 recruitment to protect nascent DNA from nucleolytic degradation.

The identification of a role of H2B-K120Ub in replication fork remodeling and stability in human cells is interesting and aligns with previous studies in yeast establishing a function of this histone mark in replication stress responses. In its present form, however, the key conclusions of the manuscript are not fully supported by the data and the mechanistic basis of how RNF20-dependent H2B-K120Ub formation promotes fork remodeling and stability via RNF169 and possibly other factors remains far from clear. Clarification and extension of several key points, as elaborated below, would clearly strengthen this study.

Specific points:

1. All experiments probing the impacts of RNF20 depletion were performed using a single siRNA. At least some key phenotypes should be validated with an independent RNF20 siRNA or a different RNF20 depletion modality and, ideally, rescue experiments involving complementation with an RNF20 expression construct. The study would also benefit from addressing whether depletion of RNF40, the binding partner of RNF20 in the H2B-K120 E3 ubiquitin ligase complex, recapitulates key impacts of RNF20 knockdown.

2. It remains unclear whether the impacts of RNF20 depletion on replication fork dynamics are primarily related to direct (H2B-K120Ub deposition at forks) or indirect (H2B-K120Ub-dependent transcriptional impacts) effects, considering that replication stress increases both fork-associated and total nuclear H2B-K120Ub abundance (Fig. 1A-E). In this context, the observation that RNF20-depleted cells display a defective response to CPT but not HU (Fig. 2E-G) seems at odds with the notion that HU but not CPT increases H2B-K120Ub abundance (Fig. 1A-E).

3. Page 7, line 14-16: The authors speculate that "Intriguingly, the chromatin fibers data suggested that only a fraction of forks accumulated H2BK120ub, possibly reflecting the subset of remodeled forks (Fig. 1G-I; S1F-H)." Can they provide more direct evidence in support of this hypothesis, e.g. by testing whether replication stress-dependent H2B-K120Ub accumulation specifically coincides with markers of reversed forks?

4. Fig. 3F: The impact of siRNF20 on blocking checkpoint activation is notable. The authors suggest that this may be due to defective fork reversal, but this remains speculative. Can they provide further evidence substantiating this idea?

5. Fig. 4: It is difficult to conclusively ascertain from these experiments whether RNF169 is a specific reader of H2B-K120Ub or merely pulls down H2B-K120Ub-modified NCPs in vitro because it is a ubiquitin-binding protein. It would be useful to test whether proteins like RNF168 and RAD18 that recognize H2A-K13/K15Ub similar to RNF169 also pull down H2B-K120Ub-modified NCPs, or whether this is exclusive to RNF169. Does mutation of the LRM motif adjacent to MIU2 impair RNF169 binding to H2B-K120Ub-modified nucleosomes?

6. The studies of the potential role of RNF169 as an effector of replication stress responses downstream of H2B-K120Ub formation (Fig. 5) appear quite premature and need to be further developed. Given the authors' observation that RNF20 is important for recruiting RNF169 to stressed replication forks, it seems somewhat counterintuitive that RNF169 depletion compromises fork protection while RNF20 knockdown has no impact in itself but rescues the fork protection defect in RNF169-depleted cells. Further studies are clearly needed to better delineate the possible fork-protective role of RNF169 and how this relates to RNF20-dependent H2B ubiquitination.

7. Following on from the point above: The model put forward by the authors (Fig. 6) does not seem to be fully supported by the data. The authors state that "the accumulation of H2BK120 in turn serves as a docking platform for the recruitment of RNF169 to the reversed replication forks, which protects nascent DNA from unscheduled nucleolytic degradation" (page 13, line 21-23), but this appears inconsistent with the observation that RNF20 knockdown has no impact on fork protection unlike RNF169 depletion despite RNF20 is required for recruiting RNF169 to stressed forks.

Additional points:

8. The PCNA input blot in Fig. 1F is of poor quality and should be replaced.

9. Page 6, line 7-10: "The results showed that RNF20 depletion did not cause a dramatic alteration of the cell cycle progression, but only a limited delay of cell entry into the S-phase and consequent accumulation in G1-phase (Fig. S2A), thereby the defective checkpoint activation is likely due to the loss of RNF20." It is unclear what defective checkpoint activation the authors are referring to here - please consider re-phrasing.

10. Whether the observed increase in H2B-K120Ub abundance caused by RNF169 overexpression (Fig. 4D,E) has any functional significance is not clear. Does RNF169 depletion reduce H2B-K120Ub levels? Does RNF169 overexpression have any impact on DNA replication and fork integrity?

11. The organization of panels in some figures is not logical (e.g. A,B,C,F,D,E... in Fig. 3 and A,B,D,C,E,F,H,I,G... in Fig. 5).

Referee #3 (Report for Author)

In their study "H2BK120ub and its reader RNF169 sequentially regulate replication fork remodeling and stability" Duzanic and colleagues present a combination of cell and molecular biological data to describe a role for the ubiquitination of H2B at K120 by RNF20 in the recruitment of RNF169 at sites of stalled replication forks, suggesting that the recruitment of RNF169 by this mechanism promotes fork stability by protecting against nuclease digestion of otherwise vulnerable stalled fork DNA.

Overall I am supportive of this study, although I have a variety of recommendations and concerns that would be good to address.

Major concerns and suggestions:

1. The study utilizes lower dose CPT and higher dose HU to produce scenarios of replication stress with differing severity. To me, a vital control that is not experimentally verified is the relative amount of DNA double strand breaks (DSBs) that occur (or not) following each of these treatments in the U2OS cells specifically used in this work, and confirming DSB levels for each siRNA treatment the team deploys to ask various questions. Controlling for DNA replication fork collapse into DSBs is highly relevant here, as it influences how the reader interprets any experimental outcomes where loss of DNA damage signalling and/or repair factors are involved.

2. Throughout this work, the depletion of RAD51 is used (and interpreted) as a means to modulate fork reversal and largely discussed with that in mind. My concern is that, if cellular conditions are also coincident with sufficient replication-associated DSBs that also vary with experimental conditions, then the relative abundance of RAD51 will also exert an influence of rates of DSB repair and therefore steady state DSB signalling to ATR and ATM, which will in turn influence RNF20 activation and downstream effects. Hence, it is important that the authors control for DSB formation across all cellular conditions, and

also discuss what implications that DSB signalling might have on the processes they are investigating.

3. While some siRNA efficiency controls are shown throughout this work, many are missing altogether or shown for only some experiments but not others. In my view, siRNA efficiency and specificity controls are required for all targets in all different experimental contexts and need to be shown first. For example, as far as I can see, we are never shown controls that demonstrate how well the siRNA targeting RAD51 works.

4. Further to my concern above, proving the specificity of siRNA effects using "add-back" siRNA resistant wildtype (and, if relevant, catalytic-dead versions) of key factors - especially RNF20, RAD51, BRCA2 - has not been done. Given the reliance of this study on siRNA to demonstrate mechanism, these controls are vital as everything the authors claim rests on these siRNA being robust. I would also encourage use of gene-edited cells in place of siRNA to prove key experiments.

5. The antibody controls for the H2BK120ub are demonstrated using immunoblot but then often deployed using immunofluorescence (IF). The outcome of antibody quality control between these methods is not always interchangeable, and very often an antibody that works well and is specific under denaturing conditions of a SDS-PAGE western blot is non-specific in IF. How can the authors be certain that their antibody is specific using microscopy? IF controls are needed.

6. A key hypothesis from this work is that the RNF20 to RNF169 pathway is helping to protect stalled DNA replication forks against nucleolytic degradation. If correct, then depletion or inhibition of nucleases associated with this degradation would be expected to alleviate the need for these factors in fork stability and can be tested using the assays presented in this work. As this is a very logical experiment that I consider foundational to the final hypothesis, I would encourage this to be done as part of a revision.

7. A variety of labs have demonstrated that RNF20 function is co-dependent on RNF40 - is that the case here? Do the authors observe their primary phenotypes of interest to be the same or similar between siRNF20 and siRNF40?

8. RNF20 mediated ubiquitination of chromatin promotes recruitment of chromatin remodelers, with several labs demonstrating that SNF2H is amongst these (most recently PMID: 37155876). Recruitment of SNF2H via RNF20 can exert influence over homologous recombination (e.g. PMID: 24357716) and DSB more generally, which matters in the context of DNA replication associated stress with the potential for DSB formation. Assessing the presence of (previously discovered) RNF20-dependent chromatin remodelers such as SNF2H could be evaluated using the SIRF approach, and would substantially enhance this work by determining if those pathways apply more universally (i.e. everywhere RNF20 is active) or are perhaps more selective.

Minor concerns:

9. Figures:

a. The placement of figure panels in this study are somewhat confusing to the reader. For example, the placement of Figure 3F and 5G are essentially non-sensical relative to panels that should come earlier/later to the reader.

b. On many graphs, font sizes of labels on axes are very small and hard to read at 100% zoom on a standard page. Generally speaking, font sizes and types need to be harmonized.

c. Figure S2A does not show cell cycle data as indicated in the description within the results.

d. Many SIRF figures are essentially duplicated outcomes / different presentation of the same from a single type of experiment. While I appreciate the authors showing figures such as 5E where we can see the scatters identifying EdU positive S-phase cells - it is really the quantification and statistical analysis of the specific protein of interest (in that case RNF169) in Figure 5D that matters and is more helpful. I would recommend moving most instances of these EdU vs DAPI charts to the supplemental data - perhaps retaining only the first occasion Figure 1B versus 1A to demonstrate how this is being done.

10. Why is the experiment in Figure 1H only done with n=2? N=3 is considered minimum acceptable.

11. In Figure 1K - the experiment shows a reduction after 1h post HU removal. How long does it take for signal to reach background levels?

12. On page 6, line 18, it would be better to say "reduced" H2Bk120ub is dispensable for XYZ, as based on the immunoblot controls the H2Bk120ub is clearly not entirely gone, as is implied.

13. On page 8 line 4, the authors employ a "data not shown" statement. As I understand it, it is not longer allowed to do this. This happens again on Page 10 line 27. Please show the data.

14. The data in figure 4D-E commenting on H2Bub signal differences requires quantifying somehow.

15. On page 10 line 18, using the word "significantly" immediately requires the author state the actual p-value for the condition they are claiming is statistically significant within the text.

Referee#1

This manuscript presents a study investigating the role of H2BK120 monoubiquitination (H2BK120ub) and its E3 ligase RNF20 in replication fork remodeling. It further explores the function of RNF169 as a proposed reader of H2BK120ub at stalled forks, contributing to the protection of nascent DNA from degradation. These findings aim to expand our understanding of chromatin-associated regulation of replication stress responses.

While the study uncovers a novel chromatin modification pathway influencing fork remodeling, several mechanistic gaps and interpretative overextensions limit the current impact of the work. In particular, the phenotypic divergence between RNF20 and RNF169 depletion is not reconciled, challenging the proposed sequential model. Therefore, publication in the current form is not recommended.

Major concerns

1. The manuscript proposes that H2BK120ub (via RNF20) recruits RNF169 to stressed forks to stabilize reversed structures. However, the phenotypes of RNF20 and RNF169 depletion differ significantly:

RNF20 depletion prevents fork slowing and suppresses degradation in BRCA2-deficient cells, suggesting a role in promoting fork reversal.

In contrast, RNF169 depletion leads to fork degradation, indicating a role in protecting reversed forks.

These divergent outcomes imply that RNF20 and RNF169 act in functionally distinct, possibly independent branches of the fork remodeling process. The current data do not support a sequential epistasis model, as implied by the title.

R: We note that the different outcome of RNF20- and RNF169-depletions does not exclude or contradict that the two factors may act at different steps of the same process. Inactivation of factors that promote fork slowing and reversal is not typically associated with defects in fork protection. Conversely, several factors that specifically protect reversed fork from degradation are not expected (and were not shown) to display defective fork slowing and remodelling.

Numerous examples of both scenarios have been previously reported (see e.g. Mijic et al., Nat Comms 2017; Berti et al., Nat Comms 2020; Liu et al., Sci Adv 2020). Perhaps the most notable example is the opposite effect observed upon RAD51 and BRCA2 depletion. In this case, BRCA2 depletion leads to a fork degradation phenotype that is not shared, and in fact rescued by RAD51 depletion. Multiple groups have shown that this reflects the BRCA2-independent role of RAD51 in the formation of reversed forks, which then act as entry points for nucleolytic degradation in BRCA2-defective cells. Hence, RAD51 and BRCA2 act sequentially in promoting reversed fork formation and protection, despite strikingly different molecular phenotypes.

We envision a similar scenario for RNF20/RNF169: RNF20-H2BK120ub are required to promote/stabilize reversed forks and their inactivation does not cause degradation of nascent DNA, but rather unrestrained fork progression. Conversely, RNF169 binds the H2BK120ub

mark and contributes to protecting nascent DNA from degradation on previously reversed forks. Hence, similarly to BRCA2 (and many other fork protection factors), RNF169 inactivation does not affect fork slowing, but does impair protection of stalled replication forks. To further support this point, we will perform additional experiments (outlined below).

2. The authors demonstrate H2BK120ub accumulation at stalled forks and show that RNF20 depletion impairs fork slowing. However, no mechanistic insight is provided into how H2BK120ub facilitates fork reversal, such as: visualization of reversed forks (e.g., EM, S1-seq), interactions with RAD51 or fork remodeling enzymes, effects on chromatin accessibility or nucleosome positioning.

This is especially important in light of growing evidence that chromatin context dictates whether forks reverse or undergo repriming.

The authors are encouraged to focus on uncovering how RNF20-H2BK120ub contributes to fork reversal at the chromatin level, potentially through structural or genomic chromatin assays. Clarifying this would substantially elevate the manuscript's impact.

R: As suggested by the reviewer, we will provide EM data to assess the effect of RNF20 depletion on the formation of reversed forks (we expected such comment, so we have already started a collaboration with the group of Massimo Lopes at our institute: they have done the experiment twice, and the third repetition is ongoing).

We will also evaluate the localization of RAD51 at stressed replication forks upon siRNF20 and the presence of H2BK120ub upon depletion of the fork remodeler ZRANB3 by SIRF.

3. The manuscript refers to RNF169 as a "reader" of H2BK120ub at replication forks.

However:

The supporting assays (Fig. 4C-E) were conducted in cell lines under overexpression, without replication stress conditions.

The binding context is not demonstrated at endogenous expression levels or during active fork remodeling.

While the in-cell interaction is suggestive, the term "reader" carries the implication of context-specific and functionally relevant recognition. Without supporting evidence under stress conditions, this claim remains premature.

The authors should either:

Provide additional evidence of replication stress-dependent recruitment or binding (e.g., ChIP-seq, CUT&RUN, or co-localization with fork-specific markers under HU/CPT treatment), or Reframe RNF169 as a candidate effector that is functionally relevant at forks, without asserting definitive H2BK120ub reading.

R: We will perform co-localization studies by PLA between RNF169 and RAD51. We will also assess the relevance of the LRM domain in H2BK120ub recognition in pulldown assays. In the worst-case scenario of negative or inconclusive results, we will reframe RNF169 as a candidate effector that is functionally relevant at forks, without asserting definitive H2BK120ub reading.

Additional suggestions

Clarify whether H2BK120ub functions in parallel or coordination with other chromatin features, such as H3K9me3, in regulating PRIMPOL access and fork reversal.

Explore whether RNF20 influences fork remodeling enzymes or replisome components, providing mechanistic insight into how H2BK120ub alters fork dynamics.

R: We will perform ChromStretch upon siRNF20 to assess whether RNF20 activity is required to deposit H3K9me3. We will also test whether the unrestrained fork progression phenotype observed upon siRNF20 depends on PRIMPOL or RECQ1. These experiments together with the EM data should address the major concerns.

Referee#2

In this study, Duzanic et al. report a function of histone H2B mono-ubiquitination at K120 (H2B-K120Ub), a key histone mark with important roles in promoting transcription and DNA repair, in replication stress responses and replication fork dynamics in human cells. They show that H2B-K120Ub catalyzed by the E3 ubiquitin ligase RNF20 accumulates at a subset of replication forks in response to fork stalling induced by hydroxyurea (HU). While RNF20 is dispensable for normal DNA replication and fork protection in response to HU treatment, its depletion impairs reduced replication fork progression after mild replication stress induced by treatment with camptothecin (CPT). Moreover, RNF20 knockdown suppresses checkpoint activation upon CPT or HU treatment and rescues the replication fork protection defect caused by BRCA2 depletion, involving a potential role of RNF20 in promoting replication fork reversal. The authors go on to show that H2B-K120Ub provides a docking platform at stalled forks for the E3 ubiquitin ligase RNF169, which has previously been implicated in DNA double-strand break repair. Finally, they show that RNF169 depletion has no impact on replication fork speed under normal and stressed conditions but impairs fork protection after HU treatment, which can be rescued by concomitant RNF20 knockdown. Based on these data, the authors propose a model wherein replication stress induces fork-associated RNF20-dependent H2B-K120Ub ubiquitination, driving fork reversal and RNF169 recruitment to protect nascent DNA from nucleolytic degradation.

The identification of a role of H2B-K120Ub in replication fork remodeling and stability in human cells is interesting and aligns with previous studies in yeast establishing a function of this histone mark in replication stress responses. In its present form, however, the key conclusions of the manuscript are not fully supported by the data and the mechanistic basis of how RNF20-dependent H2B-K120Ub formation promotes fork remodeling and stability via RNF169 and possibly other factors remains far from clear. Clarification and extension of several key points, as elaborated below, would clearly strengthen this study.

Specific points:

1. All experiments probing the impacts of RNF20 depletion were performed using a single siRNA. At least some key phenotypes should be validated with an independent RNF20 siRNA or a different RNF20 depletion modality and, ideally, rescue experiments involving complementation with an RNF20 expression construct. The study would also benefit from addressing whether depletion of RNF40, the binding partner of RNF20 in the H2B-K120 E3 ubiquitin ligase complex, recapitulates key impacts of RNF20 knockdown.

R: We have tested RNF20 knockout cells in the past, but they showed low viability because RNF20 is an essential gene, and they could not be used for DNA replication studies. For this reason, we preferred to adopt the siRNA approach in our experiments, which allowed us to achieve reduced levels of RNF20 but still good cell fitness. We will now include additional siRNAs targeting RNF20 in our analysis (DNA fiber and PLA), and we will also test the effect of the depletion of its binding partner, RNF40, as proposed by the reviewer.

The complementation approach would be problematic, both for time limitations and because the over-expression of the RNF20 ubiquitin ligase can exert detrimental effect on cell viability and should be strictly titrated.

2. It remains unclear whether the impacts of RNF20 depletion on replication fork dynamics are primarily related to direct (H2B-K120Ub deposition at forks) or indirect (H2B-K120Ub-dependent transcriptional impacts) effects, considering that replication stress increases both fork-associated and total nuclear H2B-K120Ub abundance (Fig. 1A-E). In this context, the observation that RNF20-depleted cells display a defective response to CPT but not HU (Fig. 2E-G) seems at odds with the notion that HU but not CPT increases H2B-K120Ub abundance (Fig. 1A-E).

R: RNF20 ubiquitin ligase is known to regulate transcription, but it does not trigger a universal decline in transcript levels when depleted. Instead, multiple studies report that RNF20 loss results in selective, gene-specific changes. For instance, Shema et al. (Gen&Dev, 2008) found that most genes maintain similar expression after RNF20 knockdown, a conclusion supported by Xie et al. (Gen Biol, 2017), who observed little change in genes with high H2Bub1. Moreover, to avoid the detrimental effect of acute RNF20 depletion on cell viability and DNA synthesis, we employ experimental conditions that reduce but do not completely abolish RNF20 (and H2BK120ub) levels. Hence, the overall effect on gene transcription is even more limited. To assess the effect of RNF20 depletion compared to transcription inhibition, we could potentially compare the levels of H2BK120ub in control, RNF20-depleted and cells treated with transcription inhibitors by WB.

It should be noted that CPT and HU have been used in different experiments at different concentrations and for different purposes. We apologize for this misunderstanding, and we will better clarify this in the text.

3. Page 7, line 14-16: The authors speculate that "Intriguingly, the chromatin fibers data suggested that only a fraction of forks accumulated H2BK120ub, possibly reflecting the subset of remodeled forks (Fig. 1G-I; S1F-H)." Can they provide more direct evidence in

support of this hypothesis, e.g. by testing whether replication stress-dependent H2B-K120Ub accumulation specifically coincides with markers of reversed forks?

R: Unfortunately, specific markers of reversed forks are not available. For this reason, we sought to evaluate the accumulation of H2BK120ub at stalled forks by SIRF. These data, together with those we will generate during revision (effect of RNF20 depletion on H3K9me3 deposition by ChromStretch analysis and on the formation of reversed forks by EM, role of PRIMPOL and RECQ1 on unrestrained fork progression induced by loss of siRNF20) will add more insights into the mechanism.

4. Fig. 3F: The impact of siRNF20 on blocking checkpoint activation is notable. The authors suggest that this may be due to defective fork reversal, but this remains speculative. Can they provide further evidence substantiating this idea?

R: We will provide EM data to assess the effect of RNF20 depletion on the formation of reversed forks.

5. Fig. 4: It is difficult to conclusively ascertain from these experiments whether RNF169 is a specific reader of H2B-K120Ub or merely pulls down H2B-K120Ub-modified NCPs in vitro because it is a ubiquitin-binding protein. It would be useful to test whether proteins like RNF168 and RAD18 that recognize H2A-K13/K15Ub similar to RNF169 also pull down H2B-K120Ub-modified NCPs, or whether this is exclusive to RNF169. Does mutation of the LRM motif adjacent to MIU2 impair RNF169 binding to H2B-K120Ub-modified nucleosomes?

R: This is certainly an important point. As suggested by the reviewer, to further assess if RNF169 binds specifically H2BK120ub embedded in the chromatin context, we plan to test whether mutation in the LRM motif, which helps target proteins to damaged chromatin, impairs the binding of RNF169 to H2BK120ub, by performing pulldown assay as in Fig. 4B. If time permits, we will also test the same mutant by PLA with H2BK120ub.

Conversely, in our opinion testing whether other proteins known to bind H2AK15ub, like RNF168 or RAD18, can also bind H2BK120ub will not help to ascertain the specificity of RNF169 towards chromatin, since it is possible that other DDR proteins can bind the same histone mark.

6. The studies of the potential role of RNF169 as an effector of replication stress responses downstream of H2B-K120Ub formation (Fig. 5) appear quite premature and need to be further developed. Given the authors' observation that RNF20 is important for recruiting RNF169 to stressed replication forks, it seems somewhat counterintuitive that RNF169 depletion compromises fork protection while RNF20 knockdown has no impact in itself but rescues the fork protection defect in RNF169-depleted cells. Further studies are clearly needed to better delineate the possible fork-protective role of RNF169 and how this relates to RNF20-dependent H2B ubiquitination.

R: To give additional proof of the role of RNF169 in fork protection via H2BK120ub binding, we will perform complementation experiments in RNF169 depleted cells upon re-expression of the siRNA-resistant forms of RNF169 WT, MIU and LRM mutant.

7. Following on from the point above: The model put forward by the authors (Fig. 6) does not seem to be fully supported by the data. The authors state that "the accumulation of H2BK120 in turn serves as a docking platform for the recruitment of RNF169 to the reversed replication forks, which protects nascent DNA from unscheduled nucleolytic degradation" (page 13, line 21-23), but this appears inconsistent with the observation that RNF20 knockdown has no impact on fork protection unlike RNF169 depletion despite RNF20 is required for recruiting RNF169 to stressed forks.

R: The fact that the outcomes of RNF20 and RNF169 depletion are different does not exclude they work in the same process, as already discussed above. We will clarify better this point in the text.

Additional points

8. The PCNA input blot in Fig. 1F is of poor quality and should be replaced.

R: We will provide a better blot

9. Page 6, line 7-10: "The results showed that RNF20 depletion did not cause a dramatic alteration of the cell cycle progression, but only a limited delay of cell entry into the S-phase and consequent accumulation in G1-phase (Fig. S2A), thereby the defective checkpoint activation is likely due to the loss of RNF20.". It is unclear what defective checkpoint activation the authors are referring to here - please consider re-phrasing.

R: We will re-phrase it to make it clearer.

10. Whether the observed increase in H2B-K120Ub abundance caused by RNF169 overexpression (Fig. 4D,E) has any functional significance is not clear. Does RNF169 depletion reduce H2B-K120Ub levels? Does RNF169 overexpression have any impact on DNA replication and fork integrity?

R: We consistently observed increased H2BK120ub upon RNF169 over-expression in several experiments, while no H2BK120ub reduction was observed upon depletion of RNF169. It is not clear if this has a functional significance. We did not test the effect of RNF169 over-expression on DNA replication.

11. The organization of panels in some figures is not logical (e.g. A,B,C,F,D,E... in Fig. 3 and A,B,D,C,E,F,H,I,G... in Fig. 5).

R: We will reorganized them.

Referee#3

In their study "H2BK120ub and its reader RNF169 sequentially regulate replication fork remodeling and stability" Duzanic and colleagues present a combination of cell and molecular biological data to describe a role for the ubiquitination of H2B at K120 by RNF20 in the recruitment of RNF169 at sites of stalled replication forks, suggesting that the recruitment of RNF169 by this mechanism promotes fork stability by protecting against nuclease digestion of otherwise vulnerable stalled fork DNA.

Overall I am supportive of this study, although I have a variety of recommendations and concerns that would be good to address.

Major concerns and suggestions

1. The study utilizes lower dose CPT and higher dose HU to produce scenarios of replication stress with differing severity. To me, a vital control that is not experimentally verified is the relative amount of DNA double strand breaks (DSBs) that occur (or not) following each of these treatments in the U2OS cells specifically used in this work, and confirming DSB levels for each siRNA treatment the team deploys to ask various questions. Controlling for DNA replication fork collapse into DSBs is highly relevant here, as it influences how the reader interprets any experimental outcomes where loss of DNA damage signalling and/or repair factors are involved. **R: We will assess the levels of DSBs in different experiments and treatments by monitoring gH2AX upon RNF20 depletion.**

2. Throughout this work, the depletion of RAD51 is used (and interpreted) as a means to modulate fork reversal and largely discussed with that in mind. My concern is that, if cellular conditions are also coincident with sufficient replication-associated DSBs that also vary with experimental conditions, then the relative abundance of RAD51 will also exert an influence of rates of DSB repair and therefore steady state DSB signalling to ATR and ATM, which will in turn influence RNF20 activation and downstream effects. Hence, it is important that the authors control for DSB formation across all cellular conditions, and also discuss what implications that DSB signalling might have on the processes they are investigating.

R: We will assess the levels of DSBs in different experiments and treatments by monitoring gH2AX upon RAD51 depletion.

3. While some siRNA efficiency controls are shown throughout this work, many are missing altogether or shown for only some experiments but not others. In my view, siRNA efficiency and specificity controls are required for all targets in all different experimental contexts and need to be shown first. For example, as far as I can see, we are never shown controls that demonstrate how well the siRNA targeting RAD51 works.

R: In all our experiments the levels of depletion efficiency is verified. We will provide these controls in the revised manuscript.

4. Further to my concern above, proving the specificity of siRNA effects using "add-back" siRNA resistant wildtype (and, if relevant, catalytic-dead versions) of key factors - especially RNF20, RAD51, BRCA2 - has not been done. Given the reliance of this study on siRNA to demonstrate mechanism, these controls are vital as everything the authors claim rests on these siRNA being robust. I would also encourage use of gene-edited cells in place of siRNA to prove key experiments.

R: We will perform complementation experiments for RNF169 to validate the specificity of siRNAs. Regarding RNF20, we will prove the specificity by using more siRNAs and will also include siRNAs targeting the binding partner RNF40, which is required for the function and stability of RNF20. As already mentioned, we have tested RNF20 knockout cells but they showed limited viability, as RNF20 is a rather essential gene.

5. The antibody controls for the H2BK120ub are demonstrated using immunoblot but then often deployed using immunofluorescence (IF). The outcome of antibody quality control between these methods is not always interchangeable, and very often an antibody that works well and is specific under denaturing conditions of a SDS-PAGE western blot is non-specific in IF. How can the authors be certain that their antibody is specific using microscopy? IF controls are needed.

R: We have tested the specificity of the H2BK120ub antibody upon RNF20 depletion both by Western blot (Fig. S1A,B) and by IF (Fig. 1A), by measuring the intensity of H2BK120ub signal in siLuc versus siRNF20. If needed, a representative image can be provided.

6. A key hypothesis from this work is that the RNF20 to RNF169 pathway is helping to protect stalled DNA replication forks against nucleolytic degradation. If correct, then depletion or inhibition of nucleases associated with this degradation would be expected to alleviate the need for these factors in fork stability and can be tested using the assays presented in this work. As this is a very logical experiment that I consider foundational to the final hypothesis, I would encourage this to be done as part of a revision.

R: As suggested by the reviewer, we will perform DNA fiber assay to measure fork degradation in RNF169-depleted cells upon optional treatment with MRE11 inhibitor Mirin, to test whether by inhibiting the main nuclease acting on the reversed forks could restore fork protection.

7. A variety of labs have demonstrated that RNF20 function is co-dependent on RNF40 - is that the case here? Do the authors observe their primary phenotypes of interest to be the same or similar between siRNF20 and siRNF40?

R: We will test the key phenotypes (by DNA fiber assay and SIF analysis) upon RNF40 depletion.

8. RNF20 mediated ubiquitination of chromatin promotes recruitment of chromatin remodelers, with several labs demonstrating that SNF2H is amongst these (most recently PMID: 37155876). Recruitment of SNF2H via RNF20 can exert influence over homologous

recombination (e.g. PMID: 24357716) and DSB more generally, which matters in the context of DNA replication associated stress with the potential for DSB formation. Assessing the presence of (previously discovered) RNF20-dependent chromatin remodelers such as SNF2H could be evaluated using the SIRF approach, and would substantially enhance this work by determining if those pathways apply more universally (i.e. everywhere RNF20 is active) or are perhaps more selective.

R: If time permits, we will test SNF2H localization by SIRF upon RNF20 depletion.

Minor concerns:

9. Figures:

- a. The placement of figure panels in this study are somewhat confusing to the reader. For example, the placement of Figure 3F and 5G are essentially non-sensical relative to panels that should come earlier/later to the reader.
- b. On many graphs, font sizes of labels on axes are very small and hard to read at 100% zoom on a standard page. Generally speaking, font sizes and types need to be harmonized.
- c. Figure S2A does not show cell cycle data as indicated in the description within the results.
- d. Many SIRF figures are essentially duplicated outcomes / different presentation of the same from a single type of experiment. While I appreciate the authors showing figures such as 5E where we can see the scatters identifying EdU positive S-phase cells - it is really the quantification and statistical analysis of the specific protein of interest (in that case RNF169) in Figure 5D that matters and is more helpful. I would recommend moving most instances of these EdU vs DAPI charts to the supplemental data - perhaps retaining only the first occasion Figure 1B versus 1A to demonstrate how this is being done.

R: We will carefully consider each suggestion and reorganize the figures accordingly.

10. Why is the experiment in Figure 1H only done with n=2? N=3 is considered minimum acceptable.

R: Despite the fact that for this type of experiment n=2 is considered acceptable if similar results are obtained, as also done in the original article (Gaggioli et al, NCB 2023), we will provide a third replicate of ChromStretch analysis.

11. In Figure 1K - the experiment shows a reduction after 1h post HU removal. How long does it take for signal to reach background levels?

R: We did not measure later time points.

12. On page 6, line 18, it would be better to say "reduced" H2Bk120ub is dispensable for XYZ, as based on the immunoblot controls the H2Bk120ub is clearly not entirely gone, as is implied.

R: We will fix it.

13. On page 8 line 4, the authors employ a "data not shown" statement. As I understand it, it

is not longer allowed to do this. This happens again on Page 10 line 27. Please show the data.

R: We did not include RAD51 data as we do not have triplicates of it.

14. The data in figure 4D-E commenting on H2Bub signal differences requires quantifying somehow.

R: We will provide quantification.

15. On page 10 line 18, using the word "significantly" immediately requires the author state the actual p-value for the condition they are claiming is statistically significant within the text.

R: We will add this information.

Prof. Lorenza Penengo
Universität Zürich
Institute of Molecular Cancer Research
Strickhofstrasse 40a
Zurich 8057
Switzerland

14th May 2025

Re: EMBOJ-2025-120711
H2BK120ub and its reader RNF169 sequentially regulate replication fork remodeling and stability

Dear Lorenza,

Thank you again for sending your revision plan and discussing it with me in detail. Since you appear to be in a good position to address/clarify not only the technical concerns of the referees, but also to extend the study with regard to the two main conceptual aspects of RNF169 roles and of fork reversal/remodeling, I would be happy to consider this study further for EMBO Journal publication. I am therefore now formally inviting you to prepare a new version revised along the lines presented in your point-by-point revision proposal, and to resubmit it using the link below.

Please keep in mind that it is our policy to allow only a single round of (major) revision, and please do update me should there be any unexpected problems with the revisions, or should you require an extension beyond the default 3-months deadline. As always, competing manuscript published during the course of this revision will not affect our final decision on your study. Finally, please note the detailed information and guidelines on how to prepare a revision below (and in our online Guide to Authors) - closely adhering to them shall greatly facilitate the editorial process at the time of resubmission.

Thank you again for the opportunity to consider this work, and I look forward to receiving your revision in due time.

With kind regards,

Hartmut

9) To facilitate reproducibility and cross-laboratory adoption of methodologies, please structure the Materials & Methods section as outlined in our guide to authors, including a completed Reagents and Tools Table that can be downloaded from our author guidelines as well (<https://www.embopress.org/page/journal/14602075/authorguide#structuredmethods>).

10) Digital image enhancement is acceptable practice, as long as it accurately represents the original data and conforms to community standards. If a figure has been subjected to significant electronic manipulation, this must be clearly noted in the figure legend and/or the 'Materials and Methods' section. The editors reserve the right to request original versions of figures and the original images that were used to assemble the figure. Finally, we generally encourage uploading of numerical as well as gel/blot image source data; for details see: embopress.org/page/journal/14602075/authorguide#sourcedata

In the interest of ensuring the conceptual advance provided by the work, we recommend submitting a revision within 3 months (12th Aug 2025). Please discuss the revision progress ahead of this time with the editor if you require more time to complete the revisions. Use the link below to submit your revision:

Link Not Available

We are grateful to the reviewers for their valuable time and insightful feedback on our manuscript. We have given thorough consideration to all comments and suggestions, which have substantially contributed to strengthening the quality of the work.

A detailed, point-by-point response to each comment is provided below.

REVIEWER COMMENTS

Referee#1

This manuscript presents a study investigating the role of H2BK120 monoubiquitination (H2BK120ub) and its E3 ligase RNF20 in replication fork remodeling. It further explores the function of RNF169 as a proposed reader of H2BK120ub at stalled forks, contributing to the protection of nascent DNA from degradation. These findings aim to expand our understanding of chromatin-associated regulation of replication stress responses.

While the study uncovers a novel chromatin modification pathway influencing fork remodeling, several mechanistic gaps and interpretative overextensions limit the current impact of the work. In particular, the phenotypic divergence between RNF20 and RNF169 depletion is not reconciled, challenging the proposed sequential model. Therefore, publication in the current form is not recommended.

Major concerns

1. The manuscript proposes that H2BK120ub (via RNF20) recruits RNF169 to stressed forks to stabilize reversed structures. However, the phenotypes of RNF20 and RNF169 depletion differ significantly:

RNF20 depletion prevents fork slowing and suppresses degradation in BRCA2-deficient cells, suggesting a role in promoting fork reversal.

In contrast, RNF169 depletion leads to fork degradation, indicating a role in protecting reversed forks.

These divergent outcomes imply that RNF20 and RNF169 act in functionally distinct, possibly independent branches of the fork remodeling process. The current data do not support a sequential epistasis model, as implied by the title.

R: We appreciate the reviewer's comment and the opportunity to provide additional clarification regarding our model.

We note that the different outcome of RNF20- and RNF169-depletions does not exclude or contradict that the two factors may act at different steps of the same process. Inactivation of

factors that promote fork slowing and reversal is not typically associated with defects in fork protection. Conversely, several factors that specifically protect reversed fork from degradation are not expected (and were not shown) to display defective fork slowing and remodelling. Numerous examples of both scenarios have been previously reported (see e.g. Mijic et al., Nat Comms 2017; Berti et al., Nat Comms 2020; Liu et al., Sci Adv 2020). Perhaps the most notable example is the opposite effect observed upon RAD51 and BRCA2 depletion. In this case, BRCA2 depletion leads to a fork degradation phenotype that is not shared, and in fact rescued by RAD51 depletion. Multiple groups have shown that this reflects the BRCA2-independent role of RAD51 in the formation of reversed forks, which then act as entry points for nucleolytic degradation in BRCA2-defective cells. Hence, RAD51 and BRCA2 act sequentially in promoting reversed fork formation and protection, despite strikingly different molecular phenotypes.

We envision a similar scenario for RNF20/RNF169: RNF20-H2BK120ub are required to promote/stabilize reversed forks and their inactivation does not cause degradation of nascent DNA, but rather unrestrained fork progression mediated by the RECQ1 activity. Conversely, RNF169 binds the H2BK120ub mark and contributes to protecting nascent DNA from degradation on previously reversed forks. Hence, similarly to BRCA2 (and many other fork protection factors), RNF169 inactivation does not affect fork slowing, but does compromise protection of stalled replication forks.

To further support this point, we have performed additional experiments (see below).

2. The authors demonstrate H2BK120ub accumulation at stalled forks and show that RNF20 depletion impairs fork slowing. However, no mechanistic insight is provided into how H2BK120ub facilitates fork reversal, such as: visualization of reversed forks (e.g., EM, S1-seq), interactions with RAD51 or fork remodeling enzymes, effects on chromatin accessibility or nucleosome positioning.

This is especially important in light of growing evidence that chromatin context dictates whether forks reverse or undergo repriming.

The authors are encouraged to focus on uncovering how RNF20-H2BK120ub contributes to fork reversal at the chromatin level, potentially through structural or genomic chromatin assays.

Clarifying this would substantially elevate the manuscript's impact.

R: As suggested by the reviewer, in collaboration with Massimo Lopes' group, we performed EM analysis to assess the effect of RNF20 depletion on the formation of reversed forks. In three independent experiments, we observed a consistent and significant reduction in the number of detectable reversed forks in HU-treated RNF20-depleted cells (NEW Fig. 3G-I; Fig. S3I).

We also evaluated the localization of RAD51 at stressed replication forks by SIRF upon siRNF20 and observed that RNF20 depletion did not cause major changes in the localization of RAD51 on nascent DNA (NEW Fig. S3G,H).

3. The manuscript refers to RNF169 as a "reader" of H2BK120ub at replication forks. However: The supporting assays (Fig. 4C-E) were conducted in cell lines under overexpression, without replication stress conditions.

The binding context is not demonstrated at endogenous expression levels or during active fork remodeling.

While the in-cell interaction is suggestive, the term "reader" carries the implication of context-specific and functionally relevant recognition. Without supporting evidence under stress conditions, this claim remains premature.

The authors should either:

Provide additional evidence of replication stress-dependent recruitment or binding (e.g., ChIP-seq, CUT&RUN, or co-localization with fork-specific markers under HU/CPT treatment), or Reframe RNF169 as a candidate effector that is functionally relevant at forks, without asserting definitive H2BK120ub reading.

R: We appreciate the reviewer's comments, which prompted us to investigate more in this direction. The levels of endogenous RNF169 are rather low and the biochemical analysis as done in Fig. 4 would not be possible using endogenous RNF169. On the other hand, we did monitor endogenous RNF169 by IF technique and observed that endogenous RNF169 is present on nascent DNA (RNF169 SIRF) and accumulates upon replication stress in a manner dependent on RNF20 and RAD51 (Fig. 5A,B,D; S5A-D,F).

Moreover, to further establish whether RNF169 is a specific reader of H2BK120ub in the chromatin context or rather interacts with H2BK120ub because of its ubiquitin binding domain (which was also a concern of Reviewer 2), we generated an additional mutant of RNF169 targeting the LR motif, previously shown to mediate the binding to the acidic patch of nucleosomes. Notably, inactivation of the LR motif highly reduced the interaction with NCPs-carrying H2BK120ub (NEW Fig. 4C) in vitro. In line with this, we also found that the proximity of H2BK120ub with ectopically expressed Myc-RNF169 WT, measured by PLA, is markedly reduced upon expression of the MIU2* or LRM* mutants (NEW Fig. 5C). Overall, this new evidence - combined with the results already included in the first submission, strongly supports RNF169 a role of RNF169 at the replication forks under stress conditions.

Additional suggestions

Clarify whether H2BK120ub functions in parallel or coordination with other chromatin features, such as H3K9me3, in regulating PRIMPOL access and fork reversal.

Explore whether RNF20 influences fork remodeling enzymes or replisome components, providing mechanistic insight into how H2BK120ub alters fork dynamics.

R: In collaboration with the group of Nitika Taneja, we performed ChromStretch upon siRNF20 and found that RNF20 depletion strongly decreases the presence of H2BK120ub on chromatin but does not affect the H3K9me3 mark (NEW Fig. 1H; S1L,M; n=3). We also tested whether the

inhibition of G9a impacts on H2BK120ub and observed no major effect (Fig. 1A,B for Reviewers only; n=1).

Last, we investigated whether the unrestrained fork progression phenotype observed upon siRNF20 depends on PRIMPOL or RECQ1. While the data obtained upon PRIMPOL depletion were inconclusive and are not reported in the revised manuscript, depletion of RECQ1 significantly rescued the unrestrained fork progression associated with RNF20 depletion upon CPT treatment (NEW Fig. 3J). These data clarify that this mark is essential to modulate the restart (and not the formation) of reversed forks, adding an important mechanistic aspect to the manuscript, which is also integrated in the revised model.

Referee#2

In this study, Duzanic et al. report a function of histone H2B mono-ubiquitination at K120 (H2B-K120Ub), a key histone mark with important roles in promoting transcription and DNA repair, in replication stress responses and replication fork dynamics in human cells. They show that H2B-K120Ub catalyzed by the E3 ubiquitin ligase RNF20 accumulates at a subset of replication forks in response to fork stalling induced by hydroxyurea (HU). While RNF20 is dispensable for normal DNA replication and fork protection in response to HU treatment, its depletion impairs reduced replication fork progression after mild replication stress induced by treatment with camptothecin (CPT). Moreover, RNF20 knockdown suppresses checkpoint activation upon CPT or HU treatment and rescues the replication fork protection defect caused by BRCA2 depletion, involving a potential role of RNF20 in promoting replication fork reversal. The authors go on to show that H2B-K120Ub provides a docking platform at stalled forks for the E3 ubiquitin ligase RNF169, which has previously been implicated in DNA double-strand break repair. Finally, they show that RNF169 depletion has no impact on replication fork speed under normal and stressed conditions but impairs fork protection after HU treatment, which can be rescued by concomitant RNF20 knockdown. Based on these data, the authors propose a model wherein replication stress induces fork-associated RNF20-dependent H2B-K120Ub ubiquitination, driving fork reversal and RNF169 recruitment to protect nascent DNA from nucleolytic degradation.

The identification of a role of H2B-K120Ub in replication fork remodeling and stability in human cells is interesting and aligns with previous studies in yeast establishing a function of this histone mark in replication stress responses. In its present form, however, the key conclusions of the manuscript are not fully supported by the data and the mechanistic basis of how RNF20-dependent H2B-K120Ub formation promotes fork remodeling and stability via RNF169 and possibly other factors remains far from clear. Clarification and extension of several key points, as elaborated below, would clearly strengthen this study.

We appreciate the constructive comments and suggestions of this reviewer, which have substantially contributed to strengthening the quality of the work.

Specific points:

1. All experiments probing the impacts of RNF20 depletion were performed using a single siRNA. At least some key phenotypes should be validated with an independent RNF20 siRNA or a different RNF20 depletion modality and, ideally, rescue experiments involving complementation with an RNF20 expression construct. The study would also benefit from addressing whether depletion of RNF40, the binding partner of RNF20 in the H2B-K120 E3 ubiquitin ligase complex, recapitulates key impacts of RNF20 knockdown.

R: We have previously tested T47D shRNF20 cells (kindly gifted by Prof Moshe Oren), but they exhibit low viability likely because RNF20 is an essential gene, and therefore they could not be used for DNA replication studies. For this reason, we preferred to adopt the siRNA approach in our experiments, which allowed us to achieve reduced levels of RNF20 but still good cell fitness. We have now included an additional siRNA targeting RNF20, and the siRNA targeting RNF40, as proposed by the reviewer. Both of them confirmed the effects we had reported with the first siRNA for RNF20, hence strongly consolidating our conclusions (NEW Fig. S1G,H; 2F; S2E). The complementation approach, although ideal, would be problematic, both for time limitations and because the over-expression of the RNF20 can exert detrimental effect on cell viability and should be strictly titrated.

2. It remains unclear whether the impacts of RNF20 depletion on replication fork dynamics are primarily related to direct (H2B-K120Ub deposition at forks) or indirect (H2B-K120Ub-dependent transcriptional impacts) effects, considering that replication stress increases both fork-associated and total nuclear H2B-K120Ub abundance (Fig. 1A-E). In this context, the observation that RNF20-depleted cells display a defective response to CPT but not HU (Fig. 2E-G) seems at odds with the notion that HU but not CPT increases H2B-K120Ub abundance (Fig. 1A-E).

R: RNF20 is known to regulate transcription, but it does not trigger a universal decline in transcript levels when depleted. Instead, multiple studies report that RNF20 loss results in selective, gene-specific changes. For instance, Shema et al. (Gen&Dev, 2008) found that most genes maintain similar expression after RNF20 knockdown, a conclusion supported by Xie et al. (Gen Biol, 2017), who observed little change in genes with high H2Bub1. Moreover, to avoid the detrimental effect of acute RNF20 depletion on cell viability and DNA synthesis, we employ experimental conditions that reduce but do not completely abolish RNF20 (and H2BK120ub) levels. Hence, the overall effect on gene transcription is even more limited. To assess the effect of RNF20 depletion compared to acute transcription inhibition, we compared the levels of H2BK120ub in control, RNF20-depleted and cells treated with transcription inhibitors (DRB and TRP) by immunoblot (Fig. 2 for Reviewers only).

Regarding the use of CPT and HU, it should be considered that these drugs have been used in different experiments at different concentrations and for different purposes. High HU (4mM, 2h) stalls forks and allows to assess the integrity of nascent DNA (fork degradation, RNF169-controlled step). Conversely, low CPT (100 nM, 1h) is permissive for further fork progression, and has been consistently used to induce active fork slowing upon mild replication stress, allowing to assess genetic contributions to early steps of fork remodeling (e.g. fork reversal), which turned out being the role for RNF20/H2BK120ub. Although the levels of the modification are admittedly different in the two cases, this may well reflect the extent of fork stalling and replication stress (clearly more pronounced when all forks are stalled by nucleotide depletion); nonetheless, the modification appears to be functionally relevant in both cases.

3. Page 7, line 14-16: The authors speculate that "Intriguingly, the chromatin fibers data suggested that only a fraction of forks accumulated H2BK120ub, possibly reflecting the subset of remodeled forks (Fig. 1G-I; S1F-H)." Can they provide more direct evidence in support of this hypothesis, e.g. by testing whether replication stress-dependent H2B-K120Ub accumulation specifically coincides with markers of reversed forks?

R: Unfortunately, specific markers of reversed forks are not available. We acknowledged the concerns of this reviewer and removed part of the sentence ('possibly reflecting the subset of remodeled forks') that sounds too speculative.

4. Fig. 3F: The impact of siRNF20 on blocking checkpoint activation is notable. The authors suggest that this may be due to defective fork reversal, but this remains speculative. Can they provide further evidence substantiating this idea?

R: To provide stronger evidence on the regulation of the fork reversal by RNF20, we collaborated with Massimo Lopes' group and performed EM analysis to assess the effect of RNF20 depletion on the formation of reversed forks. In three independent experiments, we observed a consistent and significant reduction in the number of detectable reversed forks in HU-treated RNF20-depleted cells (NEW Fig. 3G-I; Fig. S3I). Nonetheless, we agree with the reviewer that the defect in checkpoint signaling may not necessarily reflect the defective fork reversal we observe, and is not consistently observed when fork reversal is prevented by other genetic perturbations. Hence, in this revised manuscript, we have toned down this possible interpretation (page 8, lines 24-26).

5. Fig. 4: It is difficult to conclusively ascertain from these experiments whether RNF169 is a specific reader of H2B-K120Ub or merely pulls down H2B-K120Ub-modified NCPs in vitro because it is a ubiquitin-binding protein. It would be useful to test whether proteins like RNF168 and RAD18 that recognize H2A-K13/K15Ub similar to RNF169 also pull down H2B-K120Ub-modified NCPs, or whether this is exclusive to RNF169. Does mutation of the LRM motif adjacent to MIU2 impair RNF169 binding to H2B-K120Ub-modified nucleosomes?

R: This is certainly an important point. As suggested by the reviewer, to further assess if RNF169 binds specifically H2BK120ub embedded in the chromatin context, we tested whether mutation in the LR motif, which helps target proteins to damaged chromatin, impairs the binding of RNF169 to H2BK120ub, by performing pulldown assay as in Fig. 4B. Notably, inactivation of the LR motif highly reduced the interaction with NCPs-carrying H2BK120ub (NEW Fig. 4C) in vitro. In line with this, we also found that the proximity of H2BK120ub with ectopically expressed Myc-RNF169 WT, measured by PLA, is markedly reduced upon expression of the MIU2* or LRM* mutants (NEW Fig. 5C).

As for the other suggestion, in our opinion testing whether other proteins known to bind H2AK15ub, like RNF168 or RAD18, can also bind H2BK120ub will not help to ascertain the specificity of RNF169 towards chromatin, since it is possible that other DDR proteins can bind the same histone mark.

6. The studies of the potential role of RNF169 as an effector of replication stress responses downstream of H2B-K120Ub formation (Fig. 5) appear quite premature and need to be further developed. Given the authors' observation that RNF20 is important for recruiting RNF169 to stressed replication forks, it seems somewhat counterintuitive that RNF169 depletion compromises fork protection while RNF20 knockdown has no impact in itself but rescues the fork protection defect in RNF169-depleted cells. Further studies are clearly needed to better delineate the possible fork-protective role of RNF169 and how this relates to RNF20-dependent H2B ubiquitination.

R: The fact that RNF20 and RNF169 depletion results in different outcomes does not necessarily contradict that the two factors may act at different steps of the same process. Please refer to the next point for a more elaborated response.

Regarding the second point, this study is mainly centered on the role played by H2BK120ub histone mark, promoted by RNF20 (and RNF40), in the DNA replication stress and fork dynamics. While investigating molecular events linked to H2BK120ub at stalled forks, we discovered a new function of RNF169 in DNA replication stress and fork protection and started characterizing it, to uncover more downstream events primed by H2BK120ub. Admittedly, a profound molecular understanding of RNF169 role in fork protection will require further investigations, but we feel that this lies beyond the scope of this manuscript.

Nonetheless, to consolidate the role of RNF169 in fork protection, we performed complementation experiments in RNF169 depleted cells and confirmed that the doxycycline induced expression of the siRNA-resistant forms of RNF169 WT is able to restore fork stability (NEW Fig. 5H-J).

7. Following on from the point above: The model put forward by the authors (Fig. 6) does not seem to be fully supported by the data. The authors state that "the accumulation of H2BK120 in turn serves as a docking platform for the recruitment of RNF169 to the reversed replication forks,

which protects nascent DNA from unscheduled nucleolytic degradation" (page 13, line 21-23), but this appears inconsistent with the observation that RNF20 knockdown has no impact on fork protection unlike RNF169 depletion despite RNF20 is required for recruiting RNF169 to stressed forks.

R: The fact that the outcomes of RNF20 and RNF169 depletion are different does not exclude they work in the same process, as already discussed above. Inactivation of factors that promote fork slowing and reversal is not typically associated with defects in fork protection. Conversely, several factors that specifically protect reversed fork from degradation are not expected (and were not shown) to display defective fork slowing and remodeling. Numerous examples of both scenarios have been previously reported (see e.g. Mijic et al., Nat Comms 2017; Berti et al., Nat Comms 2020; Liu et al., Sci Adv 2020). Perhaps the most notable example is the opposite effect observed upon RAD51 and BRCA2 depletion. In this case, BRCA2 depletion leads to a fork degradation phenotype that is not shared, and in fact rescued by RAD51 depletion. Multiple groups have shown that this reflects the BRCA2-independent role of RAD51 in the formation of reversed forks, which then act as entry points for nucleolytic degradation in BRCA2-defective cells. Hence, RAD51 and BRCA2 act sequentially in promoting reversed fork formation and protection, despite strikingly different molecular phenotypes.

We envision a similar scenario for RNF20/RNF169: RNF20-H2BK120ub are required to promote/stabilize reversed forks and their inactivation does not cause degradation of nascent DNA, but rather unrestrained fork progression mediated by the RECQ1 activity. Conversely, RNF169 binds the H2BK120ub mark and contributes to protecting nascent DNA from degradation on previously reversed forks. Hence, similarly to BRCA2 (and many other fork protection factors), RNF169 inactivation does not affect fork slowing, but does compromise protection of stalled replication forks.

Additional points

8. The PCNA input blot in Fig. 1F is of poor quality and should be replaced.

R: We reloaded the input samples and included a higher quality blot in the revised manuscript.

9. Page 6, line 7-10: "The results showed that RNF20 depletion did not cause a dramatic alteration of the cell cycle progression, but only a limited delay of cell entry into the S-phase and consequent accumulation in G1-phase (Fig. S2A), thereby the defective checkpoint activation is likely due to the loss of RNF20.". It is unclear what defective checkpoint activation the authors are referring to here - please consider re-phrasing.

R: We re-phrased it and omit part of the sentence.

10. Whether the observed increase in H2B-K120Ub abundance caused by RNF169 overexpression (Fig. 4D,E) has any functional significance is not clear. Does RNF169 depletion

reduce H2B-K120ub levels? Does RNF169 overexpression have any impact on DNA replication and fork integrity?

R: We consistently observed increased H2BK120ub upon RNF169 over-expression in our experiments (mainly in HEK293T cells), while no H2BK120ub reduction was observed upon depletion of RNF169. The increase in H2BK120ub levels following RNF169 overexpression is an interesting observation and could suggest the speculative hypothesis that RNF169 binding protects H2BK120ub from the action of deubiquitinating enzymes. It is difficult to verify whether this has functional relevance, as overexpression of RNF169 at the levels necessary to increase H2BK120ub (as in HEK293T cells) is quite toxic to cells, making it impossible to perform functional studies.

11. The organization of panels in some figures is not logical (e.g. A,B,C,F,D,E... in Fig. 3 and A,B,D,C,E,F,H,I,G... in Fig. 5).

R: We have reorganized them in a more logical manner.

Referee#3

In their study "H2BK120ub and its reader RNF169 sequentially regulate replication fork remodeling and stability" Duzanic and colleagues present a combination of cell and molecular biological data to describe a role for the ubiquitination of H2B at K120 by RNF20 in the recruitment of RNF169 at sites of stalled replication forks, suggesting that the recruitment of RNF169 by this mechanism promotes fork stability by protecting against nuclease digestion of otherwise vulnerable stalled fork DNA.

Overall I am supportive of this study, although I have a variety of recommendations and concerns that would be good to address.

Major concerns and suggestions

1. The study utilizes lower dose CPT and higher dose HU to produce scenarios of replication stress with differing severity. To me, a vital control that is not experimentally verified is the relative amount of DNA double strand breaks (DSBs) that occur (or not) following each of these treatments in the U2OS cells specifically used in this work, and confirming DSB levels for each siRNA treatment the team deploys to ask various questions. Controlling for DNA replication fork collapse into DSBs is highly relevant here, as it influences how the reader interprets any experimental outcomes where loss of DNA damage signalling and/or repair factors are involved.

R: We are grateful to this reviewer for raising this important point, which may indeed deserve some mention in the main text of our manuscript. Treatment with CPT (100 nM) and HU (4 mM) have been widely used for studies concerned with replication fork remodeling and stability. We certainly do not exclude (in fact believe) that minimal levels of DSBs are induced, in parallel to a widespread effect of fork slowing (CPT) and stalling (HU). However, we have tested the levels of DSBs by monitoring gH2AX by immunoblot upon different treatments (CPT and HU) in siLUC, siRNF20, siRNF169 and siRAD51; none of these genetic perturbations – not even the widely characterized depletion of RAD51, a central HR factor with key roles in DSB repair – markedly affected DSB signaling (Fig. 3A-C for Reviewers only). Moreover, we also measured gH2AX signals in EdU-positive cells upon EdU-pulse labeling following treatment with CPT (100 nM, 1 h) and HU (4 mM, 2 h) and again, we did not observe major increase of signal upon siRNF20, siRNF169 and siRAD51 (Fig. 4 for Reviewers only). Overall, these data suggest that the associated DSBs are very unlikely to underlie the differential effects on fork progression and stability reported in our work upon those different conditions (see also our response to point 2 here below). Importantly, to avoid raising similar confusion and concerns in the readers of the manuscript, we have now included in the main text an explicit comment to this point (page 4, lines 27, 30-32).

2. Throughout this work, the depletion of RAD51 is used (and interpreted) as a means to modulate fork reversal and largely discussed with that in mind. My concern is that, if cellular conditions are also coincident with sufficient replication-associated DSBs that also vary with experimental conditions, then the relative abundance of RAD51 will also exert an influence of rates of DSB repair and therefore steady state DSB signalling to ATR and ATM, which will in turn influence RNF20 activation and downstream effects. Hence, it is important that the authors control for DSB formation across all cellular conditions, and also discuss what implications that DSB signalling might have on the processes they are investigating.

R: By now numerous reports have looked at the impact of RAD51 depletion and/or its mutation in specific domains, as well as inactivation of accessory factors modulating its recruitment on replication fork progression and stability, using the very same CPT and HU treatments that we used here (Zellweger et al., JCB 2015; Mijic et al., Nat Comms 2017, Lemacon et al., Taglialatela et al., Mol Cell 2017; Kolijnjivadi et al., Mol Cell 2017; Berti et al., Nat Comms 2020; ; Liu et al., Sci Adv 2020; Krishnamoorthy et al., Mol Cell 2021; Liu et al., Science 2023; ...). In all these studies, the possible marginal induction of DSBs – and therefore their further accumulation upon inactivation of specific HR activities – did not prevent discriminating the relative contributions of the investigated factors in replication fork remodeling and stability. Moreover, as mentioned above, the fact that even RAD51 inactivation does not significantly increase the observed signaling upon these treatments supports the idea that differential effects on fork remodeling or restart cannot reflect marginal DSB repair accumulation in these experimental conditions.

3. While some siRNA efficiency controls are shown throughout this work, many are missing altogether or shown for only some experiments but not others. In my view, siRNA efficiency and specificity controls are required for all targets in all different experimental contexts and need to be shown first. For example, as far as I can see, we are never shown controls that demonstrate how well the siRNA targeting RAD51 works.

R: We thank the reviewer for mentioning this point. In all our experiments the levels of depletion efficiency are verified. We now provided all these controls in the revised manuscript.

4. Further to my concern above, proving the specificity of siRNA effects using "add-back" siRNA resistant wildtype (and, if relevant, catalytic-dead versions) of key factors - especially RNF20, RAD51, BRCA2 - has not been done. Given the reliance of this study on siRNA to demonstrate mechanism, these controls are vital as everything the authors claim rests on these siRNA being robust. I would also encourage use of gene-edited cells in place of siRNA to prove key experiments.

R: We performed complementation experiments in RNF169 depleted cells and found that the doxycycline induced expression of the siRNA-resistant forms of RNF169 WT is able to restore fork stability (NEW Fig. 5H-J).

Coming to RNF20, we have previously tested T47D shRNF20 cells (kindly gifted by Prof Moshe Oren), but they exhibit low viability likely because RNF20 is an essential gene, and therefore they could not be used for DNA replication studies. For this reason, we opted for the use of siRNAs in our experiments, which allowed us to achieve reduced levels of RNF20 maintaining good cell fitness. We have now included an additional siRNA targeting RNF20, and a siRNA targeting RNF40, which is required for the function and stability of RNF20 (NEW Fig. S1G,H; 2F; S2E). The complementation approach, although ideal, would be problematic, both for time limitations and because the over-expression of the RNF20 can exert detrimental effect on cell viability and should be strictly titrated.

5. The antibody controls for the H2BK120ub are demonstrated using immunoblot but then often deployed using immunofluorescence (IF). The outcome of antibody quality control between these methods is not always interchangeable, and very often an antibody that works well and is specific under denaturing conditions of a SDS-PAGE western blot is non-specific in IF. How can the authors be certain that their antibody is specific using microscopy? IF controls are needed.

R: We have tested the specificity of the H2BK120ub antibody upon RNF20 depletion both by immunoblot (Fig. S1A,B) and by IF (Fig. 1A, the H2BK120ub signal intensity in siLuc and siRNF20 is shown) already in the previous version of the manuscript. To make this important control more evident, we now included representative images of the IF and the quantification of the H2BK120ub in EdU positive and negative cells, previously pulse-treated with EdU (NEW Fig. S1C,D).

6. A key hypothesis from this work is that the RNF20 to RNF169 pathway is helping to protect stalled DNA replication forks against nucleolytic degradation. If correct, then depletion or inhibition of nucleases associated with this degradation would be expected to alleviate the need for these factors in fork stability and can be tested using the assays presented in this work. As this is a very logical experiment that I consider foundational to the final hypothesis, I would encourage this to be done as part of a revision.

R: We agree with this reviewer that assessing nuclease dependency of fork degradation in RNF169 defective cells would have been interesting. We note however that when this has been done for other fork degradation phenotypes, it was often found that multiple nucleases (primarily MRE11, DNA2, EXO1) are involved and often play partially redundant roles. Assessing the relative contribution of these nucleases in the context of RNF169 defects would have required identifying proper condition for single and multiple nuclease downregulation. Considering the overall amount of work required to address other suggestions and concerns in the limited time available for revision, we have preferred to prioritize experiments that we consider more central to the take home message of this manuscript.

7. A variety of labs have demonstrated that RNF20 function is co-dependent on RNF40 - is that the case here? Do the authors observe their primary phenotypes of interest to be the same or similar between siRNF20 and siRNF40?

R: We tested the key phenotypes (by DNA fiber assay and SIF analysis) upon RNF40 depletion and found similar effect (NEW Fig. S1G,H; 2F; S2E).

8. RNF20 mediated ubiquitination of chromatin promotes recruitment of chromatin remodelers, with several labs demonstrating that SNF2H is amongst these (most recently PMID: 37155876). Recruitment of SNF2H via RNF20 can exert influence over homologous recombination (e.g. PMID: 24357716) and DSB more generally, which matters in the context of DNA replication associated stress with the potential for DSB formation. Assessing the presence of (previously discovered) RNF20-dependent chromatin remodelers such as SNF2H could be evaluated using the SIF approach, and would substantially enhance this work by determining if those pathways apply more universally (i.e. everywhere RNF20 is active) or are perhaps more selective.

R: This is indeed a valuable point; however, due to time constraints, we were unable to perform this experiment, which remains a relevant direction for future studies.

Minor concerns:

9. Figures:

a. The placement of figure panels in this study are somewhat confusing to the reader. For example, the placement of Figure 3F and 5G are essentially non-sensical relative to panels that

should come earlier/later to the reader.

b. On many graphs, font sizes of labels on axes are very small and hard to read at 100% zoom on a standard page. Generally speaking, font sizes and types need to be harmonized.

c. Figure S2A does not show cell cycle data as indicated in the description within the results.

d. Many SIRF figures are essentially duplicated outcomes / different presentation of the same from a single type of experiment. While I appreciate the authors showing figures such as 5E where we can see the scatters identifying EdU positive S-phase cells - it is really the quantification and statistical analysis of the specific protein of interest (in that case RNF169) in Figure 5D that matters and is more helpful. I would recommend moving most instances of these EdU vs DAPI charts to the supplemental data - perhaps retaining only the first occasion Figure 1B versus 1A to demonstrate how this is being done.

R: We carefully considered each suggestion and reorganized the figures accordingly. We changed the order of different panels, increased the size of labeling and uniformed the style, corrected mislabeled figures, and moved the scatter plots showing cell cycle distribution of the IF/SIRF signals to the supplementary figures.

10. Why is the experiment in Figure 1H only done with n=2? N=3 is considered minimum acceptable.

R: For this type of experiment, and given the large number of events quantified in each replicate, a sample size of n=2 is generally considered acceptable when consistent results are obtained, as also demonstrated in the original publication (Gaggioli et al, NCB 2023). Nonetheless, we now provided a third replicate of ChromStretch analysis, as well as a new set of experiments performed upon siLUC and siRNF20 (NEW Fig. 1H; S1L,M).

11. In Figure 1K - the experiment shows a reduction after 1h post HU removal. How long does it take for signal to reach background levels?

R: We did not measure later time points, although this will surely be an important task for follow-up projects.

12. On page 6, line 18, it would be better to say "reduced" H2Bk120ub is dispensable for XYZ, as based on the immunoblot controls the H2Bk120ub is clearly not entirely gone, as is implied.

R: We re-phrased it as follows: 'indicating that reduced levels of H2BK120ub do not affect DNA replication under unperturbed conditions'.

13. On page 8 line 4, the authors employ a "data not shown" statement. As I understand it, it is not longer allowed to do this. This happens again on Page 10 line 27. Please show the data.

R: We did not include RAD51 data as the experiment was done only once (as a control in the first experiment) and did not bring more information. We now removed this sentence from the revised

manuscript. Regarding the second point, we now show the DNA fiber experiment of unrestrained for progression upon CPT in siRNF169 condition (n = 2; Fig. S5H,I).

14. The data in figure 4D-E commenting on H2Bub signal differences requires quantifying somehow.

R: We now included the quantification, as suggested by the reviewer.

15. On page 10 line 18, using the word "significantly" immediately requires the author state the actual p-value for the condition they are claiming is statistically significant within the text.

R: We agree with the reviewer and now added the p-value in the text.

Prof. Lorenza Penengo
Universität Zürich
Institute of Molecular Cancer Research
Strickhofstrasse 40a
Zurich 8057
Switzerland

29th Sep 2025

Re: EMBOJ-2025-120711R
H2BK120ub and its reader RNF169 sequentially regulate replication fork remodeling and stability

Dear Lorenza,

Thank you for submitting your revised manuscript to The EMBO Journal. It has now been re-reviewed by one of the original referees (see comments below), and I am happy to say that they considered the study substantially improved and the original concerns satisfactorily clarified. Following incorporation of the discussion points mentioned by the referee, we should therefore be ready to quickly proceed with acceptance and production of the article. However, there are at this point still a number of important editorial points that would need to be urgently addressed:

- First of all, we still need you to complete our author checklist (see download link below), and upload it with the final manuscript.
- Importantly, please upload all Figure files (main and Expanded View) as individual (image-only) files and with sufficient resolution/quality not just for production, but also to facilitate our routine pre-acceptance image checking. The current images are unfortunately all very pixelated.
- Please rename the "supplementary" figures into Expanded View Figures - naming/referencing them as "Figure EV1/2/3...", and including their legends at the end of the main manuscript text, after the main figure legends.
- Please adjust the order of the manuscript sections, and also make sure to use the correct section headers: Title page with complete author information, Abstract, Keywords, Introduction, Results, Discussion, Methods, Data Availability, Acknowledgements, Disclosure and Competing Interests Statement, References, Main Figure Legends, Tables, Expanded Figure Legends.
- On the title page, please make sure to label the corresponding author and to include corresponding author's email address.
- On the abstract page of the manuscript, please include 4-5 general keyword terms to enhance searchability.
- Please rename the Conflict of Interest section into "Disclosure and Competing Interests Statement", in accordance with our updated Guide to Authors (<https://www.embopress.org/competing-interests>)
- As we are switching from a free-text author contribution statement towards a more formal statement based on Contributor Role Taxonomy (CRediT) terms, please remove the present Author Contribution section and instead specify each author's contribution(s) directly in the Author Information page of our submission system during upload of the final manuscript. See <https://casrai.org/credit/> for more information.
- Please double-check to make sure to all relevant funding information in the manuscript is congruent with the info entered into our submission system. Currently missing in the submission system is:
NWO-Vidi funding (project no. 114122)
- Please move the Reagents and Tools table from the main article file, and upload it as a separate text file. Also, please make sure to adhere to the template table downloadable from our author guidelines:
<https://www.embopress.org/page/journal/14693178/authorguide#structuredmethods>
- Please carefully go through the reference list, which currently contains many wrongly formatted entries:
 - * many citations are incomplete, missing e.g. citation year, volume, and page/locator numbers
 - * some are wrongly listed as "preprints" and amended with DOI, even though they are regularly published. DOI information is only required for "advance online publications" that really do not have a formal citation information yet
- Please double-check the provided Source Data and Source Data Checklist for congruency: for Fig. 3 - SD is provided for panel 3A, listed in checklist is panel 3D.

- Please include a dedicated "Data Availability" section at the end of the Material and Methods (suggested wording: "The [structural coordinates | microarray | mass spectrometry] data from this publication have been deposited to the [name of the database] database [URL] and assigned the identifier [accession | permalink | hashtag]."). Should there no data deposition to public repositories linked to the study, this should still be stated as "This study includes no data deposited in external repositories."

- Please provide suggestions for a short 'blurb' text prefacing and summing up the study in two sentences (max. 250 characters), followed by 3-5 one-sentence 'bullet points' with brief factual statements of key results of the paper; they will form the basis of an editor-written 'Synopsis' accompanying the online version of the article. Please also upload a synopsis image, which can be used as a "visual title" for the synopsis section of your paper (maybe based on a simplified/condensed version of Fig 6?). The image should be in PNG or JPG format with the modest dimensions (!) of 550 x 300-600 pixels (width x height).

- Finally, during routine pre-acceptance checks, our data editors have raised the following queries regarding figures, data, and legends, which I would ask you to address (ideally using the Track Changes option):

1. Please define the annotated p values ****/**/**/* as well as provide the exact p-values for the same in the legend of figure 1H, J; 3I, S1H, J, K, L; S3 H, S5 I as appropriate.
2. Please note that the exact p values have to be provided in the legends of figures 1B, D, K; 2B, D, F; 3A, C, E, F, J; 5B, C, D, E, G, J, L; S1 D, S5 A
3. Please indicate the statistical test used for data analysis in the legends of figures 1H, J; 3I, S1 H, J, K, L; S3 H, S5 I
4. Please note that the box plots need to be defined in terms of minima, maxima, centre, bounds of box and whiskers, and percentile in the legends of figures 1H, J; S1 K, L
5. Please note that information related to n is missing in the legends of figure 1H, J
6. Please note that the error bars have to be defined in the legends of figure 3I

I am returning the manuscript to you for a final round of minor revision - hoping that you can resubmit a final version, comprehensively and definitively addressing each of these points, before the end of the week, in order to facilitate that the paper can swiftly go into production.

With kind regards,

Hartmut

- size of the scale bars that are mandatory for all micrograph panels
- the statistical test used to generate error bars and P-values
- the type error bars (e.g., S.E.M., S.D.)
- the number (n) and nature (biological or technical replicate) of independent experiments underlying each data point
- Figures may not include error bars for experiments with n<3; scatter plots showing individual data points should be used instead.

6) Please complete our Author Checklist, and make sure that information entered into the checklist is also reflected in the manuscript; the checklist will be available to readers as part of the Review Process File. A download link is found at the top of our Guide to Authors: [embopress.org/page/journal/14602075/authorguide](https://www.embopress.org/page/journal/14602075/authorguide)

8) Please note that supplementary information at EMBO Press has been superseded by the 'Expanded View' for inclusion of additional figures, tables, movies or datasets; with up to five EV Figures being typeset and directly accessible in the HTML version of the article. For details and guidance, please refer to: [embopress.org/page/journal/14602075/authorguide#expandedview](https://www.embopress.org/page/journal/14602075/authorguide#expandedview)

9) To facilitate reproducibility and cross-laboratory adoption of methodologies, please structure the Materials & Methods section as outlined in our guide to authors, including a completed Reagents and Tools Table that can be downloaded from our author guidelines as well (<https://www.embopress.org/page/journal/14602075/authorguide#structuredmethods>).

10) Digital image enhancement is acceptable practice, as long as it accurately represents the original data and conforms to community standards. If a figure has been subjected to significant electronic manipulation, this must be clearly noted in the figure legend and/or the 'Materials and Methods' section. The editors reserve the right to request original versions of figures and the original images that were used to assemble the figure. Finally, we generally encourage uploading of numerical as well as gel/blot image source data; for details see: [embopress.org/page/journal/14602075/authorguide#sourcedata](https://www.embopress.org/page/journal/14602075/authorguide#sourcedata)

In the interest of ensuring the conceptual advance provided by the work, we recommend submitting a revision within 3 months (28th Dec 2025). Please discuss the revision progress ahead of this time with the editor if you require more time to complete the revisions. Use the link below to submit your revision:

Link Not Available

Referee #2:

In the revised version of their manuscript, Duzanic et al. carried out a range of new experiments and addressed many of my original concerns, which collectively have strengthened this study considerably. I am therefore supportive of publication of the revised manuscript in The EMBO Journal. However, I believe the following point would be important to address before publication: In their point-by-point response, the authors provide a rationale rooted in the existing literature for how their data could be consistent with a model in which RNF169 acts as an effector of replication stress responses downstream of RNF20-dependent H2B-K120Ub formation despite RNF20 and RNF169 depletion have distinct functional impacts. To avoid confusion and facilitate readers' interpretation of the data, I would strongly recommend that a discussion along similar lines be included in the manuscript.